# KANomaly: Fourier-KAN-based Multi-Scale Patch Mixer for Multivariate Time Series Anomaly Detection

## Abstract

Multivariate Time Series Anomaly Detection (MTSAD) is crucial for system stability in domains such as industrial monitoring, yet rare events, nonlinear dependencies, and limited labels necessitate unsupervised methods. However, existing approaches struggle to model subtle anomalies and detect diverse patterns, as they rarely integrate explicit frequency-domain representations and rely on fixed-scale analysis. To address these limitations, we propose KANomaly, a novel model inspired by Fourier-based Kolmogorov–Arnold Networks (KANs) for MTSAD. The model incorporates three key innovations: (i) Fourier basis functions embedded within the KAN architecture to capture subtle periodic and spectral anomalies; (ii) a coarse-to-fine multi-scale patching strategy that enhances detection of both point and pattern anomalies; and (iii) a Fourier-KAN Mixer that aggregates information across channel, patch, and temporal dimensions to model complex local and global interdependencies. Extensive experiments demonstrate that KANomaly consistently outperforms state-of-the-art models on multiple real-world datasets, validating the effectiveness of each component.

## 1 Introduction

Multivariate Time Series Anomaly Detection (MTSAD) is crucial for system stability in domains such as industrial monitoring (Dix et al., 2021), healthcare (Li & Jung, 2023), and finance (Cao & Guo, 2024), where high-dimensional sensor data are collected at high frequency. Detection is challenging due to rare anomalies, nonlinear dependencies with delays, and limited labeled data, making unsupervised methods more suitable.

Recent unsupervised MTSAD studies have explored deep learning approaches, demonstrating strong performance. Specifically, Transformer-based models have gained popularity for their scalability and sequence modeling capabilities (Vaswani et al., 2017). However, most models implicitly learn temporal and inter-channel dependencies. Frequency-domain representations are typically limited to preprocessing or attention modules, lacking direct functional integration into the model architecture. Moreover, these models operate at a single, fixed scale, which limits their ability to detect both short-term point and long-term pattern anomalies. To address these limitations, we propose KANomaly, a model that integrates Fourier-based Kolmogorov–Arnold Networks (KANs) (Liu et al., 2025) to explicitly capture subtle frequency-domain anomalies such as periodic shifts and spectral disturbances.

KANomaly introduces three key architectural innovations that distinguish it from prior approaches. First, to explicitly capture frequency characteristics within the model architecture, we integrate Fourier basis functions into the KAN, a function approximation framework grounded in the Kolmogorov–Arnold Representation Theorem (Kolmogorov, 1961). This enables the model to explicitly represent periodic and spectral anomalies, which are typically challenging to detect in the time domain and are often only implicitly captured by conventional deep learning models. Second, to detect anomalies occurring at varying temporal scales, we propose a coarse-to-fine multi-scale patching strategy. Coarse patches are suited for capturing long-term and structural patterns, while fine patches enhance sensitivity to short-term and localized anomalies. This overcomes the limitations of fixed-scale models that struggle to address both types simultaneously. Third, to further enhance representational power and jointly capture local and global dependencies, we introduce a mixing

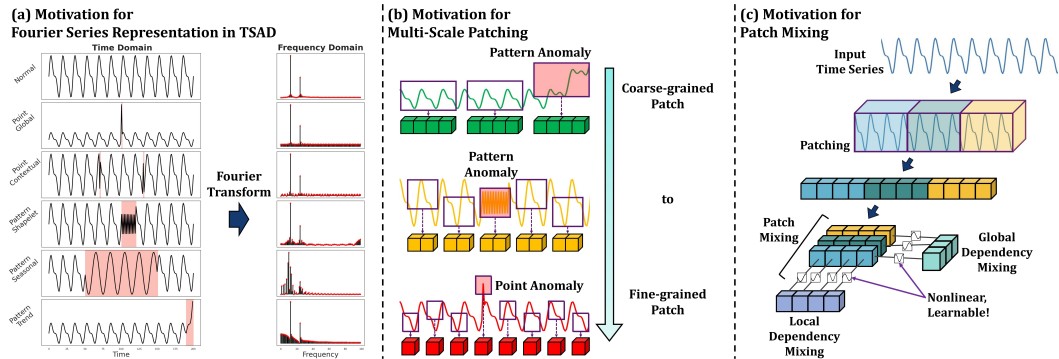

Figure 1: Motivation and Conceptual Overview of KANomaly. (a) Frequency-domain representations effectively capture subtle anomalies that are often missed in the time domain. (b) Coarse-to-fine multi-scale patching sequentially detects pattern anomalies and refines detection of point anomalies. (c) Patch mixing captures complex global-local dependencies.

module across channel, patch, and temporal dimensions. This module enables the model to effectively learn rich interactions across axes, capturing the complex interdependencies inherent in high-dimensional time series data.

Based on these design principles, KANomaly can precisely detect various types of time series anomalies simultaneously. Our contributions can be summarized as follows:

- We propose KANomaly, the first MTSAD model that integrates Fourier basis functions into KAN, enabling explicit modeling of frequency-sensitive anomalies typically missed by conventional architectures.

- To capture pattern and point anomalies, we propose a coarse-to-fine multi-scale patching strategy that detects long-term patterns with coarse patches and refines detection of short-term deviations using fine patches.

- We design the Fourier-KAN Mixer, a frequency-aware module that captures local-global dependencies across channel, temporal, and patch dimensions through nonlinear mixing and explicit spectral representation.

- KANomaly outperforms state-of-the-art (SOTA) models on five benchmark datasets; ablation studies validate that each architectural component substantially contributes to anomaly detection effectiveness.

## 2 RELATED WORK

### 2.1 MULTIVARIATE TIME SERIES ANOMALY DETECTION

Both traditional and deep learning methods have been applied to MTSAD. Traditional methods include Local Outlier Factor (Breunig et al., 2000), One-Class SVM (Manevitz & Yousef, 2001), and Isolation Forest (Liu et al., 2008). Recent deep learning methods fall into clustering-, reconstruction-, and contrastive learning-based approaches. Clustering-based methods map normal data into a compact space and identify outliers as anomalies; examples include DeepSVDD (Ruff et al., 2018) and THOC (Shen et al., 2020). Reconstruction-based methods detect anomalies based on reconstruction error after compressing and reconstructing the input, with models such as DAGMM (Zong et al., 2018), LSTM-VAE (Park et al., 2018), OmniAnomaly (Su et al., 2019), BeatGAN (Zhou et al., 2019), and USAD (Audibert et al., 2020). Transformer-based models, including Anomaly Transformer (Xu et al., 2021), TranAD (Tuli et al., 2022), MEMTO (Song et al., 2023), and TFMAE (Fang et al., 2024), have also demonstrated strong performance. In contrastive learning-based approaches, DCdetector (Yang et al., 2023) maximizes representational differences using dual attention. While effective, existing methods often struggle with periodicity or complex pattern anomalies. In contrast, we combine Fourier-based KAN with multi-scale patching to detect both point and pattern anomalies.

## 2.2 KOLMOGOROV-ARNOLD NETWORKS

Kolmogorov–Arnold Networks (KANs) (Liu et al., 2025) are neural networks inspired by the Kolmogorov–Arnold representation theorem (Kolmogorov, 1961), which states that any multivariate continuous function can be approximated using a combination of univariate functions. Unlike conventional Multi-Layer Perceptrons (MLPs) with fixed activation functions, KANs employ learnable univariate basis functions, offering greater flexibility and adaptability. The architecture supports various basis types, such as B-spline, Chebyshev, and Fourier, enabling more effective modeling of complex patterns than traditional MLPs. KANs have recently been applied to diverse domains, including physics-informed neural networks (Wang et al., 2025), scientific machine learning (Liu et al., 2024), partial differential equation approximation (Zhang et al., 2025), image segmentation (Li et al., 2025), and time series forecasting (Huang et al., 2025). However, their application in time series analysis has largely focused on forecasting, with recent work such as KAN-AD (Zhou et al., 2025) fundamentally designed as a Univariate Time Series Anomaly Detection (UTSAD) framework. To the best of our knowledge, KANomaly is the first Fourier-KAN-based method explicitly tailored for MTSAD. We propose a novel KAN-based architecture that explicitly models temporal dynamics and periodic anomalies in an unsupervised setting.

## 2.3 FREQUENCY-AWARE TIME SERIES ANALYSIS MODELS

Frequency-domain information effectively captures periodicity and trends, making it a valuable time series representation. Autoformer (Wu et al., 2021) improves forecasting performance through an Auto-Correlation mechanism and time series decomposition, while FEDformer (Zhou et al., 2022b) combines Fourier-based attention with a mixture-of-experts architecture. FiLM (Zhou et al., 2022a) enhances low-frequency representational capacity using the Legendre Projection Unit and Frequency Enhanced Layer. TimesNet (Wu et al., 2022) decomposes time series into multiple periods and transforms them into 2D tensors to effectively capture both intra- and inter-period variations. FreTS (Yi et al., 2023) is a frequency-domain MLP that demonstrates the contribution of frequency representations to performance improvement. FITS (Xu et al., 2023) efficiently learns frequency representations with a small number of parameters, and FSatten (Wu, 2025) enables accurate modeling of inter-series dependencies using a frequency-based attention module. CATCH (Wu et al., 2025) extracts patches in the frequency domain and captures inter-channel relationships within each frequency band. However, these models rely on input-level frequency information and lack structural integration, limiting their ability to model functional expressiveness. We address this with a Fourier-based KAN that directly learns frequency representations for precise anomaly detection.

# 3 PRELIMINARIES

## 3.1 PROBLEM FORMULATION

Unsupervised MTSAD refers to the task of distinguishing between normal and anomalous segments in a time series, using a model trained exclusively on normal data. A multivariate time series can be formally defined as follows:

$$X = \{x_1, x_2, ..., x_T\}, \quad x_t \in \mathbb{R}^D \tag{1}$$

where $T$ denotes the length of time series, $D$ is the number of channels at each time step, and $x_t$ represents the observation at time $t$. The goal of anomaly detection is to perform binary classification at each time step $t$ to determine whether it is normal or anomalous. To this end, an anomaly score $s_t$, typically derived from reconstruction or prediction errors, is computed for the input time series $X$, and a time step is classified as anomalous if its score exceeds a threshold $\tau$.

$$s_t = \text{AnomalyScore}(x_t, \hat{x}_t), \quad y_t = \begin{cases} 1 & \text{if } s_t > \tau \\ 0 & \text{otherwise} \end{cases} \tag{2}$$

where $\hat{x}_t$ denotes the reconstructed or predicted value at time step $t$ produced by the model, and $y_t \in \{0, 1\}$ indicates the anomaly label (1: anomalous, 0: normal).

## 3.2 KOLMOGOROV–ARNOLD REPRESENTATION THEOREM

The Kolmogorov–Arnold Representation Theorem (Kolmogorov, 1961) guarantees that any multivariate continuous function can be expressed as a composition and sum of a finite number of univariate continuous functions. That is, for a continuous function $f : [0,1]^n \to \mathbb{R}$ with high-dimensional input, the following representation always exists:

$$f(x_1, x_2, \ldots, x_n) = \sum_{q=1}^{2n+1} \Phi_q \left( \sum_{p=1}^{n} \phi_{q,p}(x_p) \right) \quad (3)$$

where $\Phi_q : \mathbb{R} \to \mathbb{R}$ are outer univariate continuous functions, $\phi_{q,p} : [0,1] \to \mathbb{R}$ are inner univariate continuous functions, and each $x_p$ represents a value from the input dimensions. This theorem guarantees that any continuous function can be represented through linear combinations and compositions of univariate functions, suggesting that high-dimensional structures can be decomposed into simpler function-based forms. KAN implements this theorem in the form of a neural network, and unlike traditional MLPs, it achieves high expressiveness by learning and combining separately defined basis functions for each input dimension.

## 3.3 BASIS FUNCTIONS

**Basis Functions for Functional Approximation.** According to the Kolmogorov–Arnold representation theorem, any multivariate continuous function can be expressed as a composition and sum of univariate functions, where the specific forms of these functions depend on the choice of basis. KAN explicitly defines and learns these basis functions, providing greater expressiveness than conventional MLPs. Commonly used basis functions in KAN include:

- B-spline Functions: Piecewise polynomial functions of the form $c \cdot B(x)$, effective for smooth curve approximation and used in the vanilla KAN architecture.

- Chebyshev Polynomials: Defined as $\cos(n \cdot \arccos(x))$, these are high-order polynomials that are orthogonal over the interval $[-1, 1]$, offering strong convergence and approximation accuracy.

- Fourier Basis: Periodic functions in the form of $\cos(nx)$ and $\sin(nx)$, effective for approximating periodic structures and repeating patterns.

These basis functions exhibit distinct convergence rates and approximation behaviors; therefore, selecting an appropriate basis is crucial depending on the characteristics of the data.

**Fourier Basis Functions in Time Series.** Time series data exhibit not only temporal variations but also frequency characteristics such as periodicity, phase, and amplitude. Fourier basis functions decompose these signals into sine and cosine components, facilitating analysis in the frequency domain. For example, a real-valued time series $x(t) \in L^2(\mathbb{R})$ can be represented using the following Fourier series:

$$x(t) = a_0 + \sum_{n=1}^{\infty} a_n \cos(nt) + b_n \sin(nt) \quad (4)$$

This representation offers several advantages:

- High-frequency anomaly detection: Anomalies often manifest as abrupt amplitude changes in the high-frequency domain.

- Pattern change detection: The breakdown of periodic patterns typically results in noticeable deviations in the Fourier spectrum.

- Boundary clarification: In the frequency domain, normal and anomalous segments are often more easily distinguishable than in the time domain.

A theoretical justification for the use of Fourier bases in KANomaly is provided in Appendix D.

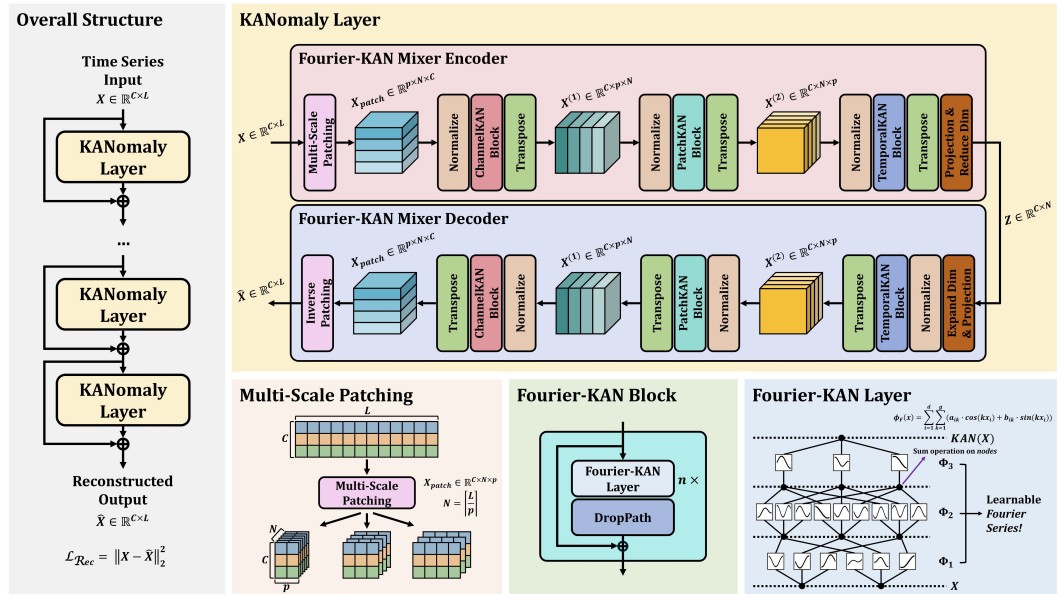

Figure 2: Overview of the proposed KANomaly. The model consists of stacked KANomaly layers, each composed of a Fourier-KAN Mixer Encoder–Decoder structure. Multi-scale patching enables coarse-to-fine representations, while the Fourier-KAN Layer integrates learnable Fourier series for frequency-aware function approximation.

# 4 PROPOSED METHOD: KANOMALY

This section presents the architecture and components of KANomaly. By integrating Fourier basis functions into the KAN, the model captures the frequency structure of time series. To detect diverse anomalies, a multi-scale patching strategy is employed. The Fourier-KAN Mixer aggregates information across channel, patch, and temporal dimensions, modeling complex dependencies in high-dimensional time series. This dimension-wise mixing strategy has also proven effective in recent forecasting (Zhong et al., 2024). Finally, an encoder–decoder reconstructs the input, and anomaly scores are computed from reconstruction errors. The following subsections detail each component. Additionally, an algorithmic description of KANomaly in pseudo-code is provided in Appendix H.

## 4.1 MULTI-SCALE PATCHING STRATEGY

Time series data can exhibit a wide range of anomalies, from abrupt point anomalies to long-term structural deviations. Therefore, it is essential to use a representation that captures both local and global information. To this end, we introduce a Multi-Scale Patching Strategy, which transforms the input time series into patches of various scales.

Given a multivariate time series input $X \in \mathbb{R}^{C \times L}$, where $C$ is the number of channels and $L$ is the sequence length, we define a set of patch lengths $p_1, p_2, \ldots, p_n$. For each $p_i$, the time series is partitioned into non-overlapping segments, resulting in a 3D tensor representation at each scale:

$$X_{\text{patch}}^{(i)} \in \mathbb{R}^{C \times N_i \times p_i}, \quad \text{where} \quad N_i = \left\lceil \frac{L}{p_i} \right\rceil \tag{5}$$

where $N_i$ denotes the number of patches at each scale, and each patch corresponds to a segment of the original time series. This transformation is defined as follows:

$$X_{\text{patch}}^{(i)} = \text{Unfold}(X, p_i), \quad \text{for } i = 1, \ldots, n \tag{6}$$

where $\text{Unfold}(\cdot, p_i)$ divides the time series into patches of length $p_i$. This multi-scale representation enables the model to capture patterns at different temporal resolutions. For each scale, a KAN-based encoder–decoder reconstructs the time series in a coarse-to-fine manner ($p_1 \rightarrow p_2 \rightarrow \cdots \rightarrow p_n$): longer patches (large $p_i$) capture pattern anomalies, while shorter patches (small $p_i$) refine the reconstruction to detect point anomalies.

## 4.2 Fourier-KAN Mixer Architecture

**Fourier-KAN.** Most existing time series anomaly detection models learn patterns in the time domain and detect anomalies based on them. However, certain anomalies may be subtle or ambiguous in the time domain but become more pronounced in the frequency domain. For instance, a sudden increase in high-frequency components or a disruption in periodic structure can be more effectively identified via the Fourier Transform. To address this, we integrate Fourier basis functions into KAN, a neural architecture grounded in the Kolmogorov–Arnold representation theorem. Unlike conventional MLPs with fixed activation functions, KAN employs learnable nonlinear functions per input dimension, providing strong representational power.

Leveraging this architectural design, the Fourier-KAN explicitly models periodicity and spectral anomalies in time series. Specifically, the basis functions of KAN are defined using Fourier series as follows:

$$\phi_F(\mathbf{x}) = \sum_{i=1}^{d} \sum_{k=1}^{g} \left( a_{ik} \cdot \cos(kx_i) + b_{ik} \cdot \sin(kx_i) \right) \tag{7}$$

where $\mathbf{x} \in \mathbb{R}^d$ is the input vector, $a_{ik}$ and $b_{ik}$ are learnable coefficients, and $g$ is the number of frequency components. This formulation lets the model approximate periodic patterns, phase shifts, and amplitude variations by combining sine and cosine functions at multiple frequencies across input dimensions. Using these basis functions, KAN reconstructs the input and detects anomalies via reconstruction error $s_t = |X(t) - \hat{X}(t)|$, reflecting discrepancies in time and frequency domains. This enables detection of frequency-based anomalies, including periodicity collapse, phase distortion, and high-frequency oscillations. The Fourier-based representation is effective not only for identifying point anomalies but also for capturing spectral anomalies across broader temporal ranges. To support this, Fourier-KAN uses the following basis functions to model structural interactions across channels, patches, and time steps:

$$B(x) = \{\cos(kx), \sin(kx)\}_{k=1}^{K} \tag{8}$$

These basis functions span diverse frequency spectra, enabling each KAN layer to learn smooth nonlinearities via function approximation. Each Fourier-KAN Block includes DropPath (Huang et al., 2016) to improve generalization by randomly dropping subpaths. The Fourier-KAN Block is then applied sequentially in the following three stages:

**Channel-wise Mixing.** In the first stage, the model captures channel interactions at the same time position within each patch by applying KAN along the channel axis. This process is defined as follows:

$$X^{(1)} = \text{ChannelKANBlock}(X_{\text{patch}}) \tag{9}$$

This stage ignores temporal and patch-level information, focusing solely on nonlinear inter-channel interactions. KAN's Fourier-based mapping effectively captures these complex dependencies.

**Patch-wise Mixing.** In the second stage, the model captures relationships among patches at different positions while keeping the time step and channel fixed. This facilitates learning repetitive structures and global patterns in the time series and is formulated as follows:

$$X^{(2)} = \text{PatchKANBlock}(X^{(1)}) \tag{10}$$

Through this process, the model captures global information to identify long-term trends in the time series.

**Temporal-wise Mixing.** In the final stage, KAN is applied on the temporal axis at each channel–patch position to capture fine-grained temporal variations and abrupt anomalies:

$$Z = \text{TemporalKANBlock}(X^{(2)}) \tag{11}$$

The final output $Z \in \mathbb{R}^{C \times N}$ encodes nonlinear, frequency-aware features across the channel, patch, and temporal axes of the input. The entire operation of the Fourier-KAN Mixer Encoder is summarized as follows:

$$\text{Enc}_i(X) = \text{TemporalKANBlock}_i \circ \text{PatchKANBlock}_i \circ \text{ChannelKANBlock}_i \left( X_{\text{patch}}^{(i)} \right) \tag{12}$$

This hierarchical mixing structure encodes patterns across multiple axes, supporting the decoder in reconstructing the input and detecting anomalies.

### 4.3 ENCODER & DECODER BASED ON FOURIER-KAN MIXER

The encoder–decoder architecture of KANomaly detects anomalies by reconstructing the input time series. Given a multivariate time series $X \in \mathbb{R}^{C \times L}$, the encoder first performs multi-scale patching and then generates a latent representation $Z$ for each scale using the Fourier-KAN Mixer:

$$Z = \text{Enc}(X) = \text{FourierKANMixer}(\text{Patch}(X)) \tag{13}$$

The decoder reconstructs the time series from the encoder output via a Fourier-KAN Mixer and Unpatch operation:

$$\hat{X} = \text{Dec}(Z) = \text{UnPatch}(\text{FourierKANMixer}(Z)) \tag{14}$$

A key feature of KANomaly is multi-scale coarse-to-fine reconstruction. At each scale, the decoder output feeds into the next, enabling cumulative refinement. A residual connection (He et al., 2016) adds the reconstruction to the input, which is then passed forward. This allows progressive output refinement, capturing global and local variations. The overall architecture is summarized as follows:

$$X_{i+1}^{\text{input}} = X_{i+1}^{\text{raw}} + \hat{X}_i^{\text{patch}}[\,:,\, -L_{i+1}] \tag{15}$$

where $\hat{X}_i^{\text{patch}}$ is the decoder output at scale $i$, and $L_{i+1}$ is the sequence length for the next scale. This approach maintains continuity and consistency across scales, enabling effective detection of diverse anomalies.

### 4.4 MODEL TRAINING AND ANOMALY DETECTION CRITERION

KANomaly is trained unsupervised by learning normal patterns to minimize reconstruction error $\mathcal{L}_{\text{rec}}$ between input and output. Let $\mathbf{X} \in \mathbb{R}^{C \times L}$ be input and $\hat{\mathbf{X}} \in \mathbb{R}^{C \times L}$ the reconstruction. The objective function is defined as:

$$\mathcal{L}_{\text{rec}} = \frac{1}{CL} \sum_{c=1}^{C} \sum_{t=1}^{L} \left( X_{c,t} - \hat{X}_{c,t} \right)^2 \tag{16}$$

In the testing phase, the reconstruction error $e_t$ is computed at each time step $t \in \{1, \dots, L\}$, yielding a sequence of anomaly scores:

$$e_t = \frac{1}{C} \sum_{c=1}^{C} \left( X_{c,t} - \hat{X}_{c,t} \right)^2 \tag{17}$$

The final anomaly score sequence $\text{AnomalyScore}(X) = [e_1, e_2, \dots, e_L] \in \mathbb{R}^L$ is given by:

$$\text{AnomalyScore}(X) = \left[ \left\| X_{i,:} - \hat{X}_{i,:} \right\|_2^2 \right]_{i=1}^{L} \in \mathbb{R}^L \tag{18}$$

A higher $\text{AnomalyScore}(X)$ indicates a greater likelihood of anomaly at the corresponding time step. A time step $t$ is labeled anomalous if its reconstruction error $e_t$ exceeds a threshold $\delta$, determined using a validation set.

## 5 EXPERIMENTS

**Datasets.** We conduct experiments on five widely used real-world datasets: MSL, SMAP (Hundman et al., 2018), PSM (Abdulaal et al., 2021), SMD (Su et al., 2019), and SWaT (Mathur & Tippenhauer, 2016). More detailed dataset descriptions and statistics are provided in Appendix F.1.

**Baselines.** We select 19 representative models, ranging from traditional methods such as Local Outlier Factor (Breunig et al., 2000), One-Class SVM (Manevitz & Yousef, 2001), and Isolation Forest (Liu et al., 2008), to recent SOTA deep learning methods. The latter include DeepSVDD (Ruff et al., 2018), DAGMM (Zong et al., 2018), LSTM-VAE (Park et al., 2018), OmniAnomaly (Su et al., 2019), BeatGAN (Zhou et al., 2019), THOC (Shen et al., 2020), USAD (Audibert et al., 2020), TranAD (Tuli et al., 2022), Anomaly Transformer (AT) (Xu et al., 2021), TimesNet (Wu et al., 2022),

DCdetector (Yang et al., 2023), MEMTO (Song et al., 2023), FITS (Xu et al., 2023), TFMAE (Fang et al., 2024), CATCH (Wu et al., 2025), and KAN-AD[1] (Zhou et al., 2025).

**Implementation Details.** The model is trained using the Adam optimizer (Kingma & Ba, 2017), with an initial learning rate 1e-4. Following standard practices in MTSAD (Xu et al., 2021; Song et al., 2023), the input sequence is segmented using a non-overlapping sliding window. All experiments are conducted using PyTorch (Paszke et al., 2019) with an NVIDIA RTX 3090 24GB GPU. Additional implementation details are provided in Appendix F.2.

**Evaluation Metrics.** Model performance is evaluated using Precision, Recall, and F1-score. Following prior studies, we adopt the widely used point-adjustment strategy (Xu et al., 2021; Tuli et al., 2022; Song et al., 2023; Fang et al., 2024). We also evaluate the model using Affiliation-based (Huet et al., 2022) and Range-based Metrics (Tatbul et al., 2018), and report the results in Appendix N. Additionally, detailed descriptions of the evaluation metrics are provided in Appendix G.

Table 1: Comparison of Precision (P), Recall (R), and F1-score (F1) across five real-world datasets. All values are reported in %. **Bold** and underlined numbers denote the best and second-best performance, respectively.

| Dataset | | MSL | | | PSM | | | SMAP | | |
|---|---|---|---|---|---|---|---|---|---|---|
| Model | Venue | P | R | F1 | P | R | F1 | P | R | F1 |
| LOF | MOD-00 | 91.59 | 19.84 | 32.62 | 100.00 | 28.26 | 44.07 | 81.65 | 26.41 | 39.91 |
| OCSVM | JMLR-01 | 92.63 | 17.46 | 29.38 | 99.99 | 31.90 | 48.36 | 56.53 | 19.45 | 28.94 |
| IForest | ICDM-08 | 33.80 | 31.48 | 32.60 | 89.47 | 69.53 | 78.25 | 38.02 | 51.80 | 43.85 |
| DeepSVDD | ICML-18 | 73.60 | 88.14 | 80.27 | 97.16 | 88.68 | 92.72 | 80.34 | 65.78 | 72.34 |
| DAGMM | ICLR-18 | 89.60 | 63.93 | 74.62 | 93.49 | 70.03 | 80.08 | 86.45 | 56.73 | 68.51 |
| LSTM-VAE | RA-L-18 | 85.49 | 79.94 | 82.62 | 73.62 | 89.92 | 80.96 | 92.20 | 67.75 | 78.10 |
| OmniAnomaly | KDD-19 | 89.02 | 86.37 | 87.67 | 88.39 | 74.46 | 80.83 | 92.49 | 81.99 | 86.92 |
| BeatGAN | IJCAI-19 | 89.75 | 85.42 | 87.53 | 90.30 | 93.84 | 92.04 | 92.38 | 55.85 | 69.61 |
| THOC | NeurIPS-20 | 88.45 | 90.97 | 89.69 | 88.14 | 90.99 | 89.54 | 92.06 | 89.34 | 90.68 |
| USAD | KDD-20 | 27.36 | 75.88 | 40.22 | 87.35 | 75.56 | 81.03 | 65.23 | 53.06 | 58.52 |
| TranAD | VLDB-22 | 79.10 | 89.03 | 83.77 | 93.64 | 88.91 | 91.21 | 94.16 | 55.41 | 69.77 |
| AT | ICLR-22 | 91.67 | 93.43 | 92.55 | 97.29 | 98.13 | 97.71 | 93.77 | 99.13 | 96.38 |
| TimesNet | ICLR-23 | 89.55 | 75.29 | 81.80 | 98.52 | 96.31 | 97.40 | 89.44 | 55.44 | 68.45 |
| DCdetector | KDD-23 | 95.33 | 91.44 | 93.34 | 97.24 | 97.72 | 97.48 | 94.28 | 98.64 | 96.41 |
| MEMTO | NeurIPS-23 | 91.87 | 95.87 | 93.82 | 97.44 | 98.51 | 97.97 | 93.62 | 99.26 | 96.36 |
| FITS | ICLR-24 | 77.94 | 75.42 | 76.66 | 98.98 | 88.34 | 93.36 | 95.41 | 53.06 | 68.19 |
| TFMAE | ICDE-24 | 92.45 | 92.67 | 92.56 | 97.62 | 98.11 | 97.86 | 94.72 | 98.32 | 96.49 |
| CATCH | ICLR-25 | 58.82 | 75.62 | 66.17 | 87.52 | 96.64 | 91.85 | 95.00 | 55.41 | 70.00 |
| KAN-AD | ICML-25 | 90.62 | 80.01 | 84.99 | 99.36 | 92.71 | 95.92 | 93.45 | 81.13 | 86.86 |
| KANomaly | - | 93.99 | 97.35 | **95.64** | 98.58 | 98.76 | **98.67** | 97.12 | 97.81 | **97.46** |
| Dataset | | SMD | | | SWaT | | | Average | | |
| Model | Venue | P | R | F1 | P | R | F1 | P | R | F1 |
| LOF | MOD-00 | 99.44 | 4.99 | 9.51 | 100.00 | 12.28 | 21.88 | 94.54 | 18.36 | 29.60 |
| OCSVM | JMLR-01 | 93.30 | 5.89 | 11.08 | 100.00 | 12.36 | 22.00 | 88.49 | 17.41 | 27.95 |
| IForest | ICDM-08 | 52.51 | 13.20 | 21.10 | 93.09 | 23.74 | 37.83 | 61.38 | 37.95 | 42.73 |
| DeepSVDD | ICML-18 | 66.91 | 71.79 | 69.26 | 34.95 | 76.74 | 48.03 | 70.59 | 78.23 | 72.52 |
| DAGMM | ICLR-18 | 67.30 | 49.89 | 57.30 | 89.92 | 57.84 | 70.40 | 85.35 | 59.68 | 70.18 |
| LSTM-VAE | RA-L-18 | 75.76 | 90.08 | 82.30 | 76.00 | 89.50 | 82.20 | 80.61 | 83.44 | 81.24 |
| OmniAnomaly | KDD-19 | 83.68 | 86.82 | 85.22 | 81.42 | 84.30 | 82.83 | 87.00 | 82.79 | 84.69 |
| BeatGAN | IJCAI-19 | 72.90 | 84.09 | 78.10 | 64.01 | 87.46 | 73.92 | 81.87 | 81.33 | 80.24 |
| THOC | NeurIPS-20 | 79.76 | 90.95 | 84.99 | 83.94 | 86.36 | 85.13 | 86.47 | 89.72 | 88.01 |
| USAD | KDD-20 | 22.92 | 57.08 | 32.71 | 65.85 | 99.91 | 79.38 | 53.74 | 72.30 | 58.37 |
| TranAD | VLDB-22 | 73.86 | 83.78 | 78.51 | 91.70 | 79.05 | 84.91 | 86.49 | 79.24 | 81.63 |
| AT | ICLR-22 | 88.81 | 92.39 | 90.56 | 89.87 | 78.29 | 83.68 | 92.28 | 92.27 | 92.18 |
| TimesNet | ICLR-23 | 87.88 | 81.54 | 84.59 | 92.14 | 93.10 | 92.62 | 91.51 | 80.34 | 84.97 |
| DCdetector | KDD-23 | 85.35 | 81.22 | 83.23 | 93.12 | 99.96 | 96.42 | 93.06 | 93.80 | 93.38 |
| MEMTO | NeurIPS-23 | 87.58 | 95.89 | 91.55 | 97.30 | 93.13 | 95.17 | 93.56 | 96.53 | 94.97 |
| FITS | ICLR-24 | 77.03 | 83.90 | 80.32 | 94.53 | 83.96 | 88.93 | 88.78 | 76.94 | 81.49 |
| TFMAE | ICDE-24 | 91.41 | 91.07 | 91.24 | 97.13 | 97.91 | 97.52 | 94.67 | 95.62 | 95.13 |
| CATCH | ICLR-25 | 77.85 | 81.54 | 79.65 | 91.71 | 85.48 | 88.48 | 82.18 | 78.94 | 79.23 |
| KAN-AD | ICML-25 | 98.71 | 87.05 | 83.95 | 92.88 | 93.46 | 93.17 | 95.00 | 86.87 | 88.98 |
| KANomaly | - | 88.56 | 95.16 | **91.74** | 97.74 | 99.46 | **98.59** | 95.20 | 97.71 | **96.42** |

---

[1]KAN-AD is fundamentally designed for UTSAD. However, to adapt it for MTSAD, each multivariate input was reshaped from `(batch_size, window_length, n_features)` to `(batch_size * n_features, window_length)`, following the auxiliary configuration described in the original paper.

## 5.1 Main Results

The proposed KANomaly is evaluated against 19 competing models across five real-world MT-SAD datasets and consistently achieves the best performance. As shown in Table 1, it outperforms Transformer-based models such as TranAD, Anomaly Transformer, MEMTO, TFMAE, and CATCH; time-domain-only models such as Anomaly Transformer, DCdetector, and MEMTO; and frequency-aware models including TimesNet, FITS, TFMAE, and CATCH. These results highlight the effectiveness of KANomaly's integration of frequency-domain representations with the Kolmogorov–Arnold function approximation framework, surpassing models that do not explicitly encode frequency-based functions within their architecture. Furthermore, KANomaly also outperforms KAN-AD, which employs a channel-independent strategy in its multivariate configuration. These results confirm that the observed improvement is not merely attributable to inheriting the KAN backbone, but rather stems from architectural innovations, such as the multi-scale patching strategy and the Fourier-KAN Mixer, that enable effective modeling of cross-channel dependencies.

## 5.2 Model Analysis

**Ablation Study.** To analyze the contribution of each component in KANomaly, we perform a comprehensive ablation study by removing or replacing components with alternatives and evaluating performance on benchmark datasets. As shown in Table 2, replacing the Fourier-KAN with MLP (Linear→GELU→Linear), Chebyshev-KAN, or Vanilla-KAN leads to notable performance drops, confirming the value of explicit frequency-based representations. Removing the multi-scale patching and using a fixed patch size (square root of the input length) reduces the model's ability to detect anomalies across different temporal scales. Eliminating patching entirely causes even greater degradation, highlighting the critical role of patch-level feature extraction. Reversing the patching order from coarse-to-fine to fine-to-coarse also impairs performance, confirming the effectiveness of the original hierarchical design. We further examine the impact of the channel, patch, and temporal mixing modules individually. Removing any one of them results in performance degradation, highlighting the importance of capturing interactions across all three dimensions.

Furthermore, to isolate the effect of frequency features independent from the KAN structure, we replace the Fourier-KAN in KANomaly with a pure MLP that directly operates on Fourier-transformed inputs (Fourier features + MLP). In this variant, each input sequence is first transformed into the frequency domain via the Fourier Transform, passed through the MLP (Linear → GELU → Linear), and then reconstructed back to the time domain using the inverse Fourier Transform. This configuration yields the lowest performance among all variants, demonstrating that the performance gains of KANomaly do not originate from frequency feature learning alone but from the functional-level integration of learnable Fourier basis functions within the KAN framework. These results validate the overall design of KANomaly.

Table 2: Ablation results of core components in KANomaly. **Bold** numbers indicate the best F1-scores, and the values in parentheses denote the F1-score decrease compared to the full model. All values are reported in %.

| Variations | MSL | PSM | SMAP | SMD | SWaT | Average F1 |
|---|---|---|---|---|---|---|
| KANomaly (ours) | **95.64** | **98.67** | **97.46** | **91.74** | **98.59** | **96.42** |
| (1) w/o Fourier-KAN (→ MLP) | 85.48 | 97.66 | 68.43 | 87.78 | 96.68 | 87.21 (-9.21%) |
| (2) w/o Fourier-KAN (→ Cheby-KAN) | 83.97 | 97.22 | 69.24 | 85.58 | 97.22 | 86.65 (-9.77%) |
| (3) w/o Fourier-KAN (→ Vanilla-KAN) | 82.40 | 97.69 | 68.39 | 88.27 | 97.40 | 86.83 (-9.59%) |
| (4) w/o Multi-Scale Patching (Fixed Patch Size) | 86.60 | 96.94 | 96.09 | 83.55 | 97.66 | 92.17 (-4.25%) |
| (5) w/o Patching | 88.98 | 98.14 | 90.14 | 80.29 | 95.36 | 90.58 (-5.84%) |
| (6) Fine-to-Coarse Patch (Inverse Patch Scale) | 88.54 | 95.83 | 94.30 | 82.22 | 97.41 | 91.66 (-4.76%) |
| (7) w/o Channel Mixing | 85.77 | 98.16 | 69.30 | 86.58 | 96.87 | 87.34 (-9.08%) |
| (8) w/o Patch Mixing | 86.85 | 95.38 | 76.01 | 89.54 | 95.77 | 88.71 (-7.71%) |
| (9) w/o Temporal Mixing | 87.27 | 97.26 | 74.90 | 90.86 | 96.42 | 89.34 (-7.08%) |
| (10) Fourier features + MLP | 80.72 | 89.40 | 68.24 | 84.84 | 82.37 | 81.11 (-15.31%) |

**Visual Analysis.** To verify the ability of KANomaly to detect various types of anomalies, we conduct a visual analysis using the synthetic TODS dataset (Lai et al., 2021), which includes five distinct anomaly types: Point Global, Point Contextual, Pattern Shapelet, Pattern Seasonal, and Pattern Trend. As shown in Figure 3 (Left), KANomaly successfully identifies both point and pattern anomalies across different temporal patterns. The precise localization of each anomaly type

by the model demonstrates the robustness of KANomaly in capturing diverse abnormal behaviors. These results confirm the effectiveness of KANomaly in handling a wide spectrum of anomaly characteristics in time series. Additional Visual Analysis comparing KANomaly with representative frequency-aware SOTA baselines, such as TFMAE and FITS, is provided in Appendix O.

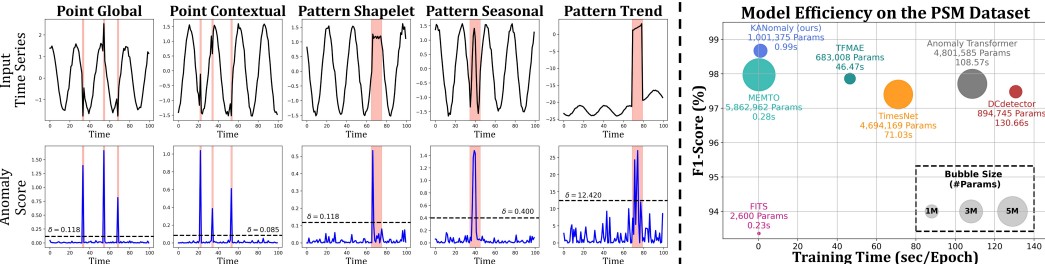

Figure 3: Visualization and efficiency analysis of KANomaly. (Left) Detection results across different anomaly types. The top row shows input time series, and the bottom row shows corresponding anomaly scores; red regions indicate ground-truth anomalies. (Right) Model efficiency comparison on the PSM dataset.

**Efficiency Analysis.** Figure 3 (Right) compares KANomaly's efficiency on the PSM dataset against six baseline models in terms of F1-score, training time per epoch, and model size. KANomaly achieves the highest F1-score of 98.67% with only 0.99 seconds per epoch and approximately 1M parameters. This demonstrates a favorable trade-off between accuracy, speed, and compactness for practical MTSAD. Additional Efficiency Analysis across all benchmark datasets (MSL, SMAP, PSM, SMD, and SWaT) is provided in Appendix P. Detailed results for the Efficiency Analysis Compared to Baseline Models and Ablation Models can be found in Appendix P.1 and Appendix P.2, respectively.

## 6 CONCLUSION AND FUTURE WORK

We propose KANomaly, an unsupervised MTSAD model that combines Fourier-based KAN with multi-scale patching. By explicitly learning frequency representations, the model effectively captures periodic and spectral anomalies often missed in the time domain. A Fourier-KAN Mixer captures local-global-channel dependencies via nonlinear mixing across temporal, patch, and channel dimensions. The multi-scale patching strategy enables detection of both point and pattern anomalies. Experiments indicate that our method consistently outperforms SOTA models on multiple benchmarks, demonstrating both high effectiveness and efficiency. In future work, we plan to leverage the inherent interpretability of KAN architectures to analyze learned Fourier basis functions and provide deeper insight into how frequency-aware representations contribute to anomaly detection. Moreover, extending this direction toward symbolic functional decomposition may enable interpretable diagnosis of anomaly sources, enhancing the model's applicability to domains where transparency is critical.

## ETHICS STATEMENT

We focus exclusively on the Multivariate Time Series Anomaly Detection problem, which is a scientific and technical task. It does not involve human subjects or sensitive personal data. Therefore, this work does not pose any potential ethical risks.

## REPRODUCIBILITY STATEMENT

We provide comprehensive implementation details in the main text and Appendix, including dataset descriptions, baseline methods, evaluation metrics, hyperparameter settings, and the computing infrastructure (hardware and software) used for all experiments. The source code is included in the Supplementary Material for review purposes. Upon acceptance, the complete source code and documentation will be released publicly under the MIT License via a GitHub repository.

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

## A    MIXER NETWORK IN TIME SERIES ANALYSIS

Recent advances in MLP-Mixer-style architectures have led to notable progress in time series modeling. TSMixer (Chen et al., 2023) adopts an all-MLP architecture that independently mixes temporal and feature dimensions, enabling efficient and scalable forecasting. PatchMLP (Tang & Zhang, 2025) builds on this idea by segmenting time series into temporal patches and applying inter- and intra-variable MLP mixing, thereby enhancing the modeling of localized temporal dynamics. WP-Mixer (Murad et al., 2025) incorporates multi-level wavelet decomposition with patch and embedding mixing to effectively capture multi-resolution patterns across temporal and frequency domains. MSD-Mixer (Zhong et al., 2024) further advances this line of work by explicitly decomposing time series across layers using multi-scale temporal patching with MLPs for modeling variations and correlations, and applies a residual loss to enforce decomposition completeness. While prior Mixer-based methods mainly focus on forecasting tasks, no prior work has explored KAN-based mixing. KANomaly is the first to integrate Kolmogorov–Arnold Networks (KANs) with a Mixer-style architecture, introducing a novel model for multivariate time series anomaly detection that combines expressive nonlinear approximation with robust frequency-aware representations.

## B    DIFFERENTIATION FROM SIMILAR MIXER ARCHITECTURES

Several recent time series models, such as PatchMLP (Tang & Zhang, 2025) and MSD-Mixer (Zhong et al., 2024), adopt patching and MLP-Mixer-style architectures to enhance representation learning. However, their design philosophies and objectives differ fundamentally from ours. PatchMLP is a forecasting model that applies inter- and intra-variable MLP mixing over temporal patches to improve long-term prediction. It additionally employs average pooling to decompose the input into smooth components and residuals, enabling component-specific processing. MSD-Mixer introduces a multi-scale decomposition pipeline, explicitly separating time series into trend, seasonal, and residual components. Its final output is produced by a task-specific prediction head, rather than a decoder. In contrast, KANomaly is an unsupervised anomaly detection model that reconstructs the full input sequence without decomposition. Leveraging Fourier-based Kolmogorov–Arnold Networks (KANs), our model captures high-frequency irregularities such as periodic shifts and spectral disturbances, which are often critical for anomaly detection but overlooked by forecasting models. While structurally similar in patch-wise mixing, KANomaly fundamentally diverges in its holistic, frequency-aware reconstruction approach tailored to unsupervised anomaly detection.

## C    DISTINCTION FROM PRIOR FREQUENCY-AWARE METHODS

A key distinction between KANomaly's explicit functional integration and prior frequency-aware approaches lies in how frequency-domain information is utilized and where it is integrated within the network. Several previous models have incorporated frequency-domain information; however, they typically apply it at the feature level rather than at the functional integration level adopted by KANomaly.

Conventional frequency-aware approaches such as Autoformer (Wu et al., 2021), FEDformer (Zhou et al., 2022b), TimesNet (Wu et al., 2022), TFMAE (Fang et al., 2024), FITS (Xu et al., 2023), and FreTS (Yi et al., 2023) apply a Fourier transform in a global manner over the entire time window. The resulting frequency coefficients are treated as static features and subsequently provided as inputs to downstream backbones such as MLPs or Transformers. For example, Autoformer, FEDformer, and TimesNet extract only the top-K frequency components, TFMAE uses magnitude spectra alone, discarding phase information, FITS removes high-frequency bands through cut-off filtering, and FreTS transforms the input sequence to the frequency domain before passing the real and imaginary components to a frequency-domain MLP. In all these cases, frequency-domain representations are employed at the feature level as pre-computed descriptors without modifying the functional form of the network. Consequently, these models remain bound to a fixed frequency resolution and cannot dynamically adapt to non-stationary or locally varying spectral characteristics.

In contrast, KANomaly performs explicit functional integration of frequency information within the Kolmogorov–Arnold Network (KAN) framework. Rather than treating Fourier features as static inputs, learnable Fourier basis functions are embedded directly into the network's univariate mappings,

redefining the activation space itself. Each edge function in KANomaly replaces the conventional spline basis with a learnable Fourier series:

$$\phi_F(\mathbf{x}) = \sum_{i=1}^{d} \sum_{k=1}^{g} (a_{ik} \cdot \cos(kx_i) + b_{ik} \cdot \sin(kx_i)) \tag{19}$$

where $\mathbf{x} \in \mathbb{R}^d$ denotes the input vector, $a_{ik}$ and $b_{ik}$ are learnable coefficients, and $g$ controls the number of Fourier components. This construction enables each functional mapping to adaptively determine which frequency bases, such as $\sin(kx)$ or $\cos(kx)$, best reconstruct local temporal variations.

Thus, KANomaly fundamentally differs from methods that employ Fourier series only at the input level before feeding signals into MLP or attention backbones. Instead, the proposed model integrates learnable Fourier basis functions directly within the functional mappings, redefining the activation space to be Fourier-parametric.

This architectural property embodies the essence of explicit functional integration. In KANomaly, time-domain and frequency-domain representations are co-learned within a unified, parameterized function space. Such integration endows the model with a qualitatively different capability, allowing it to flexibly approximate non-stationary patterns characterized by periodicity, phase distortion, and fine-grained spectral perturbations whose spectral signatures evolve across patches, channels, and temporal segments. These characteristics are typically difficult to capture for models that rely solely on feature-level frequency representations.

## D    THEORETICAL JUSTIFICATION FOR FOURIER BASES IN KANOMALY

This section provides the theoretical justification for employing Fourier bases within KANomaly, highlighting why they are more suitable than alternative basis functions such as B-splines or Chebyshev polynomials for modeling normal patterns and isolating anomalous patterns. When approximating a time-series signal $x(t)$ using $N$ basis functions, the Fourier expansion models it as

$$\hat{x}_N(t) = \sum_{k=1}^{N} c_k \phi_k(t) \tag{20}$$

where $\phi_k(t) = \{\cos(\omega_k t), \sin(\omega_k t)\}$. The corresponding Fourier coefficient is

$$c_k = \frac{1}{T} \int_0^T x(t) \, \phi_k^*(t) \, dt \tag{21}$$

Fourier bases are optimal in the $L^2$ function space (space of square-integrable functions) because they minimize the mean square approximation error

$$E_{MSE} = \int |x(t) - \hat{x}_N(t)|^2 dt \tag{22}$$

which ensures maximum energy compaction such that the majority of the energy of a normal signal is concentrated in a small subset of low-frequency coefficients. Parseval's identity,

$$\int |x(t)|^2 dt = \sum_{k=1}^{N} |c_k|^2 \tag{23}$$

further confirms that the time-domain energy is fully preserved in the frequency-domain coefficient energy. Under normal conditions, the signal $x_{\text{normal}}(t)$ can be reconstructed using very few Fourier

coefficients, whereas anomaly-induced deviations $x_{\text{anomaly}}(t)$ require additional or unusually distributed coefficients. Thus, the presence and intensity of anomalies can be directly inferred from deviations in spectral compactness.

Fourier basis functions also exhibit strict orthogonality, which is defined as

$$\int \phi_k(t)\phi_j^*(t)dt = 0 \quad \text{for } k \neq j \tag{24}$$

This ensures that each coefficient represents an independent component of the signal's frequency structure, meaning anomaly-related distortions can be separated from normal patterns without interference. Such orthogonality allows anomaly scoring to be computed without cross-term interactions that could obscure detection.

In contrast, B-spline functions approximate a signal using

$$\hat{x}(t) = \sum_{k=1}^{N} c_k B_k(t) \tag{25}$$

but since B-spline functions are non-orthogonal, the coefficients $c_k$ are statistically dependent. A perturbation in one region of the signal affects multiple coefficients, causing anomaly-related changes to propagate throughout the approximation and complicating separation. Chebyshev polynomials, defined as

$$T_k(t) = \cos(k \arccos(t)), \quad t \in [-1, 1] \tag{26}$$

are orthogonal under a weighted inner product but optimized to minimize the maximum approximation error (minimax criterion) rather than MSE. Their equiripple behavior may cause localized anomalies to produce widespread oscillations across coefficients. This makes anomaly localization and separation more difficult. For this reason, Chebyshev-based expansions are commonly adopted in time-series forecasting applications (e.g., TimeKAN) rather than anomaly detection.

Therefore, the use of Fourier basis functions in KANomaly is not based on assumptions about the frequency location of anomalies but on mathematical guarantees of optimal energy compaction and coefficient independence in the $L^2$ space. These properties allow normal patterns to be efficiently represented using a minimal number of coefficients while enabling anomaly-induced deviations to be identified as distinct residual spectral energy. When integrated into the Fourier-KAN Mixer, these spectral characteristics are further enhanced through the model's channel-patch-temporal mixing architecture, supporting precise localization of anomaly effects across multiple scales and variable interactions.

In summary, Fourier bases provide the most appropriate functional representation for anomaly detection due to minimum mean square approximation error and strict orthogonality, which jointly support efficient normal pattern modeling and statistically independent separation of anomaly components. Alternative bases such as B-spline functions and Chebyshev polynomials lack these combined properties, resulting in reduced anomaly separability. Consequently, the choice of Fourier basis functions is mathematically justified and directly contributes to the anomaly detection capabilities demonstrated in the proposed framework.

## E    CLARIFICATION OF PRIOR KAN-BASED METHODS FOR MTSAD

Although prior attempts such as KAN-AD (Zhou et al., 2025) and MTAD-Kanformer (Xie et al., 2025) utilized KAN-based architectures, they do not address Multivariate Time Series Anomaly Detection (MTSAD) in the manner proposed in this work. In contrast, KANomaly is the first Fourier-KAN-based method to explicitly integrate learnable Fourier basis functions with multi-scale patching and cross-channel mixing for MTSAD.

KAN-AD is fundamentally designed as a Univariate Time Series Anomaly Detection (UTSAD) framework, as explicitly stated in Section 2.1 (Problem Statement) of the KAN-AD paper. In its

multivariate experiments, the input tensor is reshaped from `(batch_size, window_length, n_features)` to `(batch_size * n_features, window_length)`, treating each channel as an independent univariate time series, as also stated in Section 4.7 (Performance on Multivariate Time Series) of the KAN-AD paper. This procedure intentionally ignores inter-channel dependencies, meaning that KAN-AD does not function as a true multivariate model. In contrast, the core contribution of KANomaly lies not in simply applying KAN, but in designing the Fourier-KAN Mixer to explicitly capture nonlinear inter-channel interactions, which are essential to multivariate anomaly detection. This architectural design enables modeling of inherent correlation structures among variables, a capability that is unattainable in channel-independent formulations such as KAN-AD. Paradoxically, KAN-AD reinforces the necessity and novelty of the proposed KANomaly architecture by demonstrating the limitations of reshaping-based univariate extensions when applied to Multivariate Time Series scenarios.

Similarly, prior work such as MTAD-Kanformer is primarily limited to modifying only the output projection of a variant of the Anomaly Transformer by replacing the final linear layer with a Vanilla-KAN, without altering the transformer backbone. These approaches neither reparameterize internal functional spaces using Fourier-KAN nor implement hierarchical mixing across channel-, patch-, and temporal dimensions. Consequently, they remain fundamentally distinct from the method proposed in this work.

Table 3: Statistics of the Datasets. (AR: Anomaly Ratio)

| Datasets | Domain | #Train | #Valid | #Test | Dim | AR (%) |
|---|---|---|---|---|---|---|
| MSL | Spacecraft | 46,653 | 11,664 | 73,729 | 55 | 10.5 |
| PSM | Server Machine | 105,984 | 26,497 | 87,841 | 25 | 27.8 |
| SMAP | Spacecraft | 108,146 | 27,037 | 427,617 | 25 | 12.8 |
| SMD | Server Machine | 566,724 | 141,681 | 708,420 | 38 | 4.2 |
| SWaT | Water treatment | 396,000 | 99,000 | 449,919 | 51 | 12.1 |

# F  DETAILS OF THE EXPERIMENT SETTINGS

## F.1  DATASET DESCRIPTIONS

This subsection provides details of the datasets used in this work. We evaluate on five widely adopted real-world benchmarks and one synthetic dataset, all of which are publicly available. Table 3 summarizes the dataset statistics.

- **MSL** (Mars Science Laboratory rover) and **SMAP** (Soil Moisture Active Passive satellite) (Hundman et al., 2018) are multivariate time series datasets derived from real-world telemetry data collected during NASA space exploration missions. MSL contains 55-channel data from the Curiosity rover's surface operations on Mars, while SMAP includes 25-channel telemetry logs from the SMAP satellite. Anomaly labels in both datasets are based on Incident Surprise Anomaly (ISA) reports documenting unexpected mission events. The MSL and SMAP datasets are publicly available at `https://github.com/khundman/telemanom`.

- **PSM** (Pooled Server Metrics) (Abdulaal et al., 2021) is a multivariate time series dataset derived from internal monitoring of multiple application server nodes at eBay. It comprises 25 features (except timestamp) representing key system-level metrics, including CPU utilization and memory usage. The dataset is partitioned into a 13-week training set and an 8-week test set, enabling robust evaluation of anomaly detection models under real-world server workloads. The PSM dataset is publicly available at `https://github.com/eBay/RANSynCoders/tree/main/data`.

- **SMD** (Server Machine Dataset) (Su et al., 2019) is a public dataset constructed from a large-scale internet service company's server data, designed for evaluating multivariate time series anomaly detection models. It consists of five weeks of data collected from 28 different machines, each equipped with 38 sensors that record various system-level metrics. The dataset captures both normal and anomalous operating conditions, where anomalies re-

flect diverse failure modes encountered during server operations. The SMD dataset is publicly available at `https://github.com/NetManAIOps/OmniAnomaly/tree/master/ServerMachineDataset`.

- **SWaT** (Secure Water Treatment) (Mathur & Tippenhauer, 2016) dataset was collected from a scaled-down yet fully operational water treatment testbed designed for cybersecurity research in Industrial Control Systems (ICS). It comprises 51 features recorded at 1-second intervals over 11 days, including 7 days of normal operation and 4 days containing 36 injected attack scenarios that emulate various types of cyber-physical anomalies. The SWaT dataset is publicly available at `https://itrust.sutd.edu.sg/itrust-labs_datasets/`.

- **TODS** (Lai et al., 2021) is a benchmark framework tailored for evaluating unsupervised time series anomaly detection methods under a variety of controlled synthetic scenarios. It provides a suite of synthetic datasets that comprehensively reflect diverse types of anomalies that may occur in time series data, including Point-Global, Point-Contextual, Pattern-Shapelet, Pattern-Seasonal, and Pattern-Trend anomalies. The synthetic data generation code is publicly available at `https://github.com/datamllab/tods/tree/benchmark/benchmark/synthetic/Generator`. For a fair comparison, we follow the official implementation presented in Listing 1, modifying only the length parameter to generate longer time series.

Listing 1: TODS synthetic dataset generation code in Python

```python
# This code is derived from the publicly released implementation of the TODS project.
# Source: https://github.com/datamllab/tods

BEHAVIOR_CONFIG = {'freq': 0.04, 'coef': 1.5, "offset": 0.0, 'noise_amp': 0.05}
BASE = [1.4529900e-01, 1.2820500e-01, 9.4017000e-02, 7.6923000e-02, 1.1111100e-01,
        1.4529900e-01, 1.7948700e-01, 2.1367500e-01, 2.1367500e-01]

train_data = DataGenerator(stream_length=20000,
                           behavior=sine,
                           behavior_config=BEHAVIOR_CONFIG)

point_global = DataGenerator(stream_length=5000,
                             behavior=sine,
                             behavior_config=BEHAVIOR_CONFIG)

point_contextual = DataGenerator(stream_length=5000,
                                 behavior=sine,
                                 behavior_config=BEHAVIOR_CONFIG)

collective_global = DataGenerator(stream_length=5000,
                                  behavior=sine,
                                  behavior_config=BEHAVIOR_CONFIG)

collective_seasonal = DataGenerator(stream_length=5000,
                                    behavior=sine,
                                    behavior_config=BEHAVIOR_CONFIG)

collective_trend = DataGenerator(stream_length=5000,
                                 behavior=sine,
                                 behavior_config=BEHAVIOR_CONFIG)

point_global.point_global_outliers(ratio=0.05, factor=3.5, radius=5)
point_contextual.point_contextual_outliers(ratio=0.05, factor=2.5, radius=5)
collective_global.collective_global_outliers(ratio=0.05, radius=5, option='square',
                                             coef=1.5, noise_amp=0.03, level=20,
                                             freq=0.04, base=BASE, offset=0.0)
collective_seasonal.collective_seasonal_outliers(ratio=0.05, factor=3, radius=5)
collective_trend.collective_trend_outliers(ratio=0.05, factor=0.5, radius=5)
```

## F.2 Additional Implementation Details

In this subsection, we present additional implementation details. We train our model using the Adam optimizer (Kingma & Ba, 2017) with an initial learning rate of 1e-4. To prevent overfitting, an early stopping strategy is applied: training is terminated if the validation loss does not improve within 3 epochs. Following standard practices in multivariate time series anomaly detection (Xu et al., 2021; Song et al., 2023), the input sequence is segmented using a non-overlapping sliding window. The normal portion of the dataset is split into 80% for training and 20% for validation, while anomalies are present exclusively in the test set.

The anomaly detection threshold $\delta$ is determined based on a predefined anomaly ratio $r\%$, such that the same proportion of data points in the validation set is classified as anomalies. All experiments reported in this paper are conducted on a standardized computational setup comprising an AMD Ryzen 7 3700X CPU @ 3.60GHz, 64GB RAM, and an NVIDIA RTX 3090 24GB GPU. The experiments are run within a Docker container configured with CUDA 11.8, cuDNN 8.7.0, and Ubuntu 20.04. Our implementation is based on PyTorch 2.1.2 (Paszke et al., 2019).

# G Details of Evaluation Metrics

In this section, we provide the detailed definitions of the evaluation metrics employed in our study. We categorize them into Classical metrics, Range-based metrics that explicitly account for event duration, and Affiliation-based metrics that further enhance robustness.

## G.1 Classical Metrics

Classical metrics are among the most widely used evaluation criteria. They evaluate detection performance based on the confusion matrix, which consists of True Positives (TP), True Negatives (TN), False Positives (FP), and False Negatives (FN). Precision, Recall, and F1-score are defined as follows:

**Precision.** Defined as the proportion of correctly detected anomalies over all predicted anomalies, it reflects the reliability of the detection results:

$$\text{Precision} = \frac{TP}{TP + FP} \tag{27}$$

where TP (True Positives) denotes the number of correctly detected anomalies, and FP (False Positives) denotes the number of normal points that are incorrectly flagged as anomalies.

**Recall.** Defined as the proportion of successfully detected anomalies over all actual anomalies, it reflects the completeness of the detection results:

$$\text{Recall} = \frac{TP}{TP + FN} \tag{28}$$

where FN (False Negatives) denotes the number of anomalies that are missed by the model.

**F1-score.** Defined as the harmonic mean of Precision and Recall, it provides a balanced evaluation:

$$\text{F1-score} = 2 \times \frac{\text{Precision} \times \text{Recall}}{\text{Precision} + \text{Recall}} \tag{29}$$

## G.2 Range-based Metrics

Many anomalies in real-world time series occur over continuous intervals rather than isolated points. Classical metrics are not well-suited for such range-based anomalies, as they fail to capture partial overlaps or event-level correctness. To address this, Range-based Precision, Range-based Recall, and Range-based F1-score were introduced (Tatbul et al., 2018).

Let $R = \{R_1, \ldots, R_{N_r}\}$ denote the set of real anomaly ranges and $P = \{P_1, \ldots, P_{N_p}\}$ denote the set of predicted anomaly ranges. The total numbers of real and predicted anomaly ranges are $N_r$ and $N_p$, respectively.

**Range-based Recall.** It is defined as the average recall over all real anomaly ranges:

$$\text{Recall}_T(R, P) = \frac{1}{N_r} \sum_{i=1}^{N_r} \text{Recall}_T(R_i, P). \tag{30}$$

For a single real anomaly range $R_i$, the recall score combines an existence reward and an overlap reward:

$$\text{Recall}_T(R_i, P) = \alpha \cdot \text{ExistenceReward}(R_i, P) + (1 - \alpha) \cdot \text{OverlapReward}(R_i, P), \tag{31}$$

where $\alpha \in [0, 1]$ is a tunable parameter. The existence reward is given by:

$$\text{ExistenceReward}(R_i, P) = \begin{cases} 1, & \text{if } \sum_{j=1}^{N_p} |R_i \cap P_j| \geq 1, \\ 0, & \text{otherwise.} \end{cases} \tag{32}$$

The overlap reward further considers cardinality, size, and positional bias:

$$\text{OverlapReward}(R_i, P) = \text{CardinalityFactor}(R_i, P) \cdot \sum_{j=1}^{N_p} \omega(R_i, R_i \cap P_j, \delta), \tag{33}$$

where $\omega(\cdot)$ is the size function weighted by positional bias $\delta(\cdot)$, and CardinalityFactor penalizes fragmented predictions:

$$\text{CardinalityFactor}(R_i, P) = \begin{cases} 1, & \text{if } R_i \text{ overlaps with at most one } P_j \in P, \\ \gamma(R_i, P), & \text{otherwise,} \end{cases} \tag{34}$$

with $0 \leq \gamma(\cdot) \leq 1$ defined by the application.

**Range-based Precision.** In analogy to Range-based Recall, Range-based Precision is defined as:

$$\text{Precision}_T(R, P) = \frac{1}{N_p} \sum_{i=1}^{N_p} \text{Precision}_T(R, P_i). \tag{35}$$

For each predicted anomaly range $P_i$, the precision score is determined by:

$$\text{Precision}_T(R, P_i) = \text{CardinalityFactor}(P_i, R) \cdot \sum_{j=1}^{N_r} \omega(P_i, P_i \cap R_j, \delta). \tag{36}$$

**Range-based F1-score.** It extends the classical F1-score formulation by combining Range-based Precision and Range-based Recall through their harmonic mean:

$$\text{F1}_T(R, P) = 2 \times \frac{\text{Precision}_T(R, P) \times \text{Recall}_T(R, P)}{\text{Precision}_T(R, P) + \text{Recall}_T(R, P)}. \tag{37}$$

The experimental results evaluated with Range-based Metrics are reported in Appendix N.

### G.3 AFFILIATION-BASED METRICS

Affiliation-based metrics were proposed to overcome the limitations of both classical and range-based metrics, which either ignore temporal adjacency or fail to account for event durations. These metrics provide a theoretically grounded, parameter-free, and interpretable extension of Precision and Recall by leveraging the concept of *affiliation* between predicted and ground-truth events (Huet et al., 2022).

**Directed distance.** Given a set $X$ and a set $Y$, the directed average distance from $X$ to $Y$ is defined as:

$$\text{dist}(X, Y) = \frac{1}{|X|} \int_{x \in X} \min_{y \in Y} |x - y| \, dx. \tag{38}$$

This directed distance measures how far elements of $X$ are from the nearest elements of $Y$, and serves as the basis for affiliation-based Precision and Recall.

**Local affiliation.** Each ground-truth anomaly event $gt_j$ is associated with an *affiliation zone* $I_j$, defined as the region of the time axis closer to $gt_j$ than to any other event. Within each affiliation zone, predicted events are affiliated to the closest ground-truth event, and local distances are computed as:

$$D_j^{\text{precision}} = \text{dist}(\text{pred} \cap I_j, gt_j), \quad D_j^{\text{recall}} = \text{dist}(gt_j, \text{pred} \cap I_j). \tag{39}$$

**Comparison against random sampling.** To normalize distances into probabilities, observed distances are compared against the distribution obtained from random predictions. For ground-truth event $gt_j$, the individual affiliation-based Precision and Recall probabilities are:

$$P_j^{\text{precision}} = \frac{1}{|\text{pred} \cap I_j|} \int_{x \in \text{pred} \cap I_j} F_j^{\text{precision}}(\text{dist}(x, gt_j)) \, dx, \tag{40}$$

$$P_j^{\text{recall}} = \frac{1}{|gt_j|} \int_{y \in gt_j} F_{y,j}^{\text{recall}}(\text{dist}(y, \text{pred} \cap I_j)) \, dy, \tag{41}$$

where $F_j^{\text{precision}}$ and $F_{y,j}^{\text{recall}}$ denote survival functions derived from the random sampling baseline.

**Affiliation-based Precision and Recall.** The final metrics are obtained by averaging over all ground-truth events:

$$Precision_{\text{aff}} = \frac{1}{|S|} \sum_{j \in S} P_j^{\text{precision}}, \quad Recall_{\text{aff}} = \frac{1}{N_r} \sum_{j=1}^{N_r} P_j^{\text{recall}}, \tag{42}$$

where $S = \{j \mid \text{pred} \cap I_j \neq \emptyset\}$.

**Affiliation-based F1-score.** Analogous to the classical definition, the Affiliation-based F1-score is the harmonic mean of the two metrics:

$$F1_{\text{aff}} = 2 \times \frac{Precision_{\text{aff}} \times Recall_{\text{aff}}}{Precision_{\text{aff}} + Recall_{\text{aff}}}. \tag{43}$$

These metrics yield values in $[0, 1]$, with $0.5$ representing the expected performance of a random detector, ensuring robustness and interpretability in the evaluation of time series anomaly detection. The experimental results evaluated with Affiliation-based Metrics are reported in Appendix N.

---

**Algorithm 1** KANomaly Forward (Coarse-to-Fine Reconstruction with Fourier-KAN Mixer)

---

**Input:** $X \in \mathbb{R}^{B \times L \times C}$; patch sizes $\mathcal{P} = [p_1, \ldots, p_m]$ (coarse→fine); Enc/Dec pairs $\{(\text{Enc}_i, \text{Dec}_i)\}_{i=1}^m$ with gridsize $g$, layers $n$, normalization $\texttt{Norm}$, DropPath rate $d$

**Output:** $\hat{X} \in \mathbb{R}^{B \times L \times C}$

1: $X \leftarrow \text{permute}(X)$ to shape $(B, C, L)$
2: **for** $i = 1$ **to** $m$ **do**                    ▷ Coarse → fine multi-scale loop
3:     $p \leftarrow p_i$;  pad $\leftarrow (p - (L \bmod p)) \bmod p$
4:     $X_{\text{in}} \leftarrow \text{pad\_left}(X, \text{pad})$                    ▷ Zero-pad on the left
5:     $Z \leftarrow \text{Enc}_i(X_{\text{in}})$                    ▷ Fourier-KAN Mixer Encoder
6:     $\widehat{X}_{\text{in}} \leftarrow \text{Dec}_i(Z)$                    ▷ Fourier-KAN Mixer Decoder
7:     $\widehat{X} \leftarrow \widehat{X}_{\text{in}}[:, :, \text{pad} :]$                    ▷ Crop to original length
8:     $X \leftarrow X + \widehat{X}$                    ▷ Progressive residual refinement
9: **end for**
10: $\hat{X} \leftarrow \text{permute}(X)$ to shape $(B, L, C)$
11: **return** $\hat{X}$

---

# H    ALGORITHMIC DESCRIPTION OF KANOMALY

We provide a detailed algorithmic description of the proposed KANomaly framework in this subsection. Algorithm 1 outlines the overall forward procedure of KANomaly, which progressively refines

the reconstruction through a coarse-to-fine multi-scale loop. Algorithm 2 presents the Fourier-KAN Mixer Encoder and Decoder, describing how channel-wise, patch-wise, and temporal-wise mixing operations are sequentially applied to capture multi-dimensional dependencies. Algorithm 3 specifies the Fourier-KAN Block, the fundamental building block of the Mixer, which stacks multiple Fourier-KAN layers with residual connections and DropPath.

---

**Algorithm 2** Fourier-KAN Mixer Encoder and Decoder

---

**Input:** $U \in \mathbb{R}^{B \times C \times L'}$ (padded length $L'$), patch size $p$ s.t. $N = L'/p$, gridsize $g$, layers $n$, normalization `Norm`, DropPath rate $d$
**Output:** Same-shape tensor after mixing (encoder or decoder role)
 1: **function** ENC($U$)             ▷ Encoder: Unfold → Channel → Patch → Temporal
 2:     $\tilde{U} \leftarrow$ reshape($U$) to $(B, C, N, p)$             ▷ Unfold to patches
 3:     $\tilde{U} \leftarrow$ Norm($\tilde{U}$);   $\tilde{U} \leftarrow$ FOURIERKANBLOCK($\tilde{U}$, axis $= C, g, n, d$)
 4:     $\tilde{U} \leftarrow$ Norm($\tilde{U}$);   $\tilde{U} \leftarrow$ FOURIERKANBLOCK($\tilde{U}$, axis $= N, g, n, d$)
 5:     $\tilde{U} \leftarrow$ Norm($\tilde{U}$);   $\tilde{U} \leftarrow$ FOURIERKANBLOCK($\tilde{U}$, axis $= p, g, n, d$)
 6:     $\tilde{U} \leftarrow$ Linear($p \rightarrow 1$)($\tilde{U}$)             ▷ Projection along temporal axis
 7:     **return** squeeze($\tilde{U}$) to $(B, C, N)$
 8: **end function**
 9: **function** DEC($V$)          ▷ Decoder: Expand → Temporal → Patch → Channel → Fold
10:     $\tilde{V} \leftarrow$ expand($V$) to $(B, C, N, 1)$
11:     $\tilde{V} \leftarrow$ Linear($1 \rightarrow p$)($\tilde{V}$)             ▷ Inverse projection
12:     $\tilde{V} \leftarrow$ Norm($\tilde{V}$);   $\tilde{V} \leftarrow$ FOURIERKANBLOCK($\tilde{V}$, axis $= p, g, n, d$)
13:     $\tilde{V} \leftarrow$ Norm($\tilde{V}$);   $\tilde{V} \leftarrow$ FOURIERKANBLOCK($\tilde{V}$, axis $= N, g, n, d$)
14:     $\tilde{V} \leftarrow$ Norm($\tilde{V}$);   $\tilde{V} \leftarrow$ FOURIERKANBLOCK($\tilde{V}$, axis $= C, g, n, d$)
15:     **return** reshape($\tilde{V}$) to $(B, C, N \cdot p)$             ▷ Fold (UnPatch)
16: **end function**

---

**Algorithm 3** Fourier-KAN Block with Residual and DropPath

---

**Input:** Tensor $T$ with mixing axis specified by `axis`; gridsize $g$ (number of Fourier components); depth $n$ (stacked Fourier-KAN layers); DropPath rate $d$
**Output:** Mixed tensor $T'$ with same shape as $T$
 1: $Y \leftarrow T$
 2: **for** $\ell = 1$ **to** $n$ **do**             ▷ Stack $n$ Fourier-KAN layers along `axis`
 3:     $Y \leftarrow$ FOURIERKANLAYER($Y$, axis, $g$)     ▷ $\sum_{k=1}^{g}(a_k \cos kx + b_k \sin kx)$ per-axis + KAN composition
 4: $Y \leftarrow$ DropPath$_d$($Y$)
 5: **return** $T + Y$             ▷ Residual connection

---

## I KANOMALY HYPERPARAMETER SETTINGS

Table 4 summarizes the hyperparameter configurations used for each dataset. Batch size denotes the number of samples processed simultaneously during training. Window Size corresponds to the input sequence length. Patch Size indicates the multi-scale segmentation levels applied to the input. DropPath is the stochastic depth rate used to regularize the model. $n$ is the number of KAN layers used in each Fourier-KAN block. $g$ determines the number of frequency components, balancing expressive capacity and computational cost. $r$ is a predefined anomaly ratio used to determine the detection threshold $\delta$, such that exactly $r\%$ of the validation set is classified as anomalous. Norm refers to the type of normalization layer adopted in the Fourier-KAN Mixer. Training Epochs indicate the total number of full passes over the training set. The hyperparameter values in Table 4 were selected through a grid search over the following ranges:

- Number of KAN layers ($n$): $n \in \{1, 2, 3\}$
- Frequency components ($g$): $g \in \{5, 10, 15, 20\}$

- Anomaly ratio ($r$): $r \in \{0.3, 0.4, 0.5, 0.6, 0.7\}$
- Normalization type (Norm): Norm $\in \{\text{BN}, \text{IN}, \text{LN}\}$
- Patch Size: Various multi-scale patching configurations (see Table 5)

A Hyperparameter sensitivity analysis was conducted over these ranges. Details and results of this analysis are provided in Appendix L.

Table 4: Detailed hyperparameter configurations of KANomaly.

| Datasets | Batch size | Window Size | Patch Size | DropPath | $n$ | $g$ | $r$ | Norm | Training Epochs |
|---|---|---|---|---|---|---|---|---|---|
| MSL | 128 | 100 | [20, 10, 5, 2, 1] | 0.3 | 1 | 5 | 0.7 | Instance Norm | 80 |
| PSM | 128 | 100 | [20, 10, 5, 2, 1] | 0.3 | 1 | 15 | 0.5 | Layer Norm | 80 |
| SMAP | 64 | 100 | [20, 10, 5, 2, 1] | 0.3 | 3 | 5 | 0.4 | Batch Norm | 80 |
| SMD | 64 | 100 | [20, 10, 5, 2, 1] | 0.3 | 3 | 5 | 0.5 | Instance Norm | 80 |
| SWaT | 64 | 100 | [20, 10, 5, 2, 1] | 0.3 | 1 | 10 | 0.3 | Layer Norm | 80 |

## J  BASELINE MODELS AND CODE REPOSITORIES

All baseline models were reproduced using official implementations or publicly available source codes provided in the original papers, where applicable. For models that were difficult to reproduce, such as DAGMM, LSTM-VAE, OmniAnomaly, BeatGAN, and THOC, we cited performance values reported in other literature (Xu et al., 2021).

- **Local Outlier Factor** (Breunig et al., 2000) detects anomalies by evaluating the local density of each data point relative to its neighbors, with substantially lower-density points regarded as outliers. Local Outlier Factor is publicly available at `https://scikit-learn.org/stable/modules/generated/sklearn.neighbors.LocalOutlierFactor.html`.
- **One-Class SVM** (Manevitz & Yousef, 2001) learns only from normal data to construct a decision boundary that encloses the normal region, and classifies any data point outside this boundary as an anomaly. One-Class SVM is publicly available at `https://scikit-learn.org/stable/modules/generated/sklearn.svm.OneClassSVM.html`.
- **Isolation Forest** (Liu et al., 2008) determines anomalies based on the number of tree splits required to isolate a data point, considering points that are separated with fewer splits as more likely to be anomalies. Isolation Forest is publicly available at `https://scikit-learn.org/stable/modules/generated/sklearn.ensemble.IsolationForest.html`.
- **DeepSVDD** (Ruff et al., 2018) learns normal data representations using a neural network by mapping them into a compact hyperspherical space, where the distance from the center is used as the anomaly score. DeepSVDD is publicly available at `https://github.com/xuhongzuo/DeepOD/blob/main/deepod/models/time_series/dsvdd.py`.
- **DAGMM** (Zong et al., 2018) detects anomalies by applying a Gaussian Mixture Model (GMM) to a joint representation composed of low-dimensional embeddings from an autoencoder and their reconstruction errors.
- **LSTM-VAE** (Park et al., 2018) combines LSTM with a Variational AutoEncoder (VAE) to effectively reconstruct time series data.
- **OmniAnomaly** (Su et al., 2019) employs a GRU-based stochastic VAE to model long-term dependencies and inherent uncertainty in time series data.
- **BeatGAN** (Zhou et al., 2019) employs a GAN-based reconstruction framework to model normal patterns and detect anomalies via reconstruction error.
- **THOC** (Shen et al., 2020) employs a temporal hierarchical structure for multi-scale clustering to detect anomalies in complex time series, and further incorporates self-supervised learning to enhance detection performance.

- **USAD** (Audibert et al., 2020) employs an adversarial training scheme within an AutoEncoder architecture to enable fast and reliable anomaly detection. USAD is publicly available at `https://github.com/manigalati/usad`.

- **TranAD** (Tuli et al., 2022) enhances Transformer-based architectures for anomaly detection by incorporating adversarial training to improve robustness and meta-learning to adapt to temporal variations. TranAD is publicly available at `https://github.com/imperial-qore/TranAD`.

- **Anomaly Transformer** (Xu et al., 2021) leverages self-attention-based association discrepancy to detect anomalies within time series. Anomaly Transformer is publicly available at `https://github.com/thuml/Anomaly-Transformer`.

- **TimesNet** (Wu et al., 2022) decomposes time series into multiple periods and transforms them into 2D tensors to effectively capture both intra- and inter-period variations. TimesNet is publicly available at `https://github.com/thuml/Time-Series-Library/blob/main/models/TimesNet.py`.

- **DCdetector** (Yang et al., 2023) is a contrastive learning-based method that combines dual attention with contrastive objectives to maximize the representational differences between normal and anomalous data, thereby enhancing the discriminability of learned features. DCdetector is publicly available at `https://github.com/DAMO-DI-ML/KDD2023-DCdetector`.

- **MEMTO** (Song et al., 2023) integrates Transformers with a memory module to dynamically learn normal patterns, and leverages bidirectional deviations in both the input and latent spaces to achieve accurate and robust anomaly detection. MEMTO is publicly available at `https://github.com/gunny97/MEMTO`.

- **FITS** (Xu et al., 2023) is a lightweight model that learns efficient and generalizable frequency representations by leveraging interpolation in the frequency domain with a small number of parameters. FITS is publicly available at `https://github.com/VEWOXIC/FITS`.

- **TFMAE** (Fang et al., 2024) enhances detection performance via a Transformer-based masked autoencoder that employs temporal-frequency masking. TFMAE is publicly available at `https://github.com/LMissher/TFMAE`.

- **CATCH** (Wu et al., 2025) is a transformer-based model that extracts patches in the frequency domain and captures inter-channel relationships within each frequency band using masked attention. CATCH is publicly available at `https://github.com/decisionintelligence/catch`.

- **KAN-AD** (Zhou et al., 2025) is a KAN-based univariate time series anomaly detection model that predicts future time points for detection, leveraging smooth functional mappings with a small number of parameters. KAN-AD is publicly available at `https://github.com/CSTCloudOps/KAN-AD`.

## K  BASELINE HYPERPARAMETER SETTINGS

For the reproduced baselines, we adopted the hyperparameter configurations as described in the original papers or official repositories. As mentioned in Appendix J, for baselines that were difficult to reproduce, we cited performance values reported in prior literature (Xu et al., 2021). In cases where multiple configurations were suggested, we selected those specifically recommended for the datasets used in this study. This ensures a fair comparison under standardized configurations and aligns with prior evaluation protocols.

- **Local Outlier Factor:** The number of neighbors is set to 20.

- **One-Class SVM:** The Radial Basis Function (RBF) kernel is used.

- **Isolation Forest:** The number of estimators is set to 100.

- **DeepSVDD:** The window size is set to 30. The representation dimension is 64, and the hidden layer size is 512. A Transformer is used as the base network.

- **USAD:** The window size is selected from $\{5, 12\}$. The dimension of the latent space is selected from $\{40, 100\}$.

- **TranAD:** The window size is set to 10. The model uses 1 Transformer encoder layer with 2 feed-forward units. Each encoder layer has 64 hidden units, and the dropout rate is set to 0.1.

- **Anomaly Transformer:** The number of encoder layers is 3. The channel number of hidden states ($d_{\mathrm{model}}$) is 512, with 8 attention heads. The balancing parameter $\lambda$, which controls the trade-off between the two parts of the loss function, is set to 3.

- **TimesNet:** The number of blocks is selected from $\{1, 2, 3\}$. The model dimension ($d_{\mathrm{model}}$) is selected from $\{8, 16, 32, 64\}$. The number of frequency components $k$ is chosen from $\{3, 5\}$, where the top-$k$ frequencies are selected based on amplitude values.

- **DCdetector:** The number of Transformer layers is set to 3. The model dimension ($d_{\mathrm{model}}$) is set to 256, and the number of attention heads is set to 1. The patch size is selected from $\{[3, 5], [5, 7], [1, 3, 5], [3, 5, 7]\}$.

- **MEMTO:** The weighting coefficient is set to 0.01. The softmax temperature is set to 0.1, and the number of memory items is set to 10.

- **FITS:** The window size is set to 200, and the downsample rate ($\eta$) is set to 4.

- **TFMAE:** The number of Transformer layers is set to 3. The hidden dimension is set to 128. The window size is 100, and a local window size of 10 is used for calculating local statistical features.

- **CATCH:** The window size is set to 192. The model dimension ($d_{\mathrm{model}}$) is selected from $\{16, 32, 64, 128\}$. The training patch size is selected from $\{8, 16\}$. The testing patch size is selected from $\{4, 8, 16, 32\}$.

- **KAN-AD:** The window size is selected from $\{64, 80, 96\}$. The number of terms in univariate functions is selected from $\{1, 2, 3, 4, 5, 6\}$.

## L  HYPERPARAMETER SENSITIVITY ANALYSIS

We conduct a hyperparameter sensitivity analysis based on the search ranges described in Appendix I. This analysis evaluates the influence of key hyperparameters on anomaly detection performance across five benchmark datasets (MSL, PSM, SMAP, SMD, and SWaT). For each experiment, we vary a single hyperparameter while fixing all others at their default values (see Table 4), allowing us to isolate its effect on model performance. Figure 4 and Table 5 present the results of this analysis.

Figure 4 presents the results for the following four hyperparameters: the number of KAN layers ($n$), the number of frequency components ($g$), the anomaly ratio threshold ($r$), and the normalization method (Norm). As shown in the figure, increasing $n$ improves performance on the SMAP and SMD datasets, whereas it slightly degrades performance on the MSL dataset. When $g$ becomes too large (e.g., $g = 20$), performance consistently drops across all datasets, suggesting potential overfitting or redundancy. Regarding the anomaly ratio $r$, optimal performance is generally achieved within the range of 0.3 to 0.7, with the PSM and SWaT datasets appearing more robust than the MSL dataset. Among normalization methods, InstanceNorm (IN) consistently delivers strong and stable performance, particularly on the MSL and SMD datasets.

Table 5 presents the results of a sensitivity analysis on Patch Size hyperparameter configurations, which control the multi-scale patching of the input time series. We evaluate various coarse-to-fine patching strategies. The default configuration $[20, 10, 5, 2, 1]$ yields the best average F1-score, confirming the effectiveness of coarse-to-fine granularity. Removing coarse-scale patches (NoCoarse) or fine-scale patches (NoFine) significantly reduces performance, emphasizing the importance of capturing both long-term and short-term patterns. Likewise, CoarseOnly, FineOnly, and Equal configurations all result in lower F1-scores than the default, further supporting the necessity of incorporating multi-scale temporal contexts rather than relying on a limited range of scales. The LongTail configuration, which emphasizes coarse context, shows solid performance on certain datasets like SMD and SWaT, but slightly lower than the default.

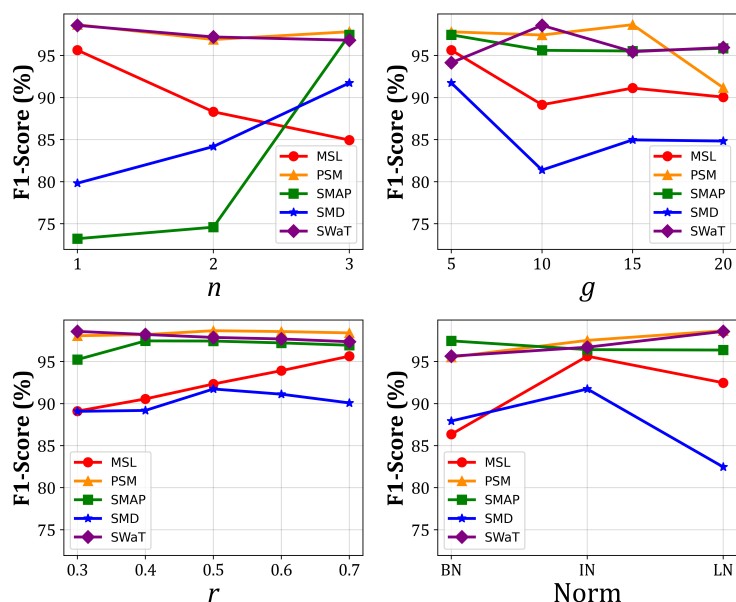

Figure 4: Sensitivity analysis of key hyperparameters in KANomaly. F1-score across five datasets is evaluated while varying four hyperparameters: number of KAN layers ($n$, top left), number of frequency components ($g$, top right), anomaly ratio threshold ($r$, bottom left), and normalization method (Norm, bottom right). In the Norm plot, BN, IN, and LN denote Batch Normalization, Instance Normalization, and Layer Normalization, respectively.

Table 5: Sensitivity analysis of Patch Size hyperparameter configurations in KANomaly. F1-score across five datasets is evaluated under different multi-scale patching setups. All values are reported in %.

| Patch Size | Config Name | Description | MSL | PSM | SMAP | SMD | SWaT | Average F1 |
|---|---|---|---|---|---|---|---|---|
| [20, 10, 5, 2, 1] | Default | Default patching configuration | 95.64 | 98.67 | 97.46 | 91.74 | 98.59 | 96.42 |
| [20, 10] | CoarseOnly | Long-term context only | 93.73 | 98.39 | 95.43 | 79.90 | 96.47 | 92.78 |
| [5, 2, 1] | FineOnly | Local context only | 87.33 | 95.44 | 68.47 | 87.81 | 97.61 | 87.33 |
| [10, 5, 2, 1] | NoCoarse | Without coarse patches | 84.56 | 95.34 | 90.25 | 83.38 | 97.36 | 90.18 |
| [20, 10, 5, 2] | NoFine | Without fine patches | 85.36 | 98.39 | 96.77 | 82.75 | 97.18 | 92.09 |
| [10, 10, 10, 10, 10] | Equal | Fixed patch size | 86.60 | 96.94 | 96.09 | 83.55 | 97.66 | 92.17 |
| [50, 25, 10, 5, 1] | LongTail | Emphasized coarse scale | 91.25 | 96.38 | 91.96 | 91.56 | 97.86 | 93.80 |

These results collectively validate our architectural design and hyperparameter configurations, highlighting the robustness and adaptability of KANomaly across diverse anomaly scenarios.

## M    BAYESIAN SIGNIFICANCE ANALYSIS

To assess the statistical significance of performance improvements across MTSAD models, we adopt the Bayesian signed-rank test (Benavoli et al., 2017), a probabilistic and interpretable alternative to conventional significance testing. This method is applied to the F1-scores obtained from five benchmark datasets (MSL, PSM, SMAP, SMD, and SWaT), using a Region Of Practical Equivalence (ROPE) of 0.5%, which reflects a conservative threshold for considering two models practically equivalent. Intuitively, if the performance difference between two models consistently falls within the ROPE, they are regarded as not significantly different.

Unlike traditional frequentist Null Hypothesis Significance Testing (NHST), which has been widely used in machine learning, the Bayesian signed-rank test avoids rigid binary decisions based on arbitrary thresholds. NHST has often been criticized for promoting black-and-white thinking, where conclusions hinge solely on whether a $p$-value crosses a predefined boundary. In contrast, the Bayesian approach estimates the full posterior distribution over performance differences and di-

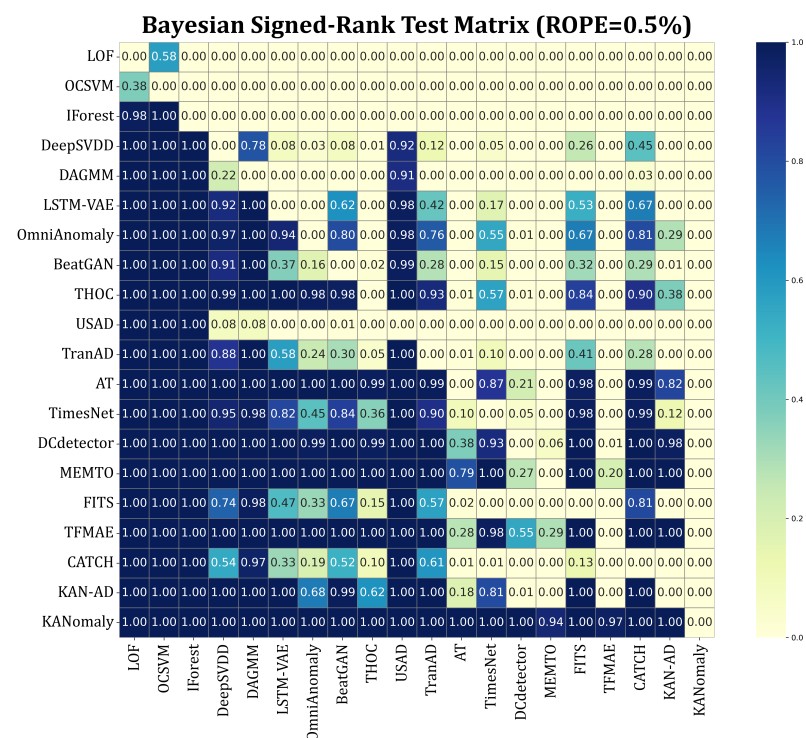

Figure 5: Bayesian signed-rank test matrix across 18 models with ROPE set to 0.5%. Cell $(i, j)$ represents the probability that model $i$ significantly outperforms model $j$.

rectly quantifies the probabilities of superiority, equivalence, or inferiority. This enables principled uncertainty quantification and leads to more informative and robust conclusions in comparative analysis.

Figure 5 presents the resulting pairwise comparison matrix, where each cell $(i, j)$ denotes the posterior probability that the model in row $i$ outperforms the model in column $j$. Darker cells indicate higher confidence in superiority. As shown in the figure, our proposed model, KANomaly, exhibits clear statistical dominance over all baselines, with posterior probabilities approaching 1.00 in most comparisons. These results indicate that the observed improvements are not due to random variation but represent statistically and practically meaningful gains, highlighting the robustness and reliability of KANomaly across diverse datasets and anomaly types.

## N RESULTS WITH AFFILIATION- AND RANGE-BASED METRICS

In addition to the classical point-adjustment metrics reported in the main text, we further evaluate our model using the recently proposed Affiliation-based Metrics (Huet et al., 2022) and Range-based Metrics (Tatbul et al., 2018), which more comprehensively capture the correctness of event-level anomaly detection. Affiliation-based metrics provide a robust evaluation by quantifying the affiliation of predicted anomalies to ground-truth events, while Range-based metrics explicitly account for event duration, partial overlaps, and fragmented predictions.

### N.1 RESULTS WITH AFFILIATION-BASED METRICS

Table 6 reports the performance comparison in terms of Affiliation-based Precision (Aff-P), Recall (Aff-R), and F1-score (Aff-F1) across five real-world datasets. KANomaly consistently achieves competitive performance across all benchmarks. In particular, it achieves either the best or the second-best Aff-F1 score on all datasets, while also delivering the highest overall average performance. These results confirm that KANomaly produces predictions that are not only accurate at the point level but also well-aligned with ground-truth events when evaluated through affiliation.

Table 6: Comparison of Affiliation-based Precision (Aff-P), Recall (Aff-R), and F1-score (Aff-F1) across five real-world datasets. All values are reported in %. **Bold** and underlined numbers denote the best and second-best performance, respectively.

| Dataset | | MSL | | | PSM | | | SMAP | | |
|---|---|---|---|---|---|---|---|---|---|---|
| Model | Venue | Aff-P | Aff-R | Aff-F1 | Aff-P | Aff-R | Aff-F1 | Aff-P | Aff-R | Aff-F1 |
| AT | ICLR-22 | 51.74 | 96.30 | 67.31 | 57.07 | 85.14 | 68.33 | 52.06 | 97.56 | 67.89 |
| TimesNet | ICLR-23 | 55.36 | 77.46 | 64.57 | 74.54 | 50.61 | 60.29 | 50.72 | 60.25 | 55.08 |
| DCdetector | KDD-23 | 50.89 | 97.15 | 66.79 | 51.32 | 77.24 | 61.67 | 51.12 | 98.40 | 67.29 |
| MEMTO | NeurIPS-23 | 52.69 | 95.94 | 68.03 | 54.50 | 83.08 | 65.82 | 51.05 | 98.57 | 67.27 |
| FITS | ICLR-24 | 56.70 | 83.02 | 67.38 | 80.62 | 69.87 | 74.86 | 60.74 | 54.40 | 57.40 |
| TFMAE | ICDE-24 | 51.61 | 94.57 | 66.78 | 54.00 | 81.27 | 64.89 | 49.92 | 97.21 | 65.97 |
| CATCH | ICLR-25 | 53.22 | 98.25 | 69.05 | 72.73 | 96.85 | **83.08** | 44.14 | 68.47 | 53.68 |
| KAN-AD | ICML-25 | 55.34 | 79.66 | 65.31 | 82.85 | 59.02 | 68.93 | 56.50 | 82.80 | 67.16 |
| KANomaly | - | 56.04 | 91.06 | **69.38** | 63.93 | 79.18 | 70.74 | 53.47 | 97.25 | **69.00** |

| Dataset | | SMD | | | SWaT | | | Average | | |
|---|---|---|---|---|---|---|---|---|---|---|
| Model | Venue | Aff-P | Aff-R | Aff-F1 | Aff-P | Aff-R | Aff-F1 | Aff-P | Aff-R | Aff-F1 |
| AT | ICLR-22 | 58.09 | 88.36 | 70.10 | 57.41 | 90.75 | 70.33 | 55.27 | 91.62 | 68.79 |
| TimesNet | ICLR-23 | 65.92 | 74.00 | 69.73 | 52.19 | 41.97 | 46.52 | 59.75 | 60.86 | 59.24 |
| DCdetector | KDD-23 | 51.60 | 90.77 | 65.79 | 52.16 | 97.98 | 68.08 | 51.42 | 92.31 | 65.92 |
| MEMTO | NeurIPS-23 | 60.97 | 96.73 | 74.80 | 59.44 | 84.93 | 69.93 | 55.73 | 91.85 | 69.17 |
| FITS | ICLR-24 | 80.92 | 70.86 | 75.56 | 62.04 | 74.69 | 67.78 | 68.20 | 70.57 | 68.60 |
| TFMAE | ICDE-24 | 59.53 | 88.11 | 71.06 | 56.72 | 92.90 | 70.43 | 54.36 | 90.81 | 67.83 |
| CATCH | ICLR-25 | 70.15 | 98.24 | **81.85** | 56.38 | 95.49 | 70.90 | 59.32 | 91.46 | 71.71 |
| KAN-AD | ICML-25 | 78.69 | 81.69 | 80.16 | 60.32 | 86.58 | 71.10 | 66.74 | 77.95 | 70.53 |
| KANomaly | - | 73.30 | 92.20 | 81.67 | 63.20 | 96.83 | **76.48** | 61.99 | 91.30 | **73.45** |

## N.2 RESULTS WITH RANGE-BASED METRICS

Table 7 summarizes the results based on Range-based Precision (R-P), Recall (R-R), and F1-score (R-F1). Compared to prior methods, KANomaly demonstrates clear improvements on most datasets. For instance, it achieves the highest R-F1 score on MSL, SMAP, and SWaT, as well as the best average performance across all datasets. These findings highlight the effectiveness of KANomaly in detecting anomalies that span continuous intervals, reducing fragmented predictions.

Table 7: Comparison of Range-based Precision (R-P), Recall (R-R), and F1-score (R-F1) across five real-world datasets. All values are reported in %. **Bold** and underlined numbers denote the best and second-best performance, respectively.

| Dataset | | MSL | | | PSM | | | SMAP | | |
|---|---|---|---|---|---|---|---|---|---|---|
| Model | Venue | R-P | R-R | R-F1 | R-P | R-R | R-F1 | R-P | R-R | R-F1 |
| AT | ICLR-22 | 11.72 | 17.41 | 14.01 | 30.66 | 14.77 | 19.94 | 12.26 | 16.88 | 14.21 |
| TimesNet | ICLR-23 | 15.33 | 12.03 | 13.48 | 56.43 | 17.78 | 27.04 | 10.26 | 4.19 | 5.95 |
| DCdetector | KDD-23 | 11.03 | 17.87 | 13.64 | 24.52 | 5.37 | 8.81 | 11.26 | 16.33 | 13.33 |
| MEMTO | NeurIPS-23 | 11.40 | 16.09 | 13.34 | 30.04 | 22.19 | 25.53 | 12.54 | 18.43 | 14.93 |
| FITS | ICLR-24 | 16.73 | 10.28 | 12.74 | 67.21 | 27.65 | 39.18 | 45.28 | 6.14 | 10.81 |
| TFMAE | ICDE-24 | 10.91 | 14.25 | 12.36 | 29.74 | 9.24 | 14.10 | 12.14 | 16.58 | 14.01 |
| CATCH | ICLR-25 | 32.54 | 12.25 | **17.80** | 70.06 | 36.55 | **48.04** | 10.21 | 10.19 | 10.20 |
| KAN-AD | ICML-25 | 10.02 | 12.31 | 11.05 | 67.72 | 26.63 | 38.22 | 8.93 | 15.40 | 11.31 |
| KANomaly | - | 19.90 | 15.55 | 17.46 | 41.84 | 25.50 | 31.69 | 17.92 | 16.31 | **17.08** |

| Dataset | | SMD | | | SWaT | | | Average | | |
|---|---|---|---|---|---|---|---|---|---|---|
| Model | Venue | R-P | R-R | R-F1 | R-P | R-R | R-F1 | R-P | R-R | R-F1 |
| AT | ICLR-22 | 9.41 | 14.18 | 11.31 | 12.86 | 16.21 | 14.34 | 15.38 | 15.89 | 14.76 |
| TimesNet | ICLR-23 | 22.85 | 19.88 | 21.26 | 9.97 | 5.70 | 7.25 | 22.97 | 11.92 | 15.00 |
| DCdetector | KDD-23 | 4.45 | 5.30 | 4.84 | 11.92 | 19.64 | 14.84 | 12.64 | 12.90 | 11.09 |
| MEMTO | NeurIPS-23 | 11.77 | 24.86 | 15.97 | 24.49 | 14.12 | 17.91 | 18.05 | 19.14 | 17.54 |
| FITS | ICLR-24 | 18.93 | 24.56 | 21.38 | 21.84 | 9.84 | 13.57 | 34.00 | 15.69 | 19.54 |
| TFMAE | ICDE-24 | 11.37 | 17.00 | 13.63 | 11.99 | 18.44 | 14.53 | 15.23 | 15.10 | 13.73 |
| CATCH | ICLR-25 | 50.07 | 8.93 | 15.16 | 22.60 | 7.73 | 11.51 | 37.10 | 15.13 | 20.54 |
| KAN-AD | ICML-25 | 17.75 | 33.11 | **23.11** | 29.08 | 14.06 | 18.96 | 26.70 | 20.30 | 20.53 |
| KANomaly | - | 14.10 | 30.59 | 19.30 | 32.54 | 19.12 | **24.09** | 25.26 | 21.41 | **21.92** |

### N.3 DISCUSSION

Overall, the results under both Affiliation- and Range-based Metrics further validate the robustness of KANomaly. While models like FITS yield strong results on selective datasets, KANomaly demonstrates robustness across diverse evaluation metrics and datasets. This indicates that our approach generalizes well across diverse scenarios of multivariate time series anomaly detection.

## O  ADDITIONAL VISUAL ANALYSIS

To further complement the visual analysis presented in Figure 3, additional experiments were conducted using the synthetic TODS dataset (Lai et al., 2021), which provides five canonical anomaly types: Point Global, Point Contextual, Pattern Shapelet, Pattern Seasonal, and Pattern Trend. Unlike real-world datasets where multiple anomaly types may coexist, the synthetic setting allows each anomaly mechanism to be examined independently in a controlled environment. This facilitates a clearer understanding of the model's behavior in detecting fine-grained deviations.

Figure 6 illustrates a comparative visualization of anomaly detection results obtained from TFMAE, FITS, and KANomaly under identical configurations. The frequency-aware baselines TFMAE and FITS generate dispersed or unstable anomaly scores, often failing to temporally localize subtle deviations, particularly in Point Contextual and Pattern Shapelet cases. In contrast, KANomaly produces sharper and temporally aligned anomaly responses across Point Contextual, Pattern Shapelet, and Pattern Seasonal scenarios, demonstrating enhanced sensitivity to localized spectral perturbations.

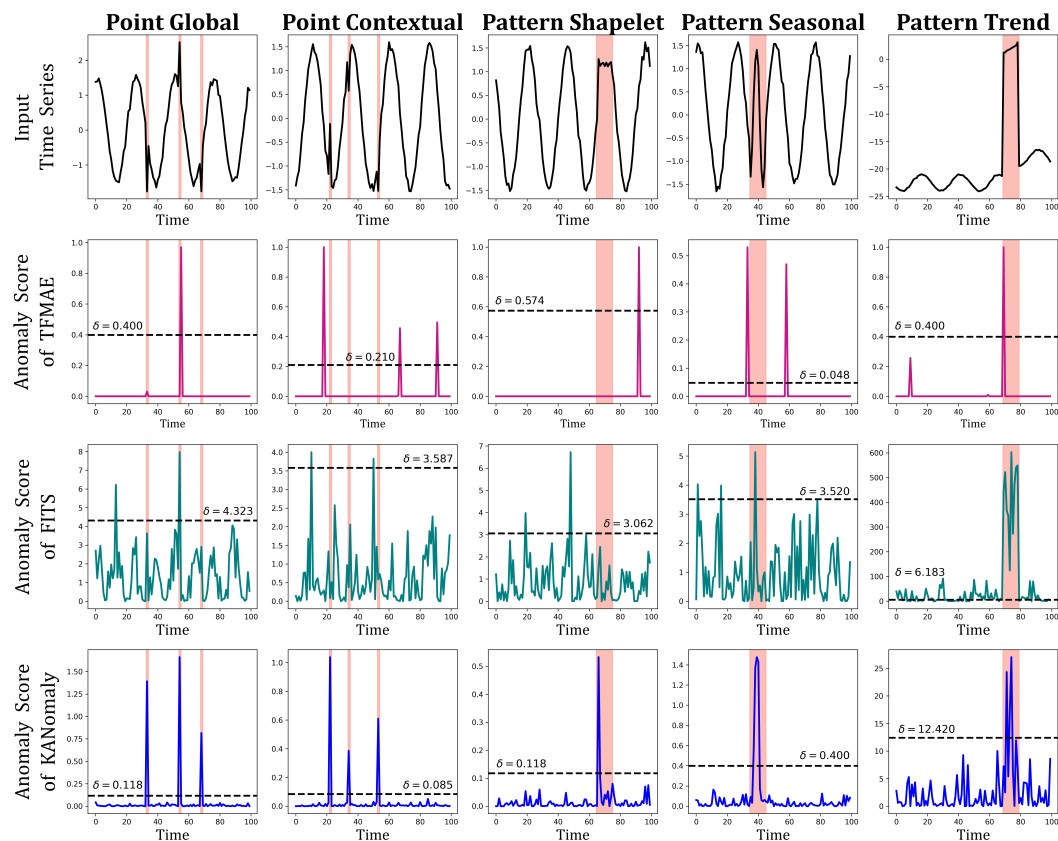

Figure 6: Additional visual analysis of detection results across different anomaly types using TFMAE, FITS, and KANomaly. The top row shows the input time series, and the lower rows show corresponding anomaly scores; red regions indicate ground-truth anomalies.

These findings are consistent with the motivation illustrated in Figure 1(a), where subtle anomalies remain statistically normal in the time domain but become more separable after spectral decomposi-

tion. This difference can be attributed to the architectural design of KANomaly. Rather than treating Fourier features as static input embeddings, learnable Fourier basis functions are directly embedded into the network's functional mappings via the Fourier-KAN blocks, effectively redefining the activation space. This enables the model to adaptively parameterize intrinsic spectral characteristics within the functional domain, rather than relying solely on feature-level frequency representations. These characteristics are typically difficult to capture for models that depend on fixed Fourier encodings or limited frequency-aware mechanisms.

In addition, the coarse-to-fine multi-scale patching strategy further enhances anomaly detection robustness by enabling simultaneous modeling of both short-term abrupt deviations and long-duration pattern-level irregularities. By hierarchically processing patches from fine to coarse temporal resolutions, the model is capable of capturing anomalies of varying durations. This capability works synergistically with the Fourier-KAN blocks, where frequency-domain cues and multi-scale temporal structures are jointly leveraged to improve anomaly localization and discrimination.

Overall, the extended visualization confirms that the proposed KANomaly model exhibits robust detection capability across both point-level and pattern-level anomalies governed by fine-grained spectral variations. This supplementary evidence not only demonstrates the effectiveness of the proposed frequency-aware design but also highlights its advantage in overcoming the inherent limitations of existing frequency-aware baselines.

## P    ADDITIONAL EFFICIENCY ANALYSIS

### P.1    EFFICIENCY ANALYSIS COMPARED TO BASELINE MODELS

To provide a more rigorous assessment beyond the preliminary comparison shown in Figure 3 (right), comprehensive computational efficiency experiments were conducted across five benchmark datasets, namely MSL, PSM, SMAP, SMD, and SWaT. The evaluation included comparisons with representative state-of-the-art baselines such as Anomaly Transformer, TimesNet, DCdetector, MEMTO, FITS, TFMAE, CATCH, and KAN-AD. The analysis covers parameter count, training time per epoch, FLOPs, inference latency, and GPU memory usage, with all measurements conducted under identical experimental settings. GPU memory was evaluated using a fixed batch size of 64 for fairness.

As shown in the detailed computational analysis (see Table 8), KANomaly achieves superior detection accuracy while maintaining favorable computational efficiency. On average, KANomaly utilizes approximately 0.97 million parameters, which is significantly smaller than Anomaly Transformer, TimesNet, MEMTO, and CATCH. Furthermore, KANomaly requires only 2.47 seconds of training time per epoch on average, outperforming baselines such as DCdetector and TFMAE, which record substantially longer epoch durations. KANomaly also demonstrates strong computational efficiency in terms of FLOPs, inference latency, and GPU memory usage, with an average of 70.73 million FLOPs, 1.54 seconds inference latency, and 1,726.64 MiB GPU memory consumption. Notably, it offers a more efficient efficiency–accuracy trade-off compared to resource-intensive models such as CATCH.

Despite its considerably lower computational cost, KANomaly achieves the highest anomaly detection performance among all compared models. Specifically, it records an average Point-Adjusted F1 of 96.42 percent, Affiliation-based F1 of 73.45 percent, and Range-based F1 of 21.92 percent, outperforming all baselines. These results indicate that the model's advantages are not achieved through increased computational complexity but stem from its efficient architectural design and effective functional integration of Fourier basis functions. Consequently, the balance between accuracy and efficiency confirms that the proposed method is suitable for real-world deployment, even under resource-constrained conditions.

In conclusion, KANomaly delivers state-of-the-art anomaly detection performance while operating with substantially lower computational cost than competing baselines. This demonstrates that the architectural design provides both practical scalability and strong detection capability, validating its suitability for industrial and high-dimensional multivariate applications.

Table 8: Computational efficiency and performance comparison between KANomaly and state-of-the-art baseline models across five benchmark datasets (MSL, SMAP, PSM, SMD, and SWaT). KANomaly achieves the best overall results while maintaining high efficiency.

| Dataset | Model | F1 | Aff-F1 | Range-F1 | Params | Train. Time (sec/epoch) | FLOPs (M) | Infer. Time (sec) | GPU Mem (MiB) |
|---|---|---|---|---|---|---|---|---|---|
| MSL | AT | 92.55 | 67.31 | 14.01 | 4,863,055 | 49.47 | 516.86 | 15.16 | 1990.98 |
| | TimesNet | 81.80 | 64.57 | 13.48 | 1,765,623 | 25.99 | 584.14 | 11.20 | 315.80 |
| | DCdetector | 93.34 | 66.79 | 13.64 | 890,935 | 104.39 | 1279.44 | 0.61 | 9961.60 |
| | MEMTO | 93.82 | 68.03 | 13.34 | 5,955,182 | 0.10 | 525.74 | 0.10 | 7445.68 |
| | FITS | 76.66 | 67.38 | 12.74 | 2,600 | 0.25 | 0.14 | 0.21 | 48.18 |
| | TFMAE | 92.56 | 66.78 | 12.36 | 706,048 | 21.52 | 76.77 | 0.32 | 483.79 |
| | CATCH | 66.17 | 69.05 | 17.80 | 210,881,152 | 206.14 | 12065.40 | 532.01 | 6087.02 |
| | KAN-AD | 84.99 | 65.31 | 11.05 | 4,491 | 2.68 | 1.55 | 1.14 | 195.92 |
| | KANomaly (ours) | 95.64 | 69.38 | 17.46 | 573,719 | 0.35 | 55.49 | 0.27 | 1130.79 |
| PSM | AT | 97.71 | 68.33 | 19.94 | 4,801,585 | 108.57 | 510.72 | 17.93 | 1986.39 |
| | TimesNet | 97.40 | 60.29 | 27.04 | 4,694,169 | 71.03 | 1556.40 | 16.72 | 340.91 |
| | DCdetector | 97.48 | 61.67 | 8.81 | 894,745 | 130.66 | 1080.02 | 0.67 | 13606.69 |
| | MEMTO | 97.97 | 65.82 | 25.53 | 5,862,962 | 0.28 | 518.06 | 0.11 | 18384.40 |
| | FITS | 93.36 | 74.86 | 39.18 | 2,600 | 0.23 | 0.06 | 0.21 | 33.86 |
| | TFMAE | 97.86 | 64.89 | 14.10 | 683,008 | 46.47 | 71.86 | 0.35 | 482.38 |
| | CATCH | 91.85 | 83.08 | 48.04 | 3,772,720 | 19.43 | 101.53 | 268.93 | 377.23 |
| | KAN-AD | 95.92 | 68.93 | 38.22 | 5,241 | 3.68 | 1.96 | 1.05 | 186.54 |
| | KANomaly (ours) | 98.67 | 70.74 | 31.69 | 1,001,375 | 0.99 | 52.60 | 0.55 | 1811.54 |
| SMAP | AT | 96.38 | 67.89 | 14.21 | 4,801,585 | 112.68 | 510.72 | 87.65 | 1983.58 |
| | TimesNet | 68.45 | 55.08 | 5.95 | 1,761,753 | 58.71 | 583.75 | 63.16 | 300.04 |
| | DCdetector | 96.41 | 67.29 | 13.33 | 883,225 | 278.28 | 895.41 | 1.62 | 13626.96 |
| | MEMTO | 96.36 | 67.27 | 14.93 | 5,862,962 | 0.28 | 518.06 | 0.50 | 21076.83 |
| | FITS | 68.19 | 57.40 | 10.81 | 2,600 | 0.23 | 0.06 | 0.25 | 37.90 |
| | TFMAE | 96.49 | 65.97 | 14.01 | 683,008 | 47.69 | 71.86 | 0.63 | 482.38 |
| | CATCH | 70.00 | 53.68 | 10.20 | 53,464,208 | 96.25 | 1403.51 | 945.61 | 1923.65 |
| | KAN-AD | 86.86 | 67.16 | 11.31 | 4,491 | 3.43 | 0.70 | 4.38 | 168.72 |
| | KANomaly (ours) | 97.46 | 69.00 | 17.08 | 1,002,419 | 1.07 | 52.60 | 1.22 | 1285.14 |
| SMD | AT | 90.56 | 70.10 | 11.31 | 4,828,222 | 5.50 | 513.38 | 1.20 | 1987.97 |
| | TimesNet | 84.59 | 69.73 | 21.26 | 4,697,510 | 6.34 | 2343.95 | 2.33 | 985.99 |
| | DCdetector | 83.23 | 65.79 | 4.84 | 867,366 | 8.10 | 751.08 | 2.10 | 6887.28 |
| | MEMTO | 91.55 | 74.80 | 15.97 | 5,902,924 | 1.48 | 521.39 | 0.81 | 22093.75 |
| | FITS | 80.32 | 75.56 | 21.38 | 2,600 | 0.38 | 0.10 | 0.27 | 43.09 |
| | TFMAE | 91.24 | 71.06 | 13.63 | 692,992 | 248.79 | 90.80 | 0.89 | 482.73 |
| | CATCH | 79.65 | 81.85 | 15.16 | 229,708,784 | 2604.56 | 9348.48 | 4850.65 | 7826.10 |
| | KAN-AD | 83.95 | 80.16 | 23.11 | 9,845 | 0.46 | 2.50 | 0.26 | 212.42 |
| | KANomaly (ours) | 91.74 | 81.67 | 19.30 | 1,246,619 | 4.58 | 94.66 | 1.97 | 1894.23 |
| SWaT | AT | 83.68 | 70.33 | 14.34 | 4,854,859 | 269.93 | 516.04 | 0.99 | 1990.69 |
| | TimesNet | 92.62 | 46.52 | 7.25 | 1,765,107 | 215.77 | 584.09 | 70.29 | 303.13 |
| | DCdetector | 96.42 | 68.08 | 14.84 | 909,875 | 12.15 | 1826.65 | 3.02 | 11595.47 |
| | MEMTO | 95.17 | 69.93 | 17.91 | 5,942,886 | 0.82 | 524.72 | 0.63 | 21484.94 |
| | FITS | 88.93 | 67.78 | 13.57 | 2,600 | 0.42 | 0.13 | 0.36 | 55.41 |
| | TFMAE | 97.52 | 70.43 | 14.53 | 702,976 | 180.80 | 85.20 | 0.71 | 484.86 |
| | CATCH | 88.48 | 70.90 | 11.51 | 230,106,816 | 2661.13 | 13269.47 | 3622.39 | 12820.03 |
| | KAN-AD | 93.17 | 71.10 | 18.96 | 6,551 | 19.79 | 0.70 | 6.49 | 172.32 |
| | KANomaly (ours) | 98.59 | 76.48 | 24.09 | 1,062,975 | 5.37 | 98.31 | 3.71 | 2511.51 |

## P.2 EFFICIENCY ANALYSIS COMPARED TO ABLATION MODELS

To address the concern regarding the efficiency-performance trade-off in the ablation configurations, a detailed computational analysis was conducted for all ablation models reported in Table 2 of the main paper. The evaluation was performed across the five standard benchmark datasets (MSL, PSM, SMAP, SMD, and SWaT), using identical experimental settings. The analysis includes parameter count, training time per epoch, FLOPs, inference latency, and GPU memory usage. GPU memory was consistently measured with a fixed batch size of 64 to ensure fair comparison.

The results demonstrate that KANomaly achieves the best detection performance while maintaining a balanced computational profile. For instance, the full model typically operates with fewer than 1.3 million parameters and achieves competitive or superior FLOPs and inference latency compared to most ablation variants. Although certain ablation models such as "without Temporal Mixing" and "without Channel Mixing" incur slight reductions in computational requirements, these variants exhibit notable degradation in accuracy, indicating the importance of tri-dimensional interactions. While the full model sometimes incurs higher computational cost compared to ablation variants, this is expected given the purpose of ablation, which intentionally removes components to simplify the architecture. On the other hand, configurations such as "without Fourier-KAN" tend to increase

Table 9: Computational efficiency and performance comparison between KANomaly and its ablation variants across five benchmark datasets (MSL, SMAP, PSM, SMD, and SWaT).

| Dataset | Variations | F1 | Params | Train. Time (sec/epoch) | FLOPs (M) | Infer. Time (sec) | GPU Mem (MiB) |
|---|---|---|---|---|---|---|---|
| MSL | KANomaly (ours) | 95.64 | 573,719 | 0.35 | 55.49 | 0.27 | 1130.79 |
| | (1) w/o Fourier-KAN (→ MLP) | 85.48 | 116,831 | 0.63 | 11.67 | 0.22 | 217.39 |
| | (2) w/o Fourier-KAN (→ Cheby-KAN) | 83.97 | 344,279 | 0.51 | 33.58 | 0.28 | 1169.98 |
| | (3) w/o Fourier-KAN (→ Vanilla-KAN) | 82.40 | 573,719 | 2.04 | 50.02 | 1.54 | 9670.90 |
| | (4) w/o Multi-Scale Patching (Fixed Patch Size) | 86.60 | 322,655 | 0.56 | 41.97 | 0.27 | 1125.44 |
| | (5) w/o Patching | 88.98 | 260,801 | 0.36 | 17.15 | 0.21 | 160.97 |
| | (6) Fine-to-Coarse Patch (Inverse Patch Scale) | 88.54 | 573,719 | 0.43 | 55.49 | 0.31 | 1128.06 |
| | (7) w/o Channel Mixing | 85.77 | 271,219 | 0.51 | 25.02 | 0.26 | 756.66 |
| | (8) w/o Patch Mixing | 86.85 | 313,219 | 0.53 | 34.92 | 0.28 | 758.29 |
| | (9) w/o Temporal Mixing | 87.27 | 563,119 | 0.50 | 51.09 | 0.27 | 762.58 |
| | (10) Fourier features + MLP | 80.72 | 78,151 | 0.54 | 5.37 | 0.28 | 135.42 |
| PSM | KANomaly (ours) | 98.67 | 1,001,375 | 0.99 | 52.60 | 0.55 | 1811.54 |
| | (1) w/o Fourier-KAN (→ MLP) | 97.66 | 68,687 | 1.22 | 3.88 | 0.48 | 146.06 |
| | (2) w/o Fourier-KAN (→ Cheby-KAN) | 97.22 | 534,335 | 1.12 | 28.24 | 0.52 | 1071.83 |
| | (3) w/o Fourier-KAN (→ Vanilla-KAN) | 97.69 | 667,775 | 3.49 | 33.46 | 1.78 | 9750.78 |
| | (4) w/o Multi-Scale Patching (Fixed Patch Size) | 96.94 | 248,255 | 0.93 | 34.15 | 0.35 | 1768.65 |
| | (5) w/o Patching | 98.14 | 638,601 | 0.46 | 18.80 | 0.22 | 304.56 |
| | (6) Fine-to-Coarse Patch (Inverse Patch Scale) | 95.83 | 1,001,375 | 1.21 | 52.60 | 0.56 | 1806.48 |
| | (7) w/o Channel Mixing | 98.16 | 813,723 | 0.96 | 33.73 | 0.45 | 1232.02 |
| | (8) w/o Patch Mixing | 95.38 | 219,723 | 0.94 | 24.73 | 0.45 | 1216.46 |
| | (9) w/o Temporal Mixing | 97.26 | 969,423 | 0.94 | 46.77 | 0.46 | 1233.62 |
| | (10) Fourier features + MLP | 89.40 | 30,007 | 0.93 | 1.65 | 0.38 | 90.03 |
| SMAP | KANomaly (ours) | 97.46 | 1,002,419 | 1.07 | 52.60 | 1.22 | 1285.14 |
| | (1) w/o Fourier-KAN (→ MLP) | 68.43 | 69,731 | 1.11 | 3.88 | 0.56 | 109.29 |
| | (2) w/o Fourier-KAN (→ Cheby-KAN) | 69.24 | 602,099 | 1.40 | 31.72 | 1.11 | 1709.11 |
| | (3) w/o Fourier-KAN (→ Vanilla-KAN) | 68.39 | 1,002,419 | 5.77 | 47.38 | 9.98 | 13362.42 |
| | (4) w/o Multi-Scale Patching (Fixed Patch Size) | 96.09 | 249,155 | 1.33 | 34.15 | 1.22 | 1271.34 |
| | (5) w/o Patching | 90.14 | 638,001 | 0.53 | 18.80 | 0.40 | 201.68 |
| | (6) Fine-to-Coarse Patch (Inverse Patch Scale) | 94.30 | 1,002,419 | 1.39 | 52.60 | 1.24 | 1285.14 |
| | (7) w/o Channel Mixing | 69.30 | 814,419 | 1.15 | 33.73 | 0.93 | 866.89 |
| | (8) w/o Patch Mixing | 76.01 | 220,419 | 1.14 | 24.73 | 0.93 | 856.70 |
| | (9) w/o Temporal Mixing | 74.90 | 970,119 | 1.16 | 46.77 | 0.93 | 869.56 |
| | (10) Fourier features + MLP | 68.24 | 31,051 | 0.85 | 1.65 | 0.57 | 70.66 |
| SMD | KANomaly (ours) | 91.74 | 1,246,619 | 4.58 | 94.66 | 1.97 | 1894.23 |
| | (1) w/o Fourier-KAN (→ MLP) | 87.78 | 84,871 | 4.22 | 6.77 | 0.88 | 155.67 |
| | (2) w/o Fourier-KAN (→ Cheby-KAN) | 85.58 | 748,019 | 7.50 | 57.00 | 1.81 | 2588.30 |
| | (3) w/o Fourier-KAN (→ Vanilla-KAN) | 88.27 | 1,246,619 | 33.68 | 85.24 | 25.51 | 19931.00 |
| | (4) w/o Multi-Scale Patching (Fixed Patch Size) | 83.55 | 493,355 | 7.52 | 66.61 | 2.01 | 1880.24 |
| | (5) w/o Patching | 80.29 | 686,941 | 1.51 | 31.53 | 0.54 | 271.75 |
| | (6) Fine-to-Coarse Patch (Inverse Patch Scale) | 82.22 | 1,246,619 | 7.57 | 94.66 | 2.01 | 1894.23 |
| | (7) w/o Channel Mixing | 86.58 | 813,419 | 5.63 | 51.19 | 1.49 | 1271.80 |
| | (8) w/o Patch Mixing | 89.54 | 465,119 | 5.50 | 52.33 | 1.50 | 1265.81 |
| | (9) w/o Temporal Mixing | 90.86 | 1,214,819 | 5.90 | 85.84 | 1.49 | 1278.67 |
| | (10) Fourier features + MLP | 84.84 | 46,191 | 3.83 | 2.99 | 0.90 | 98.52 |
| SWaT | KANomaly (ours) | 98.59 | 1,062,975 | 5.37 | 98.31 | 3.71 | 2511.51 |
| | (1) w/o Fourier-KAN (→ MLP) | 96.68 | 108,727 | 5.22 | 10.57 | 3.14 | 281.41 |
| | (2) w/o Fourier-KAN (→ Cheby-KAN) | 97.22 | 584,895 | 5.26 | 54.45 | 3.44 | 1574.38 |
| | (3) w/o Fourier-KAN (→ Vanilla-KAN) | 97.40 | 797,375 | 17.43 | 69.07 | 12.92 | 14701.91 |
| | (4) w/o Multi-Scale Patching (Fixed Patch Size) | 97.66 | 560,955 | 3.17 | 73.22 | 1.70 | 2457.32 |
| | (5) w/o Patching | 95.36 | 505,141 | 0.77 | 30.91 | 0.40 | 406.42 |
| | (6) Fine-to-Coarse Patch (Inverse Patch Scale) | 97.41 | 1,062,975 | 5.39 | 98.31 | 3.70 | 2512.12 |
| | (7) w/o Channel Mixing | 96.87 | 542,623 | 3.72 | 46.05 | 2.53 | 1704.33 |
| | (8) w/o Patch Mixing | 95.77 | 541,823 | 3.74 | 60.31 | 2.54 | 1705.88 |
| | (9) w/o Temporal Mixing | 96.42 | 1,041,623 | 3.72 | 90.30 | 2.52 | 1714.46 |
| | (10) Fourier features + MLP | 82.37 | 70,047 | 3.13 | 4.84 | 1.79 | 166.58 |

computational cost without corresponding improvements, further supporting the effectiveness of the final architectural design.

Overall, the ablation cost analysis confirms that the proposed structural components contribute not only to performance enhancement but also to computational stability. Removing key components often leads to disproportionate reductions in detection accuracy relative to savings in computational cost. These findings validate the efficiency and necessity of the integrated architectural design adopted in KANomaly. The full analysis is provided in Table 9 for clarity and reproducibility.

## Q   THE USE OF LARGE LANGUAGE MODELS (LLMS)

We used Large Language Models (LLMs) solely for minor editing purposes, such as polishing grammar and correcting typographical errors during the paper writing process. LLMs were not employed for any other purpose beyond these linguistic refinements.

