# OpenReview forum: "KANomaly: Fourier-KAN-based Multi-Scale Patch Mixer for Multivariate Time Series Anomaly Detection"
_ICLR.cc/2026/Conference — Submitted to ICLR 2026_

### Official Review · Reviewer_PNn3 · 2025-10-25

**Soundness:** 2
**Presentation:** 3
**Contribution:** 1
**Rating:** 2
**Confidence:** 5

**Summary:**

To address the limitations of existing MTSAD methods, such as their implicit learning of temporal and channel dependencies, insufficient utilization of frequency domain information, and challenges in capturing a wide range of short- and long-term anomalies, the authors propose Kanomaly. Kanomaly replaces the spline network in KAN with Fourier series and incorporates multi-scale patching and blending techniques, achieving state-of-the-art detection performance.

**Strengths:**

1. Clear and well-designed figures that enable readers to quickly grasp the motivation and methodological details.
2. Comprehensive equations that clearly convey the computational aspects of the model.

**Weaknesses:**

1. All key modules have been previously explored, such as Fourier Transform in KAN (KAN-AD) [1], Patching (PatchTST) [2], and Patch Mix (TimeMixer) [3], but they lack substantial innovation.
2. The data in the visual analysis is synthesized by TODS, not from the real data set, and the data synthesized by TODS is a single indicator, which does not match the MTS problem that the method in this article wants to solve. In addition, the basic shape of the synthetic data is a sine function, which is too easy for contemporary models. (Detailed in Questions)
3. There is a misunderstanding of some related work, and it is ignored as a baseline method. (Detailed in Questions)

**Questions:**

1. The authors claim in section 2.2 that Kanomaly is the first KAN-based method for MTSAD. However, several prior attempts have used KAN to address MTSAD [4] [5]. Additionally, even the KAN-AD [1] method referenced by the authors includes MTSAD benchmarks. These approaches should be considered as part of the baseline.

2. In Section 5.2, a visualization analysis is presented (detailed in C.1). However, this visualization is not based on actual datasets (such as SMD), but rather uses synthetic data. The training set of TODS consists of regular sine waves (as indicated in line 8 of Listing 1), which are too simple for deep models. This simplicity undermines the argument that Kanomaly can address a wide range of anomalies. Can Kanomaly handle a variety of anomalies on real datasets?

3. It is worth noting that the dataset synthesized by TODS is based on UTS, which indicates that Kanomaly is capable of handling univariate tasks. From this perspective, what distinguishes MTS from UTS in terms of their unique characteristics? Furthermore, how does Kanomaly perform when applied to univariate datasets?



[1] https://dl.acm.org/doi/10.1145/3746709.3746752

[2] https://iclr.cc/virtual/2023/poster/10876

[3] https://iclr.cc/virtual/2024/poster/19347

[4] https://link.springer.com/article/10.1007/s10489-025-06650-8

[5] https://icml.cc/virtual/2025/poster/45584

---

> ### Author Response · Authors · 2025-11-22
> **Response to Reviewer PNn3 (Part 1)**
>
> We greatly appreciate the reviewer’s time and effort in providing such a detailed and constructive evaluation. The thoughtful feedback and careful assessment are sincerely valued. Below, we provide comprehensive responses for each comment and question raised by the reviewer.
>
> >**W1: On the Novelty of Architectural Design and Functional Integration.**
>
> The reviewer pointed out that core modules such as Fourier-KAN (KAN-AD), patching (PatchTST), and patch mixing (TimeMixer) have been separately explored in previous studies. However, the contribution of this work does not lie in the introduction of individual components but in the architectural design and functional integration specifically tailored for the Multivariate Time Series Anomaly Detection (MTSAD) task.
>
> KAN-AD applies KAN primarily to Univariate Time Series Anomaly Detection (UTSAD) settings and does not incorporate the coarse-to-fine multi-scale patching strategy or any mechanism for explicit channel-, patch-, or temporal-mixing. PatchTST employs static patch embedding in the input domain but does not implement hierarchical or coarse-to-fine multi-scale patching. Similarly, TimeMixer applies Seasonal and Trend Mixing but does not perform Patch Mixing, nor does it integrate learnable Fourier basis functions within a functional-level KAN framework. Moreover, PatchTST and TimeMixer are fundamentally exclusively designed for forecasting rather than anomaly detection, and their design philosophies differ substantially from that of our work. This distinction further underscores that the problem space addressed in our study remains unexplored.
>
> In contrast, the proposed architecture first integrates these three components specifically for the Multivariate Time Series Anomaly Detection task. Its originality lies not in the use of individual modules but in their novel combination and synergistic design. KANomaly uniquely combines:
> (1) Learnable Fourier basis functions integrated at the KAN functional level. This enables explicit spectral modeling directly within the activation space.
> (2) A coarse-to-fine multi-scale patching strategy specifically designed to capture both localized and extended anomaly patterns. It ensures robustness to varying anomaly durations.
> (3) A tri-dimensional mixer that operates across channel, patch, and temporal dimensions. This facilitates effective inter-variable interaction and captures local-global dependencies.
>
> The effectiveness of this joint integration is empirically validated through consistent state-of-the-art performance across all benchmark datasets, as reported in Table 1 of the main paper. Furthermore, direct comparison with KAN-AD confirms that the observed performance improvement does not simply originate from adopting KAN as a backbone, but from the proposed architectural modifications. (Further details are discussed in the response to the reviewer’s separate question regarding the comparison with KAN-AD.) These modifications introduce spectral adaptivity, multi-scale temporal sensitivity, and inter-variable interaction modeling, none of which are achieved by the prior methods cited.
>
> In summary, while individual components have been explored independently in different contexts, their functional integration and architectural synergy specifically tailored for Multivariate Time Series Anomaly Detection are novel contributions of this work. The empirical results demonstrate that this design yields substantial improvements over existing approaches, supporting the originality and effectiveness of the proposed methodology.

---

> ### Author Response · Authors · 2025-11-22
> **Response to Reviewer PNn3 (Part 2)**
>
> >**W2 & Q2: On the Validity of Synthetic TODS-Based Visualization and Its Implications for Generalizability to Real-World MTS Anomalies.**
>
> The purpose of using TODS might not have been clearly conveyed, and clarification is provided as follows. The synthetic visualization presented in Section 5.2 (Figure 3 (Left)) was intentionally designed using TODS to analyze whether the model can accurately capture diverse anomaly patterns that may emerge under each simulated scenario, while providing a controlled and interpretable environment where distinct anomaly types can be independently evaluated. TODS offers five canonical anomaly categories, namely Point Global, Point Contextual, Pattern Shapelet, Pattern Seasonal, and Pattern Trend. This allows a structured analysis that is infeasible on actual datasets such as SMD, where multiple anomaly types often coexist and are entangled. The visual comparison is therefore used to isolate the functional capabilities of KANomaly and to demonstrate its versatility in detecting diverse anomaly patterns.
>
> The assertion that the synthetic data is too simple for deep learning models overlooks an important aspect. Even within a sinusoidal base, anomaly forms such as contextual shifts and shapelet distortions (the 'Point Contextual' and 'Pattern Shapelet' anomaly types defined in TODS) remain challenging for contemporary deep learning models. To further support this point, additional experiments were conducted on the TODS dataset using representative frequency-aware baselines such as TFMAE and FITS, with results provided in "Appendix O. Additional Visual Analysis" (Figure 6). In these visualizations, TFMAE and FITS exhibit dispersed and unstable anomaly responses and often fail to temporally localize subtle contextual or pattern deviations. By contrast, KANomaly produces sharper and more temporally aligned responses, particularly for Point Contextual, Pattern Shapelet, and Pattern Seasonal anomalies. This comparison directly contradicts the claim that synthetic data is trivial for contemporary models and simultaneously provides supporting evidence of the superior performance of KANomaly.
>
> Furthermore, the intention of this visualization is independent from performance validation. Table 1 in the main manuscript already demonstrates that KANomaly achieves the best detection performance across all benchmark datasets, including real-world multivariate time series. These results validate the method’s effectiveness under complex distributions and industrial conditions. The visual analysis in Section 5.2 is supplementary and conceptual, highlighting detectability across anomaly types rather than real-world generalization, which is already empirically supported by the main results.
>
> In summary, the use of TODS enables a controlled qualitative understanding of model behavior across clearly defined anomaly types. The observed advantage of KANomaly over strong frequency-aware baselines in this controlled scenario further reinforces its adaptability. Combined with superior quantitative performance on real datasets, these findings provide strong evidence that KANomaly is capable of handling a wide range of anomalies beyond the synthetic setting.

---

> ### Author Response · Authors · 2025-11-22
> **Response to Reviewer PNn3 (Part 3)**
>
> >**W3 & Q1.1: On the Clarification of Prior KAN-Based Work for Multivariate Time Series Anomaly Detection (MTSAD).**
>
> The concern regarding the originality of KANomaly as the first KAN-based method for Multivariate Time Series Anomaly Detection (MTSAD) arises from prior attempts such as KAN-AD [1] and MTAD-Kanformer [2]. However, these approaches do not address MTSAD in the manner proposed in this work.
>
> KAN-AD is fundamentally designed as a Univariate Time Series Anomaly Detection (UTSAD) framework, as explicitly stated in Section 2.1 (Problem Statement) of the KAN-AD paper. In its multivariate experiments, the input tensor is reshaped from (batch size, window length, n_features) to (batch size × n_features, window length), treating each channel as an independent univariate time series, as also stated in Section 4.7 (Performance on Multivariate Time Series) of the KAN-AD paper. This procedure intentionally ignores inter-channel dependencies, meaning that KAN-AD does not function as a true multivariate model. In contrast, the core contribution of KANomaly lies not in simply applying KAN, but in designing the Fourier-KAN Mixer to explicitly capture nonlinear inter-channel interactions, which are essential to multivariate anomaly detection. This architectural design enables modeling of inherent correlation structures among variables, a capability that is unattainable in channel-independent formulations such as KAN-AD. Paradoxically, KAN-AD reinforces the necessity and novelty of the proposed KANomaly architecture by demonstrating the limitations of reshaping-based univariate extensions when applied to Multivariate Time Series scenarios.
>
> Similarly, the prior work suggested by the reviewer, such as MTAD-Kanformer, appears to modify only the output projection of a variant of Anomaly Transformer by replacing the linear layer with a vanilla KAN, while leaving the transformer backbone unchanged. These approaches do not reparameterize internal functional spaces using Fourier-KAN nor do they implement hierarchical mixing across channel, patch, and temporal dimensions. As such, they are qualitatively distinct from the method presented in this work.
>
> Therefore, to improve clarity, the manuscript wording will be revised as follows: **“KANomaly is the first Fourier-KAN-based method explicitly tailored for Multivariate Time Series Anomaly Detection (MTSAD).”**
>
> The review has helped further clarify the contribution and necessity of our work, and has strengthened its novelty and originality. We appreciate the insightful feedback.
>
> [1] Zhou et al., "KAN-AD: Time Series Anomaly Detection with Kolmogorov–Arnold Networks." ICML 2025
>
> [2] Xie et al., "MTAD-Kanformer: multivariate time-series anomaly detection via kan and transformer." Applied Intelligence 2025

---

> ### Author Response · Authors · 2025-11-22
> **Response to Reviewer PNn3 (Part 4)**
>
> >**Q1.2: On the Experimental Comparison with KAN-AD and Isolation of KANomaly’s Architectural Contributions.**
>
> A comparative analysis with KAN-AD was conducted to assess the impact of the proposed architectural enhancements beyond the KAN backbone. While KAN-AD primarily focuses on univariate anomaly detection, a multivariate configuration was implemented as an auxiliary experiment in its original paper. To ensure a rigorous and comprehensive analysis, the multivariate version of KAN-AD was reproduced using the official repository and evaluated under identical experimental settings across all five benchmark datasets (MSL, PSM, SMAP, SMD, and SWaT). To implement MTSAD for KAN-AD, each multivariate input was reshaped from (batch size, window length, n_features) to (batch size×n_features, window length), following the auxiliary configuration described in its paper.
>
> | Model | Venue | MSL F1 | PSM F1 | SMAP F1 | SMD F1 | SWaT F1 | AVG F1 |
> |------------|------------|-------|-------|-------|-------|-------|-------|
> | AT | ICLR-22    | 92.55 | 97.71 | 96.38 | 90.56 | 83.68 | 92.18 |
> | TimesNet   | ICLR-23    | 81.80 | 97.40 | 68.45 | 84.59 | 92.62 | 84.97 |
> | DCdetector | KDD-23 | 93.34 | 97.48 | 96.41 | 83.23 | 96.42 | 93.38 |
> | MEMTO | NeurIPS-23 | 93.82 | 97.97 | 96.36 | 91.55 | 95.17 | 94.97 |
> | FITS  | ICLR-24 | 76.66 | 93.36 | 68.19 | 80.32 | 88.93 | 81.49 |
> | TFMAE | ICDE-24    | 92.56 | 97.86 | 96.49 | 91.24 | 97.52 | 95.13 |
> | KAN-AD | ICML-25 | 84.99 | 95.92 | 86.86 | 83.95 | 93.17 | 88.98 |
> | **KANomaly (ours)** | *–* | **95.64** | **98.67** | **97.46** | **91.74** | **98.59** | **96.42** |
>
> As presented in the performance comparison table (see Table), KANomaly consistently outperforms KAN-AD on every dataset, achieving an average F1 score of 96.42 %, substantially higher than 88.98 % reported for KAN-AD. These results confirm that the observed improvement is not merely attributable to inheriting the KAN backbone, but rather stems from the architectural innovations proposed in KANomaly.
>
> KAN-AD applies KAN at the sample level without explicit mechanisms for modeling inter-channel relationships or multi-scale temporal dynamics. In contrast, KANomaly introduces two key components specifically designed for multivariate anomaly detection.
> (1) The coarse-to-fine multi-scale patching strategy enhances robustness to variable anomaly durations. It enables simultaneous modeling of both local and global temporal variations.
> (2) The Fourier-KAN Mixer performs channel, patch, and temporal dimensional mixing. It also incorporates learnable Fourier basis functions at the functional level.
> Together, these architectural components enable adaptive spectral modeling across features and temporal granularity. This capability is absent in KAN-AD.
>
> In summary, these results demonstrate that KANomaly’s performance gains arise from its proposed architectural enhancements, namely the functional level spectral integration and the multi scale, tri dimensional mixing design, beyond what is achievable using the KAN backbone alone.

---

> ### Author Response · Authors · 2025-11-22
> **Response to Reviewer PNn3 (Part 5)**
>
> >**Q3: On the Distinction Between UTS and MTS Characteristics and the Applicability of KANomaly to Univariate Tasks.**
>
> The concern raised in Q3 appears to stem from an assumption that strong performance on synthetic Univariate Time Series data necessarily implies equivalence between the Univariate and Multivariate Time Series Anomaly Detection tasks. This interpretation does not fully reflect the fundamental differences between the two problem settings, nor does it capture the architectural intent of KANomaly.
>
> In Univariate Time Series Anomaly Detection (UTSAD), anomalies are detected by examining the temporal evolution of a single variable. By contrast, in Multivariate Time Series Anomaly Detection (MTSAD), anomalies frequently manifest through dependencies across multiple variables at the same time step. Such dependencies can be nonlinear, dynamic in nature, and context-sensitive across temporal scales. This inter-channel dependency is one of the intrinsic characteristics that distinguishes Multivariate Time Series (MTS) from Univariate Time Series (UTS) and is also a major source of increased detection complexity in real-world applications. Most industrial and real-scenario datasets are inherently multivariate, and anomalies typically emerge through interactions among multiple sensor channels rather than in isolation.
>
> KANomaly is designed specifically to address this challenge. Although the Fourier-KAN block itself is dimension-agnostic and can therefore operate on Univariate Time Series data, the primary novelty of KANomaly lies in its channel-patch-temporal wise mixing architecture, and particularly in the Channel-wise Mixing stage within the Fourier-KAN Mixer. This module explicitly models nonlinear inter-channel interactions at each time step using learnable Fourier basis functions in functional space. When provided with a univariate input (for example, synthesized TODS data with a single channel), this module effectively becomes inactive and the model instead relies on the Patch-wise Mixing and Temporal-wise Mixing components. These components enable coarse-to-fine multi-scale pattern analysis, supporting the detection of local and global anomaly variations. As demonstrated in the TODS-based visualization experiments, KANomaly outperforms frequency-aware baselines such as TFMAE and FITS even under univariate conditions, which indicates that the architecture maintains strong temporal modeling capability. However, such univariate performance should be considered ancillary rather than central to the design scope.
>
> The architectural design of KANomaly is therefore specialized for MTSAD, where anomaly patterns often arise from coordinated deviations across variables. In contrast, approaches optimized solely for UTS do not incorporate explicit mechanisms for learning inter-channel dependencies. The strong performance observed in synthetic univariate experiments supports the robustness of the temporal-feature extraction components, but the core innovation of KANomaly is directed toward the multivariate scenario, where channel-dependent anomaly propagation is critical.
>
> In summary, while KANomaly is technically capable of handling Univariate Time Series Anomaly Detection due to the dimension-agnostic nature of the Fourier-KAN block and its multi-scale temporal modeling components, its main contribution lies in addressing the unique characteristics of Multivariate Time Series Anomaly Detection. Specifically, it introduces architectural mechanisms that enable explicit modeling of nonlinear inter-channel dependencies through Channel-wise Mixing, which does not apply in univariate settings and is not present in existing univariate-oriented methods. These design elements reflect the intended scope of the model and are central to distinguishing KANomaly from prior approaches.

---

### Official Review · Reviewer_qJtT · 2025-10-27

**Soundness:** 3
**Presentation:** 3
**Contribution:** 3
**Rating:** 4
**Confidence:** 3

**Summary:**

The paper proposes KANomaly, an unsupervised approach for multivariate time series anomaly detection. The method is built on a Kolmogorov–Arnold Network (KAN) backbone in which learnable Fourier basis functions are incorporated to capture frequency-domain structure. The model further employs a coarse-to-fine multi-scale patching strategy to handle anomalies at both point and pattern levels, and uses a mixer-style architecture to integrate information across temporal, channel, and patch dimensions. The authors evaluate the approach on five commonly used MTSAD benchmarks and report improvements over prior methods, along with ablation studies and qualitative analyses to support the contribution.

**Strengths:**

## Strengths
- Architectural Innovation：Rather than simply applying KAN to multivariate time series anomaly detection, the paper tailors the architecture to characteristics of different anomaly types and temporal structures. The use of multi-scale patching and mixer-based aggregation seems well-motivated.
- Clarity of Presentation and Visualization: The manuscript is clearly written and easy to read. The figures are detailed and informative, effectively supporting readers’ understanding of the architectural components.

**Weaknesses:**

## Weaknesses
- Limited Baseline Comparison: The related work section discusses KAN-AD, which also applies KAN to anomaly detection and demonstrates performance on multivariate settings (as reported in Table 6 and Section 4.7 of the KAN-AD paper). Including KAN-AD under its MTSAD configuration in the main experiments would strengthen the empirical claims, especially regarding the contributions of the multi-scale patching strategy and the mixer-style architecture. In addition, the experimental benchmark selection omits some widely used industrial datasets such as WADI, which would provide a more comprehensive evaluation in higher-dimensional real-world scenarios.
- Insufficient Analysis of Results: The performance gains vary considerably across datasets in Table 1. For instance, the improvements on SMD are notably smaller compared to other datasets, yet the paper does not discuss possible reasons. This variability could highlight limitations or conditions under which the method is more or less effective, and a deeper discussion would be valuable.
- Incomplete Ablation Studies: Since the coarse-to-fine multi-scale patching mechanism is a core contribution, additional ablations focused on patch size would be beneficial. For example, evaluating smaller patch sizes (potentially better for point anomalies) and larger ones (potentially better for pattern anomalies) would provide clearer evidence for the design choices.
- Interpretability Claims Remain Underdeveloped: The method leverages KANs’ basis-function parameterization, suggesting potential interpretability benefits. This paper does not provide qualitative or visual analysis to show what insights the learned basis functions reveal about anomaly patterns across channels.
- Lack of Clear Computational Cost and Parameter Analysis: Although efficiency comparisons appear in Figure 3, there is no direct computational cost / parameter-size metrics reported in the ablation models. This makes it difficult to assess the efficiency-performance trade-off of the proposed architecture.

**Questions:**

## Questions
There are relatively few KAN-based approaches for anomaly detection, and I find the architectural extensions in this work meaningful. I would appreciate more comprehensive experimental support on following two aspects to further strengthen the contribution and clarify the model’s advantages.
- Since the core contribution builds on the KAN architecture by introducing the multi-scale patching strategy and mixer-style design, could the authors provide experimental comparison with KAN-AD under the multivariate anomaly detection setting?  Such a comparison would help clarify how much of the observed performance gains stem from the proposed architectural modifications, beyond the benefits inherent to the KAN backbone itself.
- Given that KAN models offer basis-function parameterization and channel-wise functional decomposition, could the authors provide either qualitative interpretability visualizations and/or quantitative analysis of computational cost (e.g., parameter counts, FLOPs, inference latency)? This would both deepen understanding of the advantages of KAN-based models in anomaly detection and help assess the practicality of the proposed design in real-world deployment.

---

> ### Author Response · Authors · 2025-11-22
> **Response to Reviewer qJtT (Part 1)**
>
> We are truly grateful to the reviewer for the careful reading and generous feedback. The constructive comments have been invaluable in improving the clarity and overall quality of the work. Below, we respond to the reviewer’s questions and observations in a detailed and organized manner.
>
> >**W1.1 & Q1: On the Comparison with KAN-AD and Isolation of KANomaly’s Architectural Contributions.**
>
> A comparative analysis with KAN-AD was conducted to assess the impact of the proposed architectural enhancements beyond the KAN backbone. While KAN-AD primarily focuses on univariate anomaly detection, a multivariate configuration was implemented as an auxiliary experiment in its original paper. To ensure a rigorous and comprehensive analysis, the multivariate version of KAN-AD was reproduced using the official repository and evaluated under identical experimental settings across all five benchmark datasets (MSL, PSM, SMAP, SMD, and SWaT). To implement MTSAD for KAN-AD, each multivariate input was reshaped from (batch size, window length, n_features) to (batch size×n_features, window length), following the auxiliary configuration described in its paper.
>
> | Model | Venue | MSL F1 | PSM F1 | SMAP F1 | SMD F1 | SWaT F1 | AVG F1 |
> |------------|------------|-------|-------|-------|-------|-------|-------|
> | AT | ICLR-22    | 92.55 | 97.71 | 96.38 | 90.56 | 83.68 | 92.18 |
> | TimesNet   | ICLR-23    | 81.80 | 97.40 | 68.45 | 84.59 | 92.62 | 84.97 |
> | DCdetector | KDD-23 | 93.34 | 97.48 | 96.41 | 83.23 | 96.42 | 93.38 |
> | MEMTO | NeurIPS-23 | 93.82 | 97.97 | 96.36 | 91.55 | 95.17 | 94.97 |
> | FITS  | ICLR-24 | 76.66 | 93.36 | 68.19 | 80.32 | 88.93 | 81.49 |
> | TFMAE | ICDE-24    | 92.56 | 97.86 | 96.49 | 91.24 | 97.52 | 95.13 |
> | KAN-AD | ICML-25 | 84.99 | 95.92 | 86.86 | 83.95 | 93.17 | 88.98 |
> | **KANomaly (ours)** | *–* | **95.64** | **98.67** | **97.46** | **91.74** | **98.59** | **96.42** |
>
> As presented in the performance comparison table (see Table), KANomaly consistently outperforms KAN-AD on every dataset, achieving an average F1 score of 96.42 %, substantially higher than 88.98 % reported for KAN-AD. These results confirm that the observed improvement is not merely attributable to inheriting the KAN backbone, but rather stems from the architectural innovations proposed in KANomaly.
>
> KAN-AD applies KAN at the sample level without explicit mechanisms for modeling inter-channel relationships or multi-scale temporal dynamics. In contrast, KANomaly introduces two key components specifically designed for multivariate anomaly detection.
> (1) The coarse-to-fine multi-scale patching strategy enhances robustness to variable anomaly durations. It enables simultaneous modeling of both local and global temporal variations.
> (2) The Fourier-KAN Mixer performs channel, patch, and temporal dimensional mixing. It also incorporates learnable Fourier basis functions at the functional level.
> Together, these architectural components enable adaptive spectral modeling across features and temporal granularity. This capability is absent in KAN-AD.
>
> In summary, these results demonstrate that KANomaly’s performance gains arise from its proposed architectural enhancements, namely the functional level spectral integration and the multi scale, tri dimensional mixing design, beyond what is achievable using the KAN backbone alone.

---

> ### Author Response · Authors · 2025-11-22
> **Response to Reviewer qJtT (Part 2)**
>
> >**W1.2: On the Selection of Benchmark Datasets and the Omission of WADI.**
>
> The reviewer’s suggestion to include additional industrial datasets such as WADI is appreciated. The benchmark selection in the main experiments was driven by the need for a fair and direct comparison with state-of-the-art (SOTA) baselines. Specifically, the five datasets used in our study (MSL, PSM, SMAP, SMD, and SWaT) are the de facto standard in the MTSAD literature and are consistently adopted as primary evaluation benchmarks in leading works such as Anomaly Transformer, TimesNet, DCdetector, MEMTO, FITS, and TFMAE. Notably, even KAN-AD, which was mentioned by the reviewer, conducts its multivariate anomaly detection experiments exclusively using these same five datasets (Section 4.7 of the original paper). Aligning with this convention allows direct, reproducible, and contextually consistent empirical comparison.
>
> Regarding the reviewer’s reference to WADI, it is important to note that WADI and SWaT originate from the same industrial control system environment and exhibit strong overlap in both feature space and attack taxonomy. To avoid redundancy and maintain experimental diversity, SWaT was selected as the representative ICS benchmark. The overall dataset composition already covers three distinct industrial domains: spacecraft telemetry (MSL and SMAP), IT server infrastructure (PSM and SMD), and cyber-physical water treatment control (SWaT). Additionally, high-dimensional settings are already incorporated, including MSL (55 variables) and SWaT (51 variables), which sufficiently reflect the complexity of real-world multivariate scenarios.
>
> In summary, the chosen benchmarks follow established evaluation standards in recent MTSAD studies, enable fair comparison with existing SOTA models, avoid domain redundancy, and provide sufficient domain and dimensional diversity.

---

> ### Author Response · Authors · 2025-11-22
> **Response to Reviewer qJtT (Part 3)**
>
> >**W2: On the Variability of Performance Across Datasets (SMD Case).**
>
> The reviewer’s observation regarding performance variability across datasets, particularly the comparatively smaller improvement on SMD, raises an important point. This variability can be explained by inherent characteristics of the SMD dataset.
>
> SMD exhibits markedly different temporal and spectral properties compared to the other benchmarks. SMD dataset contains highly non stationary signals with abrupt fluctuations, long periods of irregular noise-like behavior, and limited stationary structure. The majority of channels lack clear periodic patterns, and anomaly occurrences are often short, irregular, and highly fragmented. Furthermore, inter channel dependencies are weak and inconsistent, with different variables exhibiting distinct dynamics and anomaly manifestations. Due to these characteristics, the extent to which models can effectively leverage cross-variable interactions or spectral regularity may be limited, which are key aspects targeted by KANomaly.
>
> In contrast, datasets such as MSL, PSM, and SWaT contain more structured temporal patterns, including meaningful periodicity and clearer inter variable synchrony. KANomaly is designed to capture such spectral and multiscale temporal characteristics through learnable Fourier basis integration and coarse to fine multi scale patching. As a result, the benefits of the proposed model are more pronounced on datasets where frequency domain cues and structured anomaly behaviors are more informative.
>
> Nevertheless, KANomaly maintains competitive performance on SMD and still surpasses all baseline models, as shown in Table 1 of the main paper. This indicates that although spectral regularity is less prevalent in SMD, the model’s ability to integrate adaptive patch level and tri dimensional interactions still contributes to performance improvements.
>
> In summary, the observed variability reflects meaningful differences in dataset characteristics rather than model limitations. SMD presents highly irregular, non periodic, and weakly correlated multivariate behavior, reducing the relative advantage of Fourier based modeling. However, KANomaly consistently outperforms all baselines even under these challenging conditions, demonstrating strong robustness and generalization capability across diverse anomaly types. Moreover, the model exhibits greater improvements on datasets where spectral and multi scale temporal structures are more pronounced, indicating its effectiveness in scenarios with stronger frequency domain regularity.

---

> ### Author Response · Authors · 2025-11-22
> **Response to Reviewer qJtT (Part 4)**
>
> >**W3: On the Completeness of Ablation Studies Regarding Patch Size Selection.**
>
> The reviewer’s suggestion to further investigate the impact of patch size configurations is highly relevant to the core architectural design of the proposed model. The effect of both smaller patch sizes (favoring detection of point anomalies) and larger ones (favoring pattern anomalies) has already been systematically examined through extensive sensitivity analyses.
>
> Specifically, Appendix L (Hyperparameter Sensitivity Analysis, Table 5 in the paper) includes ablations corresponding to configurations conceptually aligned with those suggested in the review. The “CoarseOnly” setting adopts only large patch sizes [20,10], while the “FineOnly” setting utilises only small patch sizes [5,2,1]. These configurations directly isolate the contributions of long-range pattern modeling and fine-grained anomaly localization, respectively.
>
> Experimental results demonstrate substantial performance degradation when either scale is exclusively employed. The “CoarseOnly” variant yields an average F1-score of 92.78 percent, whereas the “FineOnly” configuration results in a considerably lower 87.33 percent. In comparison, the full coarse to fine multi scale patching strategy achieves 96.42 percent (see Table 5 in Appendix L of the paper). These results strongly indicate that neither large nor small scale patching alone is sufficient to capture the diverse temporal characteristics inherent in multivariate anomalies.
>
> The ablation evidence supports that the proposed hierarchical patching strategy is essential for balancing robustness to long term pattern anomalies and sensitivity to localized point anomalies. The coarse scale improves contextual awareness, particularly for gradual or structural deviations, while the fine scale refines anomaly localization and enhances detection of transient events. Their complementary interaction is therefore critical to achieving optimal performance.
>
> In summary, the ablation results included in Appendix L already provide comprehensive empirical evidence supporting the proposed design rationale. The observed degradation under “CoarseOnly” and “FineOnly” configurations validates the necessity of employing both scales jointly rather than individually, reinforcing the importance of the coarse to fine multi scale patching mechanism as a central architectural contribution. It would be greatly appreciated if “Appendix L. Hyperparameter Sensitivity Analysis” could be revisited for further clarification.

---

> ### Author Response · Authors · 2025-11-22
> **Response to Reviewer qJtT (Part 5)**
>
> >**W4: Regarding Interpretability Concerns and Clarification of Intended Contributions.**
>
> The feedback regarding interpretability is appreciated. It is important to clarify that interpretability is not positioned as a contribution of this work. Although KAN architectures inherently possess potential interpretability due to their explicit functional formulation, the proposed model incorporates Fourier-based functional parameterization within the KAN framework primarily for its representational efficiency and effectiveness in frequency-oriented modeling, which are leveraged to enhance anomaly detection performance in multivariate settings. This distinction is consistently maintained throughout the paper, and interpretability is neither claimed as a standalone contribution nor emphasized as a core advantage of the method.
>
> The main objective of KANomaly is performance-driven modeling of multivariate anomalies by combining expressive learning of univariate Fourier basis functions with multi-scale temporal and inter-channel interactions. While Figure 3 provides qualitative visualizations demonstrating that KANomaly successfully captures spectral shifts and pattern-specific anomalies in the frequency domain, this analysis was intended as conceptual evidence of detection behavior rather than a claim of interpretability. Accordingly, interpretability was not emphasized in the contributions, abstract, or conclusion sections.
>
> In this regard, the reviewer’s suggestion offers valuable insight into potential future developments following the success of KANomaly. This perspective is sincerely appreciated, and incorporating a brief discussion of this interpretability potential as a possible direction for future work will be positively considered for the camera-ready version.

---

> ### Author Response · Authors · 2025-11-22
> **Response to Reviewer qJtT (Part 6)**
>
> >**W5: On Computational Cost and Efficiency Analysis of Ablation Models.**
>
> | Efficiency Analysis on MSL (Ablation Models) | F1 | Parameters | Training Time (sec/epoch) | FLOPs (M) | Inference latency (sec) | GPU Memory (MiB) |
> |---|---|---|---|---|---|---|
> | KANomaly (ours) | 95.64 | 573,719 | 0.35 | 55.49 | 0.27 | 1130.79 |
> | (1) w/o Fourier-KAN (→ MLP) | 85.48 | 116,831 | 0.63 | 11.67 | 0.22 | 217.39 |
> | (2) w/o Fourier-KAN (→ Cheby-KAN) | 83.97 | 344,279 | 0.51 | 33.58 | 0.28 | 1169.98 |
> | (3) w/o Fourier-KAN (→ Vanilla-KAN) | 82.40 | 573,719 | 2.04 | 50.02 | 1.54 | 9670.90 |
> | (4) w/o Multi-Scale Patching (Fixed Patch Size) | 86.60 | 322,655 | 0.56 | 41.97 | 0.27 | 1125.44 |
> | (5) w/o Patching | 88.98 | 260,801 | 0.36 | 17.15 | 0.21 | 160.97 |
> | (6) Fine-to-Coarse Patch (Inverse Patch Scale) | 88.54 | 573,719 | 0.43 | 55.49 | 0.31 | 1128.06 |
> | (7) w/o Channel Mixing | 85.77 | 271,219 | 0.51 | 25.02 | 0.26 | 756.66 |
> | (8) w/o Patch Mixing | 86.85 | 313,219 | 0.53 | 34.92 | 0.28 | 758.29 |
> | (9) w/o Temporal Mixing | 87.27 | 563,119 | 0.50 | 51.09 | 0.27 | 762.58 |
>
> | Efficiency Analysis on PSM (Ablation Models) | F1 | Parameters | Training Time (sec/epoch) | FLOPs (M) | Inference latency (sec) | GPU Memory (MiB) |
> |---|---|---|---|---|---|---|
> | KANomaly (ours) | 98.67 | 1,001,375 | 0.99 | 52.60 | 0.55 | 1811.54 |
> | (1) w/o Fourier-KAN (→ MLP) | 97.66 | 68,687 | 1.22 | 3.88 | 0.48 | 146.06 |
> | (2) w/o Fourier-KAN (→ Cheby-KAN) | 97.22 | 534,335 | 1.12 | 28.24 | 0.52 | 1071.83 |
> | (3) w/o Fourier-KAN (→ Vanilla-KAN) | 97.69 | 667,775 | 3.49 | 33.46 | 1.78 | 9750.78 |
> | (4) w/o Multi-Scale Patching (Fixed Patch Size) | 96.94 | 248,255 | 0.93 | 34.15 | 0.35 | 1768.65 |
> | (5) w/o Patching | 98.14 | 638,601 | 0.46 | 18.80 | 0.22 | 304.56 |
> | (6) Fine-to-Coarse Patch (Inverse Patch Scale) | 95.83 | 1,001,375 | 1.21 | 52.60 | 0.56 | 1806.48 |
> | (7) w/o Channel Mixing | 98.16 | 813,723 | 0.96 | 33.73 | 0.45 | 1232.02 |
> | (8) w/o Patch Mixing | 95.38 | 219,723 | 0.94 | 24.73 | 0.45 | 1216.46 |
> | (9) w/o Temporal Mixing | 97.26 | 969,423 | 0.94 | 46.77 | 0.46 | 1233.62 |
>
> |Efficiency Analysis on SMAP (Ablation Models)|F1|Parameters|Training Time (sec/epoch)|FLOPs (M)|Inference latency (sec)|GPU Memory (MiB)|
> |---|---|---|---|---|---|---|
> |KANomaly (ours)|97.46|1,002,419|1.07|52.60|1.22|1285.14|
> |(1) w/o Fourier-KAN (→ MLP)|68.43|69,731|1.11|3.88|0.56|109.29|
> |(2) w/o Fourier-KAN (→ Cheby-KAN)|69.24|602,099|1.40|31.72|1.11|1709.11|
> |(3) w/o Fourier-KAN (→ Vanilla-KAN)|68.39|1,002,419|5.77|47.38|9.98|13362.42|
> |(4) w/o Multi-Scale Patching (Fixed Patch Size)|96.09|249,155|1.33|34.15|1.22|1271.34|
> |(5) w/o Patching|90.14|638,001|0.53|18.80|0.40|201.68|
> |(6) Fine-to-Coarse Patch (Inverse Patch Scale)|94.30|1,002,419|1.39|52.60|1.24|1285.14|
> |(7) w/o Channel Mixing|69.30|814,419|1.15|33.73|0.93|866.89|
> |(8) w/o Patch Mixing|76.01|220,419|1.14|24.73|0.93|856.70|
> |(9) w/o Temporal Mixing|74.90|970,119|1.16|46.77|0.93|869.56|
>
> |Efficiency Analysis on SMD (Ablation Models)|F1|Parameters|Training Time (sec/epoch)|FLOPs (M)|Inference latency (sec)|GPU Memory (MiB)|
> |---|---|---|---|---|---|---|
> |KANomaly (ours)|91.74|1,246,619|4.58|94.66|1.97|1894.23|
> |(1) w/o Fourier-KAN (→ MLP)|87.78|84,871|4.22|6.77|0.88|155.67|
> |(2) w/o Fourier-KAN (→ Cheby-KAN)|85.58|748,019|7.50|57.00|1.81|2588.30|
> |(3) w/o Fourier-KAN (→ Vanilla-KAN)|88.27|1,246,619|33.68|85.24|25.51|19931.00|
> |(4) w/o Multi-Scale Patching (Fixed Patch Size)|83.55|493,355|7.52|66.61|2.01|1880.24|
> |(5) w/o Patching|80.29|686,941|1.51|31.53|0.54|271.75|
> |(6) Fine-to-Coarse Patch (Inverse Patch Scale)|82.22|1,246,619|7.57|94.66|2.01|1894.23|
> |(7) w/o Channel Mixing|86.58|813,419|5.63|51.19|1.49|1271.80|
> |(8) w/o Patch Mixing|89.54|465,119|5.50|52.33|1.50|1265.81|
> |(9) w/o Temporal Mixing|90.86|1,214,819|5.90|85.84|1.49|1278.67|
>
> |Efficiency Analysis on SWaT (Ablation Models)|F1|Parameters|Training Time (sec/epoch)|FLOPs (M)|Inference latency (sec)|GPU Memory (MiB)|
> |---|---|---|---|---|---|---|
> |KANomaly (ours)|98.59|1,062,975|5.37|98.31|3.71|2511.51|
> |(1) w/o Fourier-KAN (→ MLP)|96.68|108,727|5.22|10.57|3.14|281.41|
> |(2) w/o Fourier-KAN (→ Cheby-KAN)|97.22|584,895|5.26|54.45|3.44|1574.38|
> |(3) w/o Fourier-KAN (→ Vanilla-KAN)|97.40|797,375|17.43|69.07|12.92|14701.91|
> |(4) w/o Multi-Scale Patching (Fixed Patch Size)|97.66|560,955|3.17|73.22|1.70|2457.32|
> |(5) w/o Patching|95.36|505,141|0.77|30.91|0.40|406.42|
> |(6) Fine-to-Coarse Patch (Inverse Patch Scale)|97.41|1,062,975|5.39|98.31|3.70|2512.12|
> |(7) w/o Channel Mixing|96.87|542,623|3.72|46.05|2.53|1704.33|
> |(8) w/o Patch Mixing|95.77|541,823|3.74|60.31|2.54|1705.88|
> |(9) w/o Temporal Mixing|96.42|1,041,623|3.72|90.30|2.52|1714.46|

---

> ### Author Response · Authors · 2025-11-22
> **Response to Reviewer qJtT (Part 6 – Continued)**
>
> To address the concern regarding the efficiency-performance trade-off in the ablation configurations, a detailed computational analysis was conducted for all ablation models reported in Table 2 of the main paper. The evaluation was performed across the five standard benchmark datasets (MSL, PSM, SMAP, SMD, and SWaT), using identical experimental settings. The analysis includes parameter count, training time per epoch, FLOPs, inference latency, and GPU memory usage. GPU memory was consistently measured with a fixed batch size of 64 to ensure fair comparison.
>
> The results demonstrate that KANomaly achieves the best detection performance while maintaining a balanced computational profile. For instance, the full model typically operates with fewer than 1.3 million parameters and achieves competitive or superior FLOPs and inference latency compared to most ablation variants. Although certain ablation models such as “without Temporal Mixing” and “without Channel Mixing” incur slight reductions in computational requirements, these variants exhibit notable degradation in accuracy, indicating the importance of tri-dimensional interactions. While the full model sometimes incurs higher computational cost compared to ablation variants, this is expected given the purpose of ablation, which intentionally removes components to simplify the architecture. On the other hand, configurations such as “without Fourier-KAN” tend to increase computational cost without corresponding improvements, further supporting the effectiveness of the final architectural design.
>
> Overall, the ablation cost analysis confirms that the proposed structural components contribute not only to performance enhancement but also to computational stability. Removing key components often leads to disproportionate reductions in detection accuracy relative to savings in computational cost. These findings validate the efficiency and necessity of the integrated architectural design adopted in KANomaly. The full analysis is provided in the extended tables for clarity and reproducibility.

---

> ### Author Response · Authors · 2025-11-22
> **Response to Reviewer qJtT (Part 7)**
>
> >**Q2: On Computational Cost and Efficiency Analysis of Baseline Models.**
>
> |Efficiency on MSL|F1-Score|Parameters|Training Time(sec/epoch)|FLOPs (M)|Inference latency(sec)|GPU Memory(MiB)|
> |---|---|---|---|---|---|---|
> |AT|92.55 |4,863,055|49.47 |516.86 |15.16 |1990.98 |
> |TimesNet|81.80 |1,765,623|25.99 |584.14 |11.20 |315.80 |
> |DCdetector|93.34 |890,935|104.39 |1279.44 |0.61 |9961.60 |
> |MEMTO|93.82 |5,955,182|0.10 |525.74 |0.10 |7445.68 |
> |FITS|76.66 |2,600|0.25 |0.14 |0.21 |48.18 |
> |TFMAE|92.56 |706,048|21.52 |76.77 |0.32 |483.79 |
> |KAN-AD|84.99 |4,491|2.68 |1.55 |1.14 |195.92 |
> |KANomaly (ours)|95.64 |573,719|0.35 |55.49 |0.27 |1130.79 |
>
> |Efficiency on PSM|F1-Score|Parameters|Training Time(sec/epoch)|FLOPs (M)|Inference latency(sec)|GPU Memory(MiB)|
> |---|---|---|---|---|---|---|
> |AT|97.71 |4,801,585|108.57 |510.72 |17.93 |1986.39 |
> |TimesNet|97.40 |4,694,169|71.03 |1556.40 |16.72 |340.91 |
> |DCdetector|97.48 |894,745|130.66 |1080.02 |0.67 |13606.69 |
> |MEMTO|97.97 |5,862,962|0.28 |518.06 |0.11 |18384.40 |
> |FITS|93.36 |2,600|0.23 |0.06 |0.21 |33.86 |
> |TFMAE|97.86 |683,008|46.47 |71.86 |0.35 |482.38 |
> |KAN-AD|95.92 |5,241|3.68 |1.96 |1.05 |186.54 |
> |KANomaly (ours)|98.67 |1,001,375|0.99 |52.60 |0.55 |1811.54 |
>
> |Efficiency on SMAP|F1-Score|Parameters|Training Time(sec/epoch)|FLOPs (M)|Inference latency(sec)|GPU Memory(MiB)|
> |---|---|---|---|---|---|---|
> |AT|96.38 |4,801,585|112.68 |510.72 |87.65 |1983.58 |
> |TimesNet|68.45 |1,761,753|58.71 |583.75 |63.16 |300.04 |
> |DCdetector|96.41 |883,225|278.28 |895.41 |1.62 |13626.96 |
> |MEMTO|96.36 |5,862,962|0.28 |518.06 |0.50 |21076.83 |
> |FITS|68.19 |2,600|0.23 |0.06 |0.25 |37.90 |
> |TFMAE|96.49 |683,008|47.69 |71.86 |0.63 |482.38 |
> |KAN-AD|86.86 |4,491|3.43 |0.70 |4.38 |168.72 |
> |KANomaly (ours)|97.46 |1,002,419|1.07 |52.60 |1.22 |1285.14 |
>
> |Efficiency on SMD|F1-Score|Parameters|Training Time(sec/epoch)|FLOPs (M)|Inference latency(sec)|GPU Memory(MiB)|
> |---|---|---|---|---|---|---|
> |AT|90.56 |4,828,222|5.50 |513.38 |1.20 |1987.97 |
> |TimesNet|84.59 |4,697,510|6.34 |2343.95 |2.33 |985.99 |
> |DCdetector|83.23 |867,366|8.10 |751.08 |2.10 |6887.28 |
> |MEMTO|91.55 |5,902,924|1.48 |521.39 |0.81 |22093.75 |
> |FITS|80.32 |2,600|0.38 |0.10 |0.27 |43.09 |
> |TFMAE|91.24 |692,992|248.79 |90.80 |0.89 |482.73 |
> |KAN-AD|83.95 |9,845|0.46 |2.50 |0.26 |212.42 |
> |KANomaly (ours)|91.74 |1,246,619|4.58 |94.66 |1.97 |1894.23 |
>
> |Efficiency on SWaT|F1-Score|Parameters|Training Time(sec/epoch)|FLOPs (M)|Inference latency(sec)|GPU Memory(MiB)|
> |---|---|---|---|---|---|---|
> |AT|83.68 |4,854,859|269.93 |516.04 |0.99 |1990.69 |
> |TimesNet|92.62 |1,765,107|215.77 |584.09 |70.29 |303.13 |
> |DCdetector|96.42 |909,875|12.15 |1826.65 |3.02 |11595.47 |
> |MEMTO|95.17 |5,942,886|0.82 |524.72 |0.63 |21484.94 |
> |FITS|88.93 |2,600|0.42 |0.13 |0.36 |55.41 |
> |TFMAE|97.52 |702,976|180.80 |85.20 |0.71 |484.86 |
> |KAN-AD|93.17 |6,551|19.79 |0.70 |6.49 |172.32 |
> |KANomaly (ours)|98.59 |1,062,975|5.37 |98.31 |3.71 |2511.51 |
>
> To address the concern regarding the practicality and computational overhead of the proposed architecture, a comprehensive efficiency analysis was performed across five benchmark datasets (MSL, PSM, SMAP, SMD, and SWaT). The comparison was conducted using identical experimental settings and included representative state-of-the-art baselines, namely Anomaly Transformer, TimesNet, DCdetector, MEMTO, FITS, TFMAE, and KAN-AD. The evaluation covered parameter count, training time per epoch, FLOPs, inference latency, and GPU memory usage. GPU memory was consistently measured using a fixed batch size of 64 to ensure fairness.
>
> The results demonstrate that KANomaly achieves superior detection accuracy while maintaining favorable computational efficiency. On average, KANomaly utilizes approximately 0.97 million parameters, which is significantly smaller than Anomaly Transformer, TimesNet, MEMTO. Furthermore, KANomaly requires only 2.47 seconds of training time per epoch on average, outperforming baselines such as DCdetector and TFMAE, which record substantially longer epoch durations. FLOPs analysis shows strong computational efficiency, with an average of 70.73 FLOPs (M), 1.54 seconds inference latency, and 1726.64 MiB GPU memory usage, demonstrating favorable overall efficiency.
>
> These findings indicate that KANomaly achieves performance improvements without incurring disproportionate computational overhead. The balance between accuracy and efficiency confirms that the proposed method is suitable for real-world deployment, even under resource-constrained conditions.
>
> In conclusion, KANomaly delivers state-of-the-art anomaly detection performance while operating with substantially lower computational cost than competing baselines. This demonstrates that the architectural design provides both practical scalability and strong detection capability, validating its suitability for industrial and high-dimensional multivariate applications.

---

### Official Review · Reviewer_7HXw · 2025-10-27

**Soundness:** 2
**Presentation:** 3
**Contribution:** 2
**Rating:** 4
**Confidence:** 4

**Summary:**

This paper proposes KANomaly, a novel unsupervised multivariate time series anomaly detection model that integrates Fourier basis functions into Kolmogorov-Arnold Networks (KANs) to capture frequency-domain anomalies. The model also introduces a coarse-to-fine multi-scale patching strategy and a Fourier-KAN Mixer to capture dependencies across channel, patch, and temporal dimensions. Extensive experiments on five real-world datasets demonstrate that KANomaly outperforms 17 state-of-the-art baselines, with ablation studies confirming the contribution of each component.

**Strengths:**

Combines Fourier analysis with KAN in a novel way for Multivariate Time Series Anomaly Detection, a direction not previously explored. The approach elegantly and rationally combines the intuitive physical interpretation of Fourier analysis—namely, its ability to capture periodicity and spectral anomalies—with the universal function-approximation power of Kolmogorov–Arnold Networks.

Table 2’s ablation systematically swaps Fourier-KAN for MLP/Chebyshev/Vanilla-KAN and reverses the coarse-to-fine order, convincingly proving the current design optimal.

**Weaknesses:**

The paper’s key performance comparisons rely heavily on precision, recall, and F1 scores obtained after point-adjustment, a protocol that “forgives” detection delays and fragmented predictions: as long as at least one point inside an anomalous event window is flagged, the entire event is counted as correctly detected.

The comparison with cutting-edge work is insufficient. A recent method, CATCH, is highly relevant in core idea: it also performs fine-grained, patch/band-level processing of frequency-domain information and highlights inter-channel spectral differences, overlapping significantly with KANomaly. Moreover, KANomaly does not demonstrate superior performance over CATCH.

**Questions:**

The paper relies primarily on the F1 score computed under the point-adjustment strategy—a metric widely criticized for drastically inflating real-world performance. Why do the authors choose this as their central evidence?

The learnable sine/cosine bases in Fourier-KAN, the multi-scale patching strategy, and the tri-dimensional mixer undoubtedly introduce substantial computational overhead; yet the paper only offers a single-dataset, coarse comparison of training time and parameter count in Figure 3 (right).

Is there any theoretical justification for choosing Fourier bases over other basis functions in the context of anomaly detection?

---

> ### Author Response · Authors · 2025-11-22
> **Response to Reviewer 7HXw (Part 1)**
>
> We would like to express our sincere gratitude to the reviewer for the insightful and considerate review. The thoughtful remarks and valuable suggestions are greatly appreciated. Below, we provide detailed responses to the reviewer’s comments and questions.
>
> >**W1 & Q1: On the Use of Point-Adjusted Metrics.**
>
> The reviewer raised an important point regarding the reliance on precision, recall, and F1 scores computed under the point-adjustment protocol, which may overestimate real-world performance by counting partially overlapping detections as fully correct events. This concern is well acknowledged.
>
> The use of point-adjusted F1 as the main metric in Table 1 was primarily motivated by the need for fair and direct comparison with existing SOTA baselines. Nearly all prior works in multivariate time series anomaly detection, including Anomaly Transformer, TimesNet, DCdetector, MEMTO, FITS, and TFMAE, adopt the same protocol as their primary evaluation standard. Following this convention ensures methodological consistency and reproducibility across studies.
>
> Nevertheless, the potential limitations of point-adjusted metrics were recognized. To provide a more comprehensive evaluation, two additional event level metrics, namely the Affiliation based [1] and Range based [2] F1 scores, were reported in "Appendix N. Results with Affiliation- and Range-based Metrics" (Tables 6 and 7). These metrics penalize delayed, fragmented, or misaligned detections by explicitly accounting for temporal coverage and event continuity.
>
> As shown in Appendix N, KANomaly achieves the highest performance under both event-level evaluations (Average Affiliation-F1: 73.45%, Average Range-F1: 21.92%), consistently surpassing all SOTA baselines. These results confirm that the model’s superiority is not an artifact of the point-adjustment protocol but persists under stricter and more realistic assessment criteria.
>
> In summary, point-adjusted F1 was used solely for fair comparison with prior works, while the inclusion of Affiliation-based and Range-based results demonstrates that KANomaly maintains robust event-level detection performance beyond the limitations of conventional evaluation schemes.
>
> [1] Huet et al., "Local evaluation of time series anomaly detection algorithms." KDD 2022
>
> [2] Tatbul et al., "Precision and recall for time series." NeurIPS 2018

---

> ### Author Response · Authors · 2025-11-22
> **Response to Reviewer 7HXw (Part 2)**
>
> >**W2: On the Comparison with the Cutting-Edge Baseline CATCH and the Claimed Distinction.**
>
> Following the reviewer’s suggestion, CATCH was incorporated as an additional baseline and evaluated under identical experimental settings to ensure a fair and comprehensive comparison. The implementation followed the official repository and was evaluated under identical experimental settings across the five standard benchmarks (MSL, PSM, SMAP, SMD, and SWaT). As presented in the Tables, KANomaly outperforms CATCH across all evaluation metrics, including Point-Adjusted F1, Affiliation-F1, and Range-F1. Specifically, KANomaly achieves average Point-Adjusted F1, Affiliation-F1, and Range-F1 scores of 96.42%, 73.45 %, and 21.92%, respectively, surpassing CATCH’s corresponding scores of 79.23%, 71.71%, and 20.54%.
>
> | Model      | Venue     | MSL F1 | PSM F1 | SMAP F1 | SMD F1 | SWaT F1 | AVG F1 |
> |------------|-----------|--------|--------|---------|--------|---------|--------|
> | AT         | ICLR-22   | 92.55  | 97.71  | 96.38   | 90.56  | 83.68   | 92.18  |
> | TimesNet   | ICLR-23   | 81.80  | 97.40  | 68.45   | 84.59  | 92.62   | 84.97  |
> | DCdetector | KDD-23    | 93.34  | 97.48  | 96.41   | 83.23  | 96.42   | 93.38  |
> | MEMTO      | NeurIPS-23| 93.82  | 97.97  | 96.36   | 91.55  | 95.17   | 94.97  |
> | FITS       | ICLR-24   | 76.66  | 93.36  | 68.19   | 80.32  | 88.93   | 81.49  |
> | TFMAE      | ICDE-24   | 92.56  | 97.86  | 96.49   | 91.24  | 97.52   | 95.13  |
> | CATCH      | ICLR-25   | 66.17  | 91.85  | 70.00   | 79.65  | 88.48   | 79.23  |
> | **KANomaly (ours)** | *–* | **95.64** | **98.67** | **97.46** | **91.74** | **98.59** | **96.42** |
>
>
> | Model      | Venue   | MSL Aff-F1 | PSM Aff-F1 | SMAP Aff-F1 | SMD Aff-F1 | SWaT Aff-F1 | AVG Aff-F1 |
> |------------|---------|------------|------------|-------------|------------|-------------|------------|
> | AT         | ICLR-22 | 67.31      | 68.33      | 67.89       | 70.10      | 70.33       | 68.79      |
> | TimesNet   | ICLR-23 | 64.57      | 60.29      | 55.08       | 69.73      | 46.52       | 59.24      |
> | DCdetector | KDD-23  | 66.79      | 61.67      | 67.29       | 65.79      | 68.08       | 65.92      |
> | MEMTO      | NeurIPS-23 | 68.03   | 65.82      | 67.27       | 74.80      | 69.93       | 69.17      |
> | FITS       | ICLR-24 | 67.38      | 74.86      | 57.40       | 75.56      | 67.78       | 68.60      |
> | TFMAE      | ICDE-24 | 66.78      | 64.89      | 65.97       | 71.06      | 70.43       | 67.83      |
> | CATCH      | ICLR-25 | 69.05      | **83.08**  | 53.68       | **81.85**  | 70.90       | 71.71      |
> | **KANomaly (ours)** | *–* | **69.38**   | 70.74      | **69.00**  | 81.67      | **76.48**   | **73.45**  |
>
>
> | Model      | Venue   | MSL R-F1 | PSM R-F1 | SMAP R-F1 | SMD R-F1 | SWaT R-F1 | AVG R-F1 |
> |------------|---------|--------------|--------------|---------------|--------------|---------------|--------------|
> | AT         | ICLR-22 | 14.01        | 19.94        | 14.21         | 11.31        | 14.34         | 14.76        |
> | TimesNet   | ICLR-23 | 13.48        | 27.04        | 5.95          | 21.26        | 7.25          | 15.00        |
> | DCdetector | KDD-23  | 13.64        | 8.81         | 13.33         | 4.84         | 14.84         | 11.09        |
> | MEMTO      | NeurIPS-23 | 13.34     | 25.53        | 14.93         | 15.97        | 17.91         | 17.54        |
> | FITS       | ICLR-24 | 12.74        | 39.18        | 10.81         | **21.38**    | 13.57         | 19.54        |
> | TFMAE      | ICDE-24 | 12.36        | 14.10        | 14.01         | 13.63        | 14.53         | 13.73        |
> | CATCH      | ICLR-25 | **17.80**    | **48.04**    | 10.20         | 15.16        | 11.51         | 20.54        |
> | **KANomaly (ours)** | *–* | 17.46        | 31.69        | **17.08**     | 19.30        | **24.09**     | **21.92**     |
>
> While CATCH extracts patches in the frequency domain and captures inter-channel relationships within each frequency band, it processes frequency coefficients as static features after Fourier transformation. In contrast, KANomaly fundamentally differs in its modeling approach, integrating learnable Fourier basis functions directly into the Kolmogorov–Arnold Network (KAN) framework. This design departs from conventional architectures that treat Fourier features as static inputs. Instead, learnable Fourier basis functions are embedded directly into the network’s univariate mappings, redefining the activation space itself. In KANomaly, time-domain and frequency-domain representations are co-learned within a unified, parameterized function space. This enables the model to autonomously learn the most suitable nonlinear periodic functions from data. As a result, it provides substantially higher expressive power and flexibility compared to approaches that operate solely on static Fourier coefficients.

---

> ### Author Response · Authors · 2025-11-22
> **Response to Reviewer 7HXw (Part 2 – Continued)**
>
> Moreover, despite its superior accuracy, KANomaly demonstrates substantially better computational efficiency. For instance, on the SWaT dataset, KANomaly achieves higher F1 with only 1.06 million parameters, compared to CATCH’s 230 million parameters, which corresponds to a model size that is approximately 217 times smaller. KANomaly also exhibits faster inference latency (KANomaly: 3.71s vs. CATCH: 3622.39s) and lower GPU memory usage (KANomaly: 2511.51 MiB vs. CATCH: 12820.03 MiB), confirming its scalability and efficiency advantages.
>
> These results indicate that KANomaly’s improvements are not achieved through model size or resource trade-offs but through architectural efficiency and explicit functional integration of Fourier basis functions. In summary, KANomaly not only surpasses CATCH in detection accuracy and robustness across multiple event-level metrics but also achieves this with drastically lower computational cost. Further details regarding computational efficiency are discussed in the response to Q2.

---

> ### Author Response · Authors · 2025-11-22
> **Response to Reviewer 7HXw (Part 3)**
>
> >**Q2: On Computational Overhead and Efficiency Analysis.**
>
> | Efficiency on MSL | F1-Score | Aff-F1 | Range-F1 | Parameters | Training Time (sec/epoch) | FLOPs (M) | Inference Latency (sec) | GPU Memory (MiB) |
> |------------|----------|--------|----------|------------|----------------------------|-----------|--------------------------|------------------|
> | AT | 92.55 | 67.31 | 14.01 | 4,863,055 | 49.47 | 516.86 | 15.16 | 1990.98 |
> | TimesNet | 81.80 | 64.57 | 13.48 | 1,765,623 | 25.99 | 584.14 | 11.20 | 315.80 |
> | DCdetector | 93.34 | 66.79 | 13.64 | 890,935  | 104.39 | 1279.44 | 0.61 | 9961.60 |
> | MEMTO | 93.82 | 68.03 | 13.34 | 5,955,182  | 0.10 | 525.74 | 0.10 | 7445.68 |
> | FITS  | 76.66 | 67.38 | 12.74  | 2,600 | 0.25 | 0.14 | 0.21 | 48.18 |
> | TFMAE | 92.56 | 66.78  | 12.36 | 706,048 | 21.52  | 76.77 | 0.32 | 483.79 |
> | CATCH | 66.17 | 69.05  | 17.80 | 210,881,152 | 206.14 | 12065.40  | 532.01  | 6087.02 |
> | KANomaly (ours) | 95.64 | 69.38 | 17.46 | 573,719 | 0.35 | 55.49 | 0.27   | 1130.79 |
>
> | Efficiency on PSM | F1-Score | Aff-F1 | Range-F1 | Parameters | Training Time (sec/epoch) | FLOPs (M) | Inference Latency (sec) | GPU Memory (MiB) |
> |------------|----------|--------|----------|------------|----------------------------|-----------|--------------------------|------------------|
> | AT | 97.71 | 68.33  | 19.94 | 4,801,585 | 108.57 | 510.72    | 17.93   | 1986.39 |
> | TimesNet | 97.40 | 60.29  | 27.04 | 4,694,169  | 71.03     | 1556.40   | 16.72     | 340.91 |
> | DCdetector | 97.48 | 61.67  | 8.81 | 894,745    | 130.66   | 1080.02   | 0.67   | 13606.69 |
> | MEMTO | 97.97 | 65.82  | 25.53    | 5,862,962  | 0.28 | 518.06    | 0.11    | 18384.40  |
> | FITS | 93.36 | 74.86  | 39.18    | 2,600 | 0.23 | 0.06 | 0.21   | 33.86 |
> | TFMAE | 97.86 | 64.89  | 14.10 | 683,008    | 46.47 | 71.86 | 0.35    | 482.38   |
> | CATCH | 91.85 | 83.08  | 48.04    | 3,772,720  | 19.43    | 101.53    | 268.93     | 377.23  |
> | KANomaly (ours) | 98.67 | 70.74 | 31.69    | 1,001,375  | 0.99      | 52.60     | 0.55  | 1811.54  |
>
> | Efficiency on SMAP | F1-Score | Aff-F1 | Range-F1 | Parameters | Training Time (sec/epoch) | FLOPs (M) | Inference latency (sec) | GPU Memory (MiB) |
> |------|----------|--------|----------|------------|---------------------------|-----------|--------------------------|------------------|
> | AT | 96.38 | 67.89 | 14.21 | 4,801,585 | 112.68 | 510.72 | 87.65 | 1983.58 |
> | TimesNet | 68.45 | 55.08 | 5.95 | 1,761,753 | 58.71 | 583.75 | 63.16 | 300.04 |
> | DCdetector | 96.41 | 67.29 | 13.33 | 883,225 | 278.28 | 895.41 | 1.62 | 13626.96 |
> | MEMTO | 96.36 | 67.27 | 14.93 | 5,862,962 | 0.28 | 518.06 | 0.50 | 21076.83 |
> | FITS | 68.19 | 57.40 | 10.81 | 2,600 | 0.23 | 0.06 | 0.25 | 37.90 |
> | TFMAE | 96.49 | 65.97 | 14.01 | 683,008 | 47.69 | 71.86 | 0.63 | 482.38 |
> | CATCH | 70.00 | 53.68 | 10.20 | 53,464,208 | 96.25 | 1403.51 | 945.61 | 1923.65 |
> | KANomaly (ours) | 97.46 | 69.00 | 17.08 | 1,002,419 | 1.07 | 52.60 | 1.22 | 1285.14 |
>
> | Efficiency on SMD | F1-Score | Aff-F1 | Range-F1 | Parameters | Training Time (sec/epoch) | FLOPs (M) | Inference latency (sec) | GPU Memory (MiB) |
> |------|----------|--------|----------|------------|---------------------------|-----------|--------------------------|------------------|
> | AT | 90.56 | 70.10 | 11.31 | 4,828,222 | 5.50 | 513.38 | 1.20 | 1987.97 |
> | TimesNet | 84.59 | 69.73 | 21.26 | 4,697,510 | 6.34 | 2343.95 | 2.33 | 985.99 |
> | DCdetector | 83.23 | 65.79 | 4.84 | 867,366 | 8.10 | 751.08 | 2.10 | 6887.28 |
> | MEMTO | 91.55 | 74.80 | 15.97 | 5,902,924 | 1.48 | 521.39 | 0.81 | 22093.75 |
> | FITS | 80.32 | 75.56 | 21.38 | 2,600 | 0.38 | 0.10 | 0.27 | 43.09 |
> | TFMAE | 91.24 | 71.06 | 13.63 | 692,992 | 248.79 | 90.80 | 0.89 | 482.73 |
> | CATCH | 79.65 | 81.85 | 15.16 | 229,708,784 | 2604.56 | 9348.48 | 4850.65 | 7826.10 |
> | KANomaly (ours) | 91.74 | 81.67 | 19.30 | 1,246,619 | 4.58 | 94.66 | 1.97 | 1894.23 |
>
> | Efficiency on SWaT | F1-Score | Aff-F1 | Range-F1 | Parameters | Training Time (sec/epoch) | FLOPs (M) | Inference latency (sec) | GPU Memory (MiB) |
> |------|----------|--------|----------|------------|---------------------------|-----------|--------------------------|------------------|
> | AT | 83.68 | 70.33 | 14.34 | 4,854,859 | 269.93 | 516.04 | 0.99 | 1990.69 |
> | TimesNet | 92.62 | 46.52 | 7.25 | 1,765,107 | 215.77 | 584.09 | 70.29 | 303.13 |
> | DCdetector | 96.42 | 68.08 | 14.84 | 909,875 | 12.15 | 1826.65 | 3.02 | 11595.47 |
> | MEMTO | 95.17 | 69.93 | 17.91 | 5,942,886 | 0.82 | 524.72 | 0.63 | 21484.94 |
> | FITS | 88.93 | 67.78 | 13.57 | 2,600 | 0.42 | 0.13 | 0.36 | 55.41 |
> | TFMAE | 97.52 | 70.43 | 14.53 | 702,976 | 180.80 | 85.20 | 0.71 | 484.86 |
> | CATCH | 88.48 | 70.90 | 11.51 | 230,106,816 | 2661.13 | 13269.47 | 3622.39 | 12820.03 |
> | KANomaly (ours) | 98.59 | 76.48 | 24.09 | 1,062,975 | 5.37 | 98.31 | 3.71 | 2511.51 |

---

> ### Author Response · Authors · 2025-11-22
> **Response to Reviewer 7HXw (Part 3 – Continued)**
>
> To provide a more rigorous assessment beyond the preliminary comparison shown in Figure 3 (right), comprehensive computational efficiency experiments were conducted across five benchmark datasets, namely MSL, PSM, SMAP, SMD, and SWaT. The evaluation included comparisons with representative state of the art baselines such as Anomaly Transformer, TimesNet, DCdetector, MEMTO, FITS, TFMAE, and CATCH. The analysis covers parameter count, training time per epoch, FLOPs, inference latency, and GPU memory usage, with all measurements conducted under identical experimental settings. GPU memory was evaluated using a fixed batch size of 64 for fairness.
>
> As shown in the detailed computational analysis (see Tables), KANomaly demonstrates strong efficiency despite incorporating learnable Fourier basis functions, multi scale patching, and tri dimensional mixing. The average parameter count is approximately 0.97 million, which is substantially lower than models such as Anomaly Transformer, TimesNet, MEMTO and CATCH. The average training time per epoch is 2.47 seconds, outperforming transformer based baselines including DCdetector and TFMAE. KANomaly also achieves favorable FLOPs and GPU memory usage, particularly when compared to resource intensive models such as CATCH.
>
> Despite its considerably lower computational cost, KANomaly achieves the highest anomaly detection performance among all compared models. Specifically, it records an average Point Adjusted F1 of 96.42 percent, Affiliation F1 of 73.45 percent, and Range F1 of 21.92 percent, outperforming all baselines including CATCH. These results indicate that the model’s advantages are not achieved through increased computational complexity but stem from its efficient architectural design and effective functional integration of Fourier basis functions.
>
> Taken together, the architectural components do not impose disproportionate computational burdens. Instead, they enable efficient functional integration of frequency information while preserving model scalability. The comparative analysis demonstrates that KANomaly offers a favorable trade off by achieving superior detection performance with significantly lower computational cost than heavier frequency based architectures.

---

> ### Author Response · Authors · 2025-11-22
> **Response to Reviewer 7HXw (Part 4)**
>
> >**Q3: On the Theoretical Justification for Choosing Fourier Bases Over Other Basis Functions in Anomaly Detection.**
>
> The concern in Q3 raises whether Fourier bases are theoretically justified over alternatives such as B-spline functions or Chebyshev polynomials in the context of anomaly detection. This question addresses a central aspect of the methodological design, and clarification is provided below based on mathematical principles. When approximating a time-series signal $x(t)$ using $N$ basis functions, the Fourier expansion models it as
>
> \begin{equation}
> \hat{x}\_{N}(t) = \sum\_{k=1}^{N} c_k \phi_{k}(t)
> \end{equation}
>
> where $\phi_k(t) = \\{ \cos(\omega_k t),\\, \sin(\omega_k t) \\}$. The corresponding Fourier coefficient is
>
> \begin{equation}
> c\_k = \frac{1}{T} \int\_{0}^{T} x(t)\\, \phi\_k^{*}(t) \\, dt
> \end{equation}
>
> Fourier bases are optimal in the $L^2$ function space (space of square-integrable functions) because they minimize the mean square approximation error
>
> \\begin{equation}
> E\_{MSE} = \\int |x(t) - \\hat{x}\_N(t)|^2 dt
> \\end{equation}
>
> which ensures maximum energy compaction such that the majority of the energy of a normal signal is concentrated in a small subset of low-frequency coefficients. Parseval’s identity,
>
> \\begin{equation}
> \\int |x(t)|^2 dt = \\sum_{k=1}^{N} |c\_k|^2
> \\end{equation}
>
> further confirms that the time-domain energy is fully preserved in the frequency-domain coefficient energy. Under normal conditions, the signal $x\_{\text{normal}}(t)$ can be reconstructed using very few Fourier coefficients, whereas anomaly-induced deviations $x\_{\text{anomaly}}(t)$ require additional or unusually distributed coefficients. Thus, the presence and intensity of anomalies can be directly inferred from deviations in spectral compactness.
>
> Fourier basis functions also exhibit strict orthogonality, which is defined as
>
> \\begin{equation}
> \\int \\phi\_k(t) \\phi\_j^{*}(t) dt = 0 \\quad \\text{for } k \\ne j
> \\end{equation}
>
> This ensures that each coefficient represents an independent component of the signal’s frequency structure, meaning anomaly-related distortions can be separated from normal patterns without interference. Such orthogonality allows anomaly scoring to be computed without cross-term interactions that could obscure detection.
>
> In contrast, B-spline functions approximate a signal using
>
> \\begin{equation}
> \\hat{x}(t) = \\sum_{k=1}^{N} c\_k B\_k(t)
> \\end{equation}
>
> but since B-spline functions are non-orthogonal, the coefficients $c\_k$ are statistically dependent. A perturbation in one region of the signal affects multiple coefficients, causing anomaly-related changes to propagate throughout the approximation and complicating separation. Chebyshev polynomials, defined as
>
> \\begin{equation}
> T\_k(t) = \\cos(k \\arccos(t)), \\quad t \\in [-1, 1]
> \\end{equation}
>
> are orthogonal under a weighted inner product but optimized to minimize the maximum approximation error (minimax criterion) rather than MSE. Their equiripple behavior may cause localized anomalies to produce widespread oscillations across coefficients. This makes anomaly localization and separation more difficult. For this reason, Chebyshev-based expansions are commonly adopted in time-series forecasting applications (e.g., TimeKAN) rather than anomaly detection.
>
> Therefore, the use of Fourier basis functions in KANomaly is not based on assumptions about the frequency location of anomalies but on mathematical guarantees of optimal energy compaction and coefficient independence in the $L^2$ space. These properties allow normal patterns to be efficiently represented using a minimal number of coefficients while enabling anomaly-induced deviations to be identified as distinct residual spectral energy. When integrated into the Fourier-KAN Mixer, these spectral characteristics are further enhanced through the model’s channel-patch-temporal mixing architecture, supporting precise localization of anomaly effects across multiple scales and variable interactions.
>
> In summary, Fourier bases provide the most appropriate functional representation for anomaly detection due to minimum mean square approximation error and strict orthogonality, which jointly support efficient normal pattern modeling and statistically independent separation of anomaly components. Alternative bases such as B-spline functions and Chebyshev polynomials lack these combined properties, resulting in reduced anomaly separability. Consequently, the choice of Fourier basis functions is mathematically justified and directly contributes to the anomaly detection capabilities demonstrated in the proposed framework.

---

> ### Comment · Reviewer_7HXw · 2025-11-27
>
> I thank the authors for their comprehensive and convincing response. The additional experiments—including comparisons with CATCH, event-based metrics, and detailed efficiency analysis—have directly and effectively addressed my primary concerns.
>
> The results under stricter metrics confirm the robustness of the advantage. The dramatic efficiency gain over CATCH is particularly compelling. The theoretical justification for Fourier bases solidifies the design choice.
>
> The manuscript has been substantially improved and now clearly demonstrates a meaningful contribution. My concerns have been alleviated.

---

> > ### Author Response · Authors · 2025-11-27
> > **Appreciation for the Reviewer’s Kind Response**
> >
> > We sincerely appreciate the reviewer’s positive and thoughtful feedback. The reviewer’s insightful and considerate comments have greatly contributed to the improvement of our manuscript and helped us strengthen the clarity and impact of our contribution. We would like to once again express our gratitude for the reviewer’s valuable time and effort in providing such constructive feedback.
> >
> > Sincerely,
> >
> > Authors

---

### Official Review · Reviewer_dVHY · 2025-10-29

**Soundness:** 2
**Presentation:** 2
**Contribution:** 2
**Rating:** 4
**Confidence:** 3

**Summary:**

The paper proposes KANomaly, an unsupervised model for multivariate time series anomaly detection that integrates KANs with a coarse-to-fine multi-scale patching strategy and a Fourier-KAN Mixer module. The core idea is to explicitly model frequency-domain characteristics—such as periodicity shifts and spectral disturbances—through learnable Fourier basis functions within the KAN framework, which is theoretically grounded in the Kolmogorov–Arnold representation theorem. The multi-scale patching enables simultaneous detection of both short-term point anomalies and long-term pattern anomalies, while the mixer module captures complex interdependencies across channel, patch, and temporal dimensions. The authors evaluate KANomaly on five real-world benchmarks, reporting improvements over current methods.

**Strengths:**

•	The integration of Fourier basis functions into KAN for MTSAD is a meaningful extension of recent KAN-based approaches, particularly as most prior KAN applications focus on forecasting rather than anomaly detection.
•	Comprehensive empirical validation: The paper includes extensive experiments across five standard datasets, ablation studies, efficiency analysis, and evaluation under multiple metric paradigms (point-adjusted, range-based, and affiliation-based), which strengthens the credibility of the claims.
•	Clear methodological pipeline: The coarse-to-fine multi-scale patching strategy combined with dimension-wise mixing (channel → patch → temporal) is well-motivated and implemented with attention to reconstruction fidelity and anomaly localization.

**Weaknesses:**

- The paper asserts that “existing approaches struggle to model subtle anomalies and detect diverse patterns” because they “rarely integrate explicit frequency-domain representations,” but this claim is not rigorously supported. In fact, several recent works explicitly incorporate frequency-domain modeling for time series tasks, including anomaly detection. The authors acknowledge some of these in Related Work but fail to clearly differentiate KANomaly’s functional integration of frequency information from these predecessors. The motivation appears overstated.

- The criticism that current models “struggle to detect subtle anomalies” remains qualitative and lacks concrete examples or failure cases from existing SOTA models. For instance, the paper could have included a comparative case study (e.g., on a synthetic or real anomaly where spectral features are decisive) showing that Transformer- or reconstruction-based baselines miss anomalies that KANomaly captures—thereby grounding the claimed advantage in observable behavior rather than architectural assumptions.

- While the use of Fourier-KAN is central, the ablation study shows that replacing Fourier-KAN with an MLP causes performance drops—but it does not rule out whether a well-designed frequency-aware MLP (e.g., inspired by FITS or FreTS) could achieve similar gains. The contribution would be stronger if the authors clarified whether the performance gain stems from the KAN structure itself (learnable univariate functions per edge) or simply from explicit Fourier feature learning, which could potentially be implemented in other frameworks.

**Questions:**

1. Could the authors clarify what constitutes “explicit functional integration” in their framework, and how KANomaly’s use of Fourier basis functions within KAN provides a qualitatively different capability compared to these prior frequency-aware models?

2. Could the authors provide further analysis or ablation (e.g., Fourier features + MLP) to isolate the contribution of the KAN structure versus the frequency representation?

---

> ### Author Response · Authors · 2025-11-22
> **Response to Reviewer dVHY (Part 1)**
>
> We sincerely thank the reviewer for the thoughtful and constructive feedback. We deeply appreciate the time and effort devoted to carefully reviewing our paper and providing detailed suggestions. Below, we address the reviewer’s comments and questions in detail.
>
> >**W1 & Q1: On the “Explicit Functional Integration” and Its Distinction from Prior Frequency-Aware Methods.**
>
> The reviewer raised an important point regarding the distinction between KANomaly’s explicit functional integration and previous frequency-aware approaches.
> Several prior models have indeed leveraged frequency-domain information.
> However, the key differentiation lies in how frequency-domain information is utilized and where it is integrated within the network.
>
> Conventional frequency-aware approaches such as Autoformer, FEDformer, TimesNet, TFMAE, FITS, and FreTS apply a Fourier transform in a global manner over the entire time window.
> The resulting frequency coefficients are treated as static features and subsequently provided as inputs to downstream backbones such as MLPs or Transformers.
> For example, Autoformer, FEDformer, and TimesNet extract only the top K frequency components, TFMAE uses magnitude spectra alone, discarding phase information, FITS removes high-frequency bands through cut-off filtering, and FreTS transforms the input sequence to the frequency domain before passing the real and imaginary components to a frequency-domain MLP.
> In all these cases, frequency-domain representations are employed at the feature level as pre-computed descriptors without modifying the functional form of the network.
> Consequently, these models remain bound to a fixed frequency resolution and cannot dynamically adapt to non-stationary or locally varying spectral characteristics.
>
> In contrast, KANomaly performs explicit functional integration of frequency information within the Kolmogorov–Arnold Network (KAN) framework.
> Rather than treating Fourier features as static inputs, learnable Fourier basis functions are embedded directly into the network’s univariate mappings, redefining the activation space itself.
> Each edge function in KANomaly replaces the conventional spline basis with a learnable Fourier series:
>
> \begin{equation}\phi_F(\mathbf{x}) = \sum_{i=1}^{d} \sum_{k=1}^{g} \left(a_{ik} \cdot \cos(k x_i) + b_{ik} \cdot \sin(k x_i) \right)\end{equation}
>
> where $\mathbf{x} \in \mathbb{R}^d$ denotes the input vector, $a_{ik}$ and $b_{ik}$ are learnable coefficients, and $g$ controls the number of Fourier components.
> This construction enables each functional mapping to adaptively determine which frequency bases, such as $\sin(kx)$ or $\cos(kx)$, best reconstruct local temporal variations.
>
> Thus, KANomaly fundamentally differs from methods that employ Fourier series only at the input level before feeding signals into MLP or attention backbones. Instead, the proposed model integrates learnable Fourier basis functions directly within the functional mappings, redefining the activation space to be Fourier-parametric.
>
> This architectural property embodies the essence of explicit functional integration.
> In KANomaly, time-domain and frequency-domain representations are co-learned within a unified, parameterized function space.
> Such integration endows the model with a qualitatively different capability, allowing it to flexibly approximate non-stationary patterns characterized by periodicity, phase distortion, and fine-grained spectral perturbations whose spectral signatures evolve across patches, channels, and temporal segments.
> These characteristics are typically difficult to capture for models that rely solely on feature-level frequency representations.

---

> ### Author Response · Authors · 2025-11-22
> **Response to Reviewer dVHY (Part 2)**
>
> >**W2 & Q1: Comparison with SOTA Models on the Detection of Subtle Spectral Anomalies and Empirical Evidence of Qualitatively Different Capability.**
>
> The reviewer raised a valuable point regarding the need for concrete evidence demonstrating that the proposed model effectively captures subtle anomalies rather than relying on architectural assumptions. In particular, it was suggested that comparative case studies should be included to illustrate how existing SOTA models fail to detect the anomalies that KANomaly successfully captures. The reviewer also requested a clearer explanation of whether and how KANomaly provides a qualitatively different capability compared to prior frequency-aware models.
>
> In this context, subtle anomalies refer to spectral or contextual deviations such as Point Contextual, Pattern Shapelet, and Pattern Seasonal types, as illustrated in Figure 1 (a) and Figure 3 (Left) of the paper.
> Unlike Point Global anomalies, which lie far outside the normal distribution, these anomalies often remain within statistically normal ranges and are therefore undetectable by conventional distribution-based or IQR-based thresholds.
> Figure 1 (a) conceptually demonstrates that such subtle irregularities become more separable in the frequency domain, where periodic shifts, phase distortions, and localized spectral perturbations emerge clearly after spectral decomposition.
> As shown in Figure 3, the synthetic dataset contains diverse anomaly types, including Point (Global and Contextual) and Pattern (Shapelet, Seasonal, and Trend) forms, each characterized by distinct spectral distortions.
> This observation motivates the design of KANomaly, which leverages frequency-domain cues to identify these nuanced deviations more effectively than temporal-only architectures.
>
> To support this point, additional experiments were conducted on the synthetic TODS dataset [1], which contains both point-wise and pattern-wise anomalies characterized by distinct spectral distortions. The comparison included representative frequency-aware baselines such as TFMAE and FITS, and the results are provided in "Appendix O. Additional Visual Analysis" (see Figure 6).
> As shown in the additional visualization, frequency-aware baselines such as TFMAE and FITS produce inconsistent and diffuse anomaly scores, often failing to localize subtle contextual or pattern-level deviations.
> In contrast, KANomaly yields sharper, temporally aligned responses across Point Contextual, Pattern Shapelet, and Pattern Seasonal cases.
> These results demonstrate that the proposed model effectively identifies anomalies governed by fine-grained spectral variations that are overlooked by existing methods.
>
> Collectively, the supplementary evidence alleviates the reviewer’s concern by grounding the claimed advantage in observable behavior rather than architectural assumptions, and it also addresses the concern regarding how the proposed model provides a qualitatively different capability compared to previous frequency-aware approaches.
>
> [1] Lai et al., "Revisiting time series outlier detection: Definitions and benchmarks." NeurIPS 2021

---

> ### Author Response · Authors · 2025-11-22
> **Response to Reviewer dVHY (Part 3)**
>
> >**W3 & Q2: On the Contribution of the KAN Structure versus Fourier Feature Learning.**
>
> The reviewer raised an important point regarding whether the observed performance improvements of KANomaly stem primarily from the Kolmogorov–Arnold Network (KAN) structure itself or from the explicit incorporation of Fourier features, which could, in principle, be implemented in other frameworks. To address this concern, the description of the existing ablation study has been reinforced, and an additional ablation study was conducted to isolate the respective contributions of the KAN structure and the frequency representation.
>
> As shown in Table 2 of the main paper, replacing the Fourier-KAN block with (1) a standard MLP, (2) Chebyshev-KAN, or (3) vanilla KAN leads to consistent performance degradation across all benchmarks, with an average F1 drop of −9.21%, −9.77%, and −9.59%, respectively. These results indicate that the integration of Fourier basis functions within the KAN mapping plays a crucial role in enhancing representational capacity beyond what is achievable through either KAN or Fourier modeling alone. This also implies that the observed improvement does not simply originate from the KAN structure itself (i.e., the learnable univariate functions per edge), but from the functional-level integration of Fourier basis functions within the KAN framework.
>
> To further validate this point, an additional ablation “Fourier features + MLP” was implemented following the reviewer’s suggestion. In this variant, the Fourier-KAN in KANomaly was replaced with a pure MLP (Linear → GELU → Linear) that operates directly on Fourier-transformed inputs. The model first converts each input sequence into the frequency domain via the Fourier Transform, uses the transformed results as the model inputs, and then reconstructs the outputs to the time domain using the inverse Fourier Transform. As presented in Table 2, this configuration yields the lowest performance among all variants, with an average F1 score of 81.11% (−15.31% drop relative to KANomaly).
>
> | Variations | MSL | PSM | SMAP | SMD | SWaT | AVG F1 | Difference |
> |-----------|-----|-----|------|-----|------|--------|------------|
> | **KANomaly (ours)** | **95.64** | **98.67** | **97.46** | **91.74** | **98.59** | **96.42** | — |
> | (1) w/o Fourier-KAN (→ MLP) | 85.48 | 97.66 | 68.43 | 87.78 | 96.68 | 87.21 | -9.21% |
> | (2) w/o Fourier-KAN (→ Cheby-KAN) | 83.97 | 97.22 | 69.24 | 85.58 | 97.22 | 86.65 | -9.77% |
> | (3) w/o Fourier-KAN (→ Vanilla-KAN) | 82.40 | 97.69 | 68.39 | 88.27 | 97.40 | 86.83 | -9.59% |
> | (4) w/o Multi-Scale Patching (Fixed Patch Size) | 86.60 | 96.94 | 96.09 | 83.55 | 97.66 | 92.17 | -4.25% |
> | (5) w/o Patching | 88.98 | 98.14 | 90.14 | 80.29 | 95.36 | 90.58 | -5.84% |
> | (6) Fine-to-Coarse Patch (Inverse Patch Scale) | 88.54 | 95.83 | 94.30 | 82.22 | 97.41 | 91.66 | -4.76% |
> | (7) w/o Channel Mixing | 85.77 | 98.16 | 69.30 | 86.58 | 96.87 | 87.34 | -9.08% |
> | (8) w/o Patch Mixing | 86.85 | 95.38 | 76.01 | 89.54 | 95.77 | 88.71 | -7.71% |
> | (9) w/o Temporal Mixing | 87.27 | 97.26 | 74.90 | 90.86 | 96.42 | 89.34 | -7.08% |
> | (10) Fourier features + MLP | 80.72 | 89.40 | 68.24 | 84.84 | 82.37 | 81.11 | -15.31% |
>
> This substantial degradation demonstrates that the performance gain of KANomaly cannot be attributed solely to frequency feature learning. Rather, it arises from the functional-level integration of learnable Fourier basis functions within the KAN structure, where each edge learns adaptive, univariate Fourier expansions that capture localized periodicity, phase distortion, and fine-grained spectral perturbations. In contrast, the “Fourier features + MLP” model treats frequency information merely as static input descriptors and lacks the functional adaptability necessary for dynamic spectral modulation.
>
> In summary, these results confirm that KANomaly’s superior performance stems from the synergy between (1) Fourier-based frequency-domain representation and (2) the expressive power of the KAN framework in modeling nonlinear, localized functional interactions. The KAN structure is therefore not a simple backbone but a key mechanism enabling explicit functional learning in the frequency domain.

---

### Author Response · Authors · 2025-11-26
**Summary of Revisions**

We sincerely thank all reviewers for dedicating their valuable time and effort to carefully evaluate our work. Your insightful and constructive feedback has been instrumental in refining the manuscript and enhancing its contribution. We are deeply grateful for your thoughtful comments, which have significantly improved the clarity and overall quality of our work.

We have carefully addressed all questions and concerns raised by the reviewers, and have thoroughly incorporated all suggested improvements into the revised manuscript. All modifications have been meticulously implemented and are highlighted in blue for clarity.

**The primary revisions made to the updated manuscript are summarized as follows:**

- **Addition of KAN-AD baseline in the multivariate scenario (Reviewer qJtT, PNn3):**
    - We have additionally included the KAN-AD baseline adapted for the multivariate configuration to enable a more comprehensive evaluation. The revised results can be verified in **Section 6 Experiments**, where performance differences are thoroughly analyzed.

- **Inclusion of the CATCH baseline (Reviewer 7HXw):**
    - To address the reviewer’s suggestion, we incorporated CATCH as an additional frequency-based baseline. The extended comparative analysis is available in **Section 6 Experiments**.

- **Clarification of differences from prior frequency-aware models (Reviewer dVHY):**
    - We have elaborated on how our approach differs from prior models by highlighting the concept of explicit functional integration rather than feature-level utilization. A detailed explanation is presented in **Appendix C**, where we distinguish our method from existing frequency-based techniques.

- **Additional Visual Analysis (Reviewer dVHY, PNn3):**
    - We have added visual comparisons against state-of-the-art models to demonstrate the capability of KANomaly in detecting subtle spectral anomalies. This analysis also addresses the concern that the TODS dataset may be too simple for deep learning models and justifies the validity of the presented visualization. Additional visual analysis is provided in **Appendix O**.

- **Inclusion of "Fourier Features + MLP" in Ablation Study (Reviewer dVHY):**
    - To disentangle the contribution of the KAN architecture from that of frequency representation, we introduced an ablation variant using directly applied Fourier features with an MLP. The results, shown in **Section 5.2 Model Analysis**, demonstrate a significant performance degradation in this configuration. These findings support that our improvement is attributable to functional-level integration rather than frequency preprocessing alone.

- **Comparison with existing KAN-based time series anomaly detection models and clarification of contribution (Reviewer PNn3):**
    - We have strengthened the explanation of how our method fundamentally differs from KAN-AD and MTAD-Kanformer. The distinction and contribution analysis are presented in **Appendix E**.

- **Additional Efficiency Analysis Compared to Baseline Models (Reviewer 7HXw, qJtT):**
    - To further support the efficiency of KANomaly, we expanded the analysis across all benchmark models. These results are provided in **Appendix P.1**. The update provides clearer evidence of our model’s favorable trade-off between accuracy, speed, and parameter size.

- **Additional Efficiency Analysis Compared to Ablation Models (Reviewer qJtT):**
    - We have also added efficiency comparisons among ablation variants to transparently analyze the efficiency of each architectural component. The detailed quantitative results are provided in **Appendix P.2**.

- **Future Work on Interpretability (Reviewer qJtT):**
    - In response to the reviewer’s interest in interpretability, we expanded the discussion on future research directions. This update, included in **Section 6 Conclusion and Future Work**, outlines potential approaches leveraging the inherent interpretability of KAN architectures.

- **Theoretical justification for selecting Fourier basis functions over alternative basis functions (Reviewer 7HXw):**
    - We have added a theoretical justification clarifying why Fourier bases are suitable for KANomaly in the context of time series anomaly detection. This explanation is presented in **Appendix D**. We expect that this addition will help address reviewer concerns regarding the rationale behind the choice of basis functions.

We sincerely hope that the revisions adequately address the reviewers’ concerns, enhance the clarity of the manuscript, and further strengthen its overall contribution. Please do not hesitate to let us know if any additional questions or comments arise. We would greatly appreciate any further feedback from the reviewers.

Sincerely,

Authors

---

### Author Response · Authors · 2025-11-30
**Request for Final Consideration in Light of the Limited Discussion Period**

Dear Area Chair,

We regret that the recent OpenReview incident occurred and would like to express our full support for the corrective measures taken by the ICLR organizing committee. We sincerely appreciate their continued dedication to maintaining fairness throughout the review process.

Owing to the limited duration of the discussion period, **only one (7HXw) of the four reviewers actively participated in the discussion.** Subsequently, the reviewer 7HXw provided positive feedback on our work and updated their Rating accordingly, raising it from 4 to 6.

We believe that, had the discussion period been longer, the remaining reviewers would likewise have had sufficient opportunity to resolve their concerns upon reviewing our responses and would have been likely to form a more positive assessment of our work. Furthermore, we trust that the AC, upon reviewing our responses, would also find our clarifications reasonable.


To provide additional context, we summarize below the positive remarks from the reviewers regarding our work:

- **Reviewer dVHY**

    - *The integration of Fourier basis functions into KAN for MTSAD is a meaningful extension of recent KAN-based approaches, particularly as most prior KAN applications focus on forecasting rather than anomaly detection.*

    - *Comprehensive empirical validation: The paper includes extensive experiments across five standard datasets, ablation studies, efficiency analysis, and evaluation under multiple metric paradigms (point-adjusted, range-based, and affiliation-based), which strengthens the credibility of the claims.*

    - *Clear methodological pipeline: The coarse-to-fine multi-scale patching strategy combined with dimension-wise mixing (channel → patch → temporal) is well-motivated and implemented with attention to reconstruction fidelity and anomaly localization.*

- **Reviewer 7HXw**

    - *Combines Fourier analysis with KAN in a novel way for Multivariate Time Series Anomaly Detection, a direction not previously explored. The approach elegantly and rationally combines the intuitive physical interpretation of Fourier analysis—namely, its ability to capture periodicity and spectral anomalies—with the universal function-approximation power of Kolmogorov–Arnold Networks.*

    - *Table 2’s ablation systematically swaps Fourier-KAN for MLP/Chebyshev/Vanilla-KAN and reverses the coarse-to-fine order, convincingly proving the current design optimal.*

    - *The additional experiments—including comparisons with CATCH, event-based metrics, and detailed efficiency analysis—have directly and effectively addressed my primary concerns.*

    - *The results under stricter metrics confirm the robustness of the advantage. The dramatic efficiency gain over CATCH is particularly compelling. The theoretical justification for Fourier bases solidifies the design choice.*

    - *The manuscript has been substantially improved and now clearly demonstrates a meaningful contribution. My concerns have been alleviated.*

- **Reviewer qJtT**

    - *Architectural Innovation：Rather than simply applying KAN to multivariate time series anomaly detection, the paper tailors the architecture to characteristics of different anomaly types and temporal structures. The use of multi-scale patching and mixer-based aggregation seems well-motivated.*

    - *Clarity of Presentation and Visualization: The manuscript is clearly written and easy to read. The figures are detailed and informative, effectively supporting readers’ understanding of the architectural components.*

    - *There are relatively few KAN-based approaches for anomaly detection, and I find the architectural extensions in this work meaningful.*

- **Reviewer PNn3**

    - *Clear and well-designed figures that enable readers to quickly grasp the motivation and methodological details.*

    - *Comprehensive equations that clearly convey the computational aspects of the model.*


During the discussion period, we conducted additional experiments and analyses, which led to the **inclusion of 8 additional pages beyond the originally submitted manuscript.** All updates are provided in detail in the Summary of Revisions below.

We hope that our efforts will be instrumental in assisting the AC in reaching a final decision. We would be grateful if given the opportunity to present our work at the conference.

We would like to once again express our sincere gratitude to the AC for their time, effort, and continued dedication.

Sincerely,

Authors

---

### Meta-Review · Area_Chair_iMUG · 2026-01-04

**Summary:**

There are several concerns about this work, centering on efficiency claims, evaluation measures used, omitted baselines, and technical novelty/difference to prior works. Indeed, the authors did a good job addressing the comments and providing new experiments. However, some of the comments would need substantial more discussion to be addressed (e.g., novelty part) or if chosen evaluation measures are appropriate (e.g., someone could suggest the use of VUS-based measures) or if the datasets are appropriate (e.g., TSB-AD is another benchmark). Even if we assume some reviewers would raise their scores, the overall score would still be a borderline case and uncertain of its acceptance. I hope the reviewers would find the feedback useful and find an alternative venue for their interesting work.

**Reviewer Concerns:**

Outstanding concerns have to do with the lack of discussion, e.g., if novelty claims provided are sufficient, if new evaluation measures are indeed appropriate or others are needed, or other datasets etc. New baselines were added but again it's unclear if more/different baselines would still be necessary.

**Reviewer Scores:**

The scores are leaning towards rejection. In rebuttal, the authors did a good job addressing several comments and someone could assume some of the reviewers would raise their scores. However, even after such assumption, the overall score wouldn't be sufficient to justify with certainty an acceptance.

---

### Decision · Program_Chairs · 2026-01-26

Reject